# BUTTERFLY EFFECTS OF SGD NOISE:
# ERROR AMPLIFICATION IN BEHAVIOR CLONING AND AUTOREGRESSION

**Adam Block**
Department of Mathematics
MIT
Cambridge, MA, 02139
ablock@mit.edu

**Cyril Zhang, Dylan Foster & Akshay Krishnamurthy**
Microsoft Research
New York, NY, 10012
{cyrilzhang,dylanfoster,akshaykr}@microsoft.com

**Max Simchowitz**
MIT
Cambridge, MA, 02139
msimchow@mit.edu

## ABSTRACT

This work studies training instabilities of behavior cloning with deep neural networks. We observe that minibatch SGD updates to the policy network during training result in sharp oscillations in long-horizon rewards, despite negligibly affecting the behavior cloning loss. We empirically disentangle the statistical and computational causes of these oscillations, and find them to stem from the chaotic propagation of minibatch SGD noise through unstable closed-loop dynamics. While SGD noise is benign in the single-step action prediction objective, it results in catastrophic error accumulation over long horizons, an effect we term *gradient variance amplification* (GVA). We show that many standard mitigation techniques do not alleviate GVA, but find an exponential moving average (EMA) of iterates to be surprisingly effective at doing so. We illustrate the generality of this phenomenon by showing the existence of GVA and its amelioration by EMA in both continuous control and autoregressive language generation. Finally, we provide theoretical vignettes that highlight the benefits of EMA in alleviating GVA and shed light on the extent to which classical convex models can help in understanding the benefits of iterate averaging in deep learning.

## 1 INTRODUCTION

Deep neural networks are increasingly used in machine learning tasks that contain *feedback loops* as a defining characteristic: outputs of language models depend on previously predicted tokens (Vaswani et al., 2017), recommendation systems influence the users to whom they give suggestions (Krauth et al., 2020; Dean & Morgenstern, 2022), and robotic policies take actions in reactive control environments (Ross & Bagnell, 2010; Laskey et al., 2017). Because these tasks are so complex, it is standard practice to optimize surrogate objectives, such as next-token prediction, that typically ignore feedback loops altogether (Pomerleau, 1988; Vaswani et al., 2017; Florence et al., 2022).

When training deep models by gradient updates on the surrogate objective, surrogate performance often improves more or less monotonically as training progresses. At the same time, successive iterates can exhibit wild variations in their performance on the task of interest. Because it is often impractical to evaluate the desired performance metric at multiple checkpoints, these oscillations imply that we have high risk of selecting and deploying a poor policy. Thus, in order to determine best practices, we must first understand whether *better training* or *better data* will fix these instabilities. This leads us to ask:

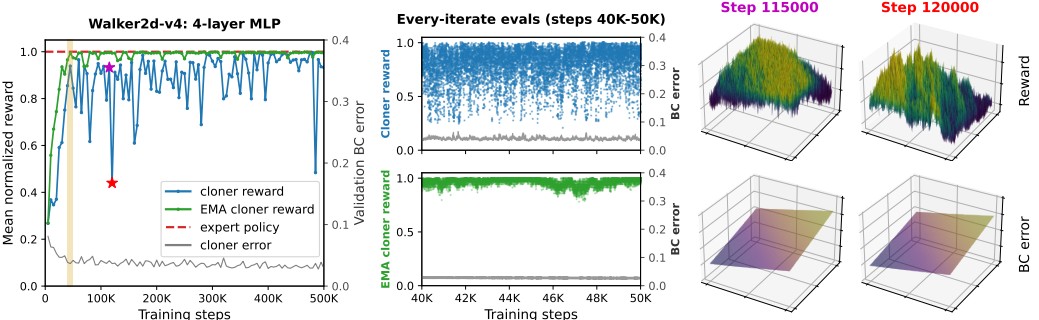

Figure 1: Typical reward instabilities over long-horizon ($H = 1000$) rollouts of neural behavior cloners for the `Walker2d-v4` MuJoCo locomotion task. *Left:* Rollout rewards (blue training curves) oscillate dramatically over the course of training (evaluated every 5000 iterations), while BC loss is stable. *Center:* Zoomed-in view of the highlighted region in (left). Large reward fluctuations are evident even between consecutive gradient iterates. *Right:* Exhaustive evaluation of small neighborhoods (in stochastic gradient directions) around iterates 115K and 120K, revealing a fractal reward landscape $\theta \mapsto J_H(\pi_\theta)$; this jaggedness is invisible in the 1-step behavior cloning objective $\ell_{\text{BC}}(\pi_\theta)$. Iterate averaging (EMA) drastically mitigates these effects (green training curves). Details are provided in Appendix C.1.1.

> *What causes instabilities in learning systems with feedback loops? To what extent can they be mitigated by algorithmic interventions alone, without resorting to collecting additional data?*

We explore this question in the context of behavior cloning (BC), a technique for training a policy to optimize a multi-step objective in a purely offline manner. This is achieved by introducing a surrogate loss function $\ell_{\text{BC}}$ (*behavior cloning loss*) that measures the distance between actions generated by some *expert policy* $\pi_{\theta^\star}$ and those taken by the learner's policy, and then minimizing $\ell_{\text{BC}}$ over an offline dataset of expert trajectories. BC is sufficiently broad as to capture important tasks ranging from robotics and autonomous driving (Pomerleau, 1988; Codevilla et al., 2018; Chi et al., 2023) to autoregressive language generation (Chang et al., 2023a), and is popular in practice due to its simplicity and purely offline nature.

Our starting point is to observe that behavior cloning with deep neural networks exhibits **training instabilities** in which the multi-step objective ($J_H$), or *rollout reward*, of nearby checkpoints oscillates wildly during training, despite a well-behaved validation loss for $\ell_{\text{BC}}$, even at the single iterate frequency; Figure 1 exhibits this phenomenon for a sample training curve of a behavior cloning policy in the `Walker2d-v4` MuJoCo locomotion task (Towers et al., 2023; Todorov et al., 2012). This oscillatory behavior is clearly undesirable; we cannot differentiate between low- and high-quality iterates based on (validation loss for) $\ell_{\text{BC}}$, and thus cannot reliably select a high-quality policy.

With regard to the final performance of BC, it is well understood that scarce or low-quality data can lead to statistical challenges and consequent performance degradation of imitator policies; unsurprisingly, better data often improves the quality of a learned policy. Unfortunately, existing approaches to obtaining better data require either interactive access to the demonstrating expert (Ross & Bagnell, 2010; Laskey et al., 2017) or additional side information (Pfrommer et al., 2022; Block et al., 2023a); these interventions may be costly or impossible in many applications. Thus, in this work we treat the data generating process as fixed and aim to investigate whether we can mitigate oscillations and improve the performance of BC solely through the application of better algorithmic choices.

## 1.1 CONTRIBUTIONS

In this paper, we aim to diagnose and ameliorate instabilities in behavior cloning that arise from training on the surrogate cost alone in the purely offline setting. Our findings are as follows.

**Diagnosis of rollout oscillations: gradient variance amplification.** In Section 3, we conduct an extensive empirical study (278 distinct interventions) of BC in continuous control tasks and inves-

tigate the effects that architecture, regularization, and optimization interventions have on training instability. We identify the presence of training oscillations and attribute them to *gradient variance amplification* (GVA): the propagation of minibatch SGD noise through closed-loop dynamics, leading to catastrophic error amplification resembling **butterfly effects** in chaotic systems. We ablate away much of the statistical difficulty, so that the presence of oscillations suggests that GVA is an *algorithmic* rather than *statistical* pathology.

**Mitigating GVA: stabilizers for unstable optimizers.** In Section 4, we investigate mitigations for GVA. Because GVA is caused by variance in the stochastic gradients, it can be ameliorated with variance reduction. Indeed, we observe (Section 3.2) that i) aggressively decaying the learning rate, and ii) greatly increasing the batch size through gradient accumulation, both have positive effects on the stability of training. Unfortunately, both of these interventions come at a great increase in compute cost. As such, our most significant finding (Section 4.1) is that *iterate averaging* by taking an Exponential Moving Average (EMA) of the optimization trajectory (Polyak & Juditsky, 1992; Ruppert, 1988), stabilizes training and mitigates GVA across a wide range of architectures and tasks, with essentially no downsides. While iterate averaging is popular in many deep learning research communities, this paper exposes iterate averaging as an *essential* design consideration when training any deep model in the presence of feedback loops.

**A preliminary study of GVA in language generation.** In Section 4.2, we broaden our focus by considering autoregressive sequence models. Our findings suggest that **unstable optimizers, when stabilized with iterate averaging to mitigate GVA, do not need full learning rate decay**, entailing potential computational and statistical benefits for training language models. For this reason, we suggest that EMA and related filters be designated as *stabilizers* in their own right and incorporated into deep learning pipelines in the same vein as modern optimizers and schedulers.

**The applicability of convex theory.** In Section 4.3, we complement our empirical results with theoretical vignettes. While the benefits of large learning rates cannot be explained in a convex setting, we demonstrate that—conditional on using theoretically suboptimal learning rates—stochastic convex optimization provides useful intuition for the causes and mitigations of GVA in deep learning. With our empirical results, these findings add to a line of work on surprising near-convex behavior in deep learning (Sandler et al., 2023; Frankle et al., 2020; Fang et al., 2022; Schaul et al., 2013).

## 1.2 RELATED WORK

Understanding and mitigating the effects of error amplification in behavior cloning has been the subject of much empirical work (Ross & Bagnell, 2010; Laskey et al., 2017), but most approaches use potentially impractical online query access to the expert policy; instead, we focus on a purely offline setting.

Complicated value function landscapes and their effect on training have been investigated in the context of planning in RL, with Dong et al. (2020) investigating natural examples of fractal reward functions, Wang et al. (2021) examining the instabilities arising from poor representations, and Emmons et al. (2021); Chang et al. (2023a) observing the fact that $\ell_{BC}$ is a poor proxy for $J_H$. To the best of our knowledge, there has not been a systematic study of training instability in the sense of rollout reward oscillation of nearby checkpoints.

In the context of stochastic optimization and optimization for deep learning, many previous works have attempted to reduce variance in theory (Polyak & Juditsky, 1992; Ruppert, 1988) and practice (Izmailov et al., 2018; Busbridge et al., 2023; Kaddour, 2022; Kaddour et al., 2023). Of particular note is Sandler et al. (2023), which demonstrates (empirically and in a toy theoretical setting) a form of equivalence between learning rate decay and iterate averaging. Our focus is not on variance reduction *per se*, but rather on the propagation of ,variance through unstable feedback loops. We expand on the relationship between our work and Sandler et al. (2023) and discuss other related work in Appendix B.

## 2 PRELIMINARIES

**MDP formalism.** We let $\mathcal{M} = (\mathcal{S}, \mathcal{A}, P, r, H, \nu)$ denote a finite-horizon Markov decision process (MDP), where $\mathcal{S}$ is an abstract state space, $\mathcal{A}$ is an abstract action space, $P : \mathcal{S} \times \mathcal{A} \to \mathbf{\Delta}(\mathcal{S})$ is

a Markov transition operator. We denote by $r : \mathcal{S} \times \mathcal{A} \to [0,1]$ a reward function and $H \in \mathbb{N}$ is the length of the horizon. Because we focus on continuous control tasks, we follow the notational conventions of control theory, denoting states by $\mathbf{x}$ and actions by $\mathbf{u}$. We let $\nu \in \mathbf{\Delta}(\mathcal{S})$ denote the initial distribution such that a trajectory from $\mathcal{M}$ consists of $\mathbf{x}_1 \sim \nu$ and $\mathbf{x}_{h+1} \sim P(\cdot \mid \mathbf{x}_h, \mathbf{u}_h)$ for all $h$.

The learner has access to a class of policies $\pi : \mathcal{S} \times \Theta \to \mathbf{\Delta}(\mathcal{A})$, where $\Theta$ is the *parameter* space and $\pi_{\boldsymbol{\theta}} : \mathcal{S} \to \Delta(\mathcal{A})$ is the policy induced by parameter $\boldsymbol{\theta} \in \Theta$. Given a policy $\pi_{\boldsymbol{\theta}}$, we denote its expected cumulative reward by $J_H(\pi_{\boldsymbol{\theta}}) = \mathbb{E}[\sum_{h=1}^{H} r(\mathbf{x}_h, \mathbf{u}_h)]$ where $\mathbf{u}_h \sim \pi_{\boldsymbol{\theta}}(\cdot|\mathbf{x}_h)$ and the expectation is with respect to both the transition dynamics of $\mathcal{M}$ and the possible stochasticity of the policy. Our experiments focus on MDPs whose transition operators $P$ are deterministic, i.e., there exists a function $f : \mathcal{S} \times \mathcal{A} \to \mathcal{S}$ such that $\mathbf{x}_{h+1} = f(\mathbf{x}_h, \mathbf{u}_h)$ for all $h$. In this case the only stochasticity of the system comes from the sampling of the initial state $\mathbf{x}_1 \sim \nu$ (and possibly the policy).

**Imitation learning and behavior cloning.** In imitation learning, we are given an offline data set of $N$ trajectories $\mathcal{D}_{\text{off}} = \{(\mathbf{x}_h^{(i)}, \mathbf{u}_h^{(i)})_{1 \leq h \leq H} \mid 1 \leq i \leq N\}$ generated by an expert policy $\pi_{\boldsymbol{\theta}^\star}$ interacting with the MDP $\mathcal{M}$. In this work, we always consider *deterministic policies*, i.e., where for all $\mathbf{x}$, $\pi_{\boldsymbol{\theta}}(\mathbf{x})$ has support on a single action; in particular this holds for the expert $\pi_{\boldsymbol{\theta}^\star}$. The goal of the learner is to produce a policy $\pi_{\widehat{\boldsymbol{\theta}}}$ that maximizes the expected cumulative reward $J_H(\pi_{\widehat{\boldsymbol{\theta}}})$ over an episode. We focus on the popular *behavior cloning* (BC) framework, where we fix a loss function $\ell_{\text{BC}} : \mathcal{A} \times \mathcal{A} \to \mathbb{R}$ that measures the distance from the actions produced by $\pi_{\boldsymbol{\theta}^\star}$, and learn $\pi_{\widehat{\boldsymbol{\theta}}}$ by attempting to minimize the empirical risk of $\ell_{\text{BC}}$ over $\mathcal{D}_{\text{off}}$; we abuse notation by denoting $\ell_{\text{BC}}(\pi_{\boldsymbol{\theta}}) := \mathbb{E}_{\mathcal{D}_{\text{off}}}[\ell_{\text{BC}}(\pi_{\boldsymbol{\theta}}(\mathbf{x}), \mathbf{u})]$ The basic premise behind behavior cloning is that $\ell_{\text{BC}}$ should be chosen such that if $\ell_{\text{BC}}(\pi_{\widehat{\boldsymbol{\theta}}}) \ll 1$ then $J_H(\pi_{\widehat{\boldsymbol{\theta}}}) \approx J_H(\pi_{\boldsymbol{\theta}^\star})$; that is, imitation of the expert is a surrogate for large cumulative reward. In line with common practice in BC (Janner et al., 2021; Shafiullah et al., 2022; Chi et al., 2023), the imitator policies in our experiments augment the state with the previous action, i.e., $\pi_{\boldsymbol{\theta}} : \mathcal{S} \times \mathcal{A} \to \mathcal{A}$, which can be integrated into the previous formalism by expanding the state space. For the special case of the first state $\mathbf{x}_1$, we always let $\mathbf{u}_0 = \mathbf{0}$.

**Notation.** Throughout the paper, we denote vectors by bold lower case letters and matrices by bold upper case letters.[1] We reserve $\boldsymbol{\theta}$ for a parameter of our policy and $J_H$ for the cumulative reward over a trajectory, omitting $H$ when it is clear from context. For conciseness, we often refer to $J_H$ as the *reward*; the per-step reward function $r$ makes no appearance in the rest of the paper. Given a set $\mathcal{U}$, we let $\mathbf{\Delta}(\mathcal{U})$ denote the class of probability distributions on $\mathcal{U}$.

## 3 DIAGNOSIS OF ROLLOUT OSCILLATIONS: GRADIENT VARIANCE AMPLIFICATION

### 3.1 INSTABILITIES IN BEHAVIOR CLONING OF MUJOCO TASKS

**Experimental setup.** We investigate instabilities in behavior cloning for the {`Walker2d`, `Hopper`, `HalfCheetah`, `Humanoid`, `Ant`}`-v4` environments from the OpenAI Gymnasium (Towers et al., 2023), all rendered in MuJoCo (Todorov et al., 2012). We focus on `Walker2d-v4` for the discussion that follows, and defer detailed discussion of further environments (which exhibit similar behavior) to Appendix C. Our expert is a multilayer perceptron (MLP) trained with Soft Actor Critic (SAC) (Haarnoja et al., 2018) for 3M steps with `stable-baselines3` (Raffin et al., 2021), with out-of-the-box hyperparameters.[2] The *default* imitator is a 4 layer MLP; details are in Appendix C. We examine several widths and depths, as well as Transformer (Vaswani et al., 2017) imitators.

Our first suite of experiments aims to isolate instability from *statistical difficulties*. We set up the experiments to make the behavior cloning problem as easy as possible. First, we focus on the "large-data" regime $N = H = 1000$, in which overfitting with respect to the BC loss $\ell_{\text{BC}}(\pi_{\widehat{\boldsymbol{\theta}}})$ is not a problem (see Figure 1), and thus poor rollout performance for $J_H(\pi_{\widehat{\boldsymbol{\theta}}})$ cannot be blamed on

---

[1]In particular, we denote states by $\mathbf{x}$ and actions by $\mathbf{u}$ in order to emphasize that, in our experiments, they are vectors in Euclidean space.

[2]By default, the Stable-Baselines3 SAC agent is stochastic, but we enforce determinism by selecting the mean action of the resulting policy. This results in negligible degradations to the rewards; see Figure 5.

insufficient data; this removes a typical source of statistical difficulty faced in applying behavior cloning to domains such as robotics (Chi et al., 2023; Pfrommer et al., 2022; Ross & Bagnell, 2010; Laskey et al., 2017). Beyond focusing on the large-data regime, (i) we consider only deterministic dynamics and deterministic experts, and (ii) we include within our default model the same class of MLPs that parameterize the expert policies, ensuring that expressivity is not an issue. As such, we have placed ourselves in perhaps the easiest possible setting for behavior cloning.

In Figure 1 (Left), we compare the evolution of the BC loss $\ell_{\mathrm{BC}}(\pi_{\widehat{\boldsymbol{\theta}}})$ (on a validation set) and reward $J_H(\pi_{\widehat{\boldsymbol{\theta}}})$ for imitator policies in the `Walker2d-v4` MuJoCo locomotion task. In this figure, we observe extreme oscillatory behavior in $J_H(\pi_{\widehat{\boldsymbol{\theta}}})$, juxtaposed with smoothly decaying $\ell_{\mathrm{BC}}(\pi_{\widehat{\boldsymbol{\theta}}})$. In Figure 1 (Middle), we zoom in on the training trajectory between iterates 40K and 50K and observe that the same instability persists even at the *every-iterate* level. Toward identifying what causes these instabilities, Figure 1 (Right) displays an experiment in which we independently sample two stochastic gradients of the training loss at a fixed checkpoint with good rollout reward. Policy weights are then perturbed by small steps in each of the two directions, and we evaluate the resulting reward $J_H(\pi_{\widehat{\boldsymbol{\theta}}})$ over 20 rollouts, along with the BC loss $\ell_{\mathrm{BC}}(\pi_{\widehat{\boldsymbol{\theta}}})$ on a held-out validation set. We see that nearby models vary erratically in terms of rollout performance, but vary smoothly in validation BC loss. These findings are reproduced consistently across the other environments and architectures in Appendix C; thus, we conclude:

(R1) **The reward landscape is highly sensitive to small changes in policy parameters**: small perturbations in model weights induce *butterfly effects* in the reward $J_H(\pi_{\widehat{\boldsymbol{\theta}}})$. In contrast, in the same regions, the *BC loss* landscape $\boldsymbol{\theta} \mapsto \ell_{\mathrm{BC}}(\pi_{\boldsymbol{\theta}})$ is well-behaved (nearly linear locally).

## 3.2 INSTABILITY IS CAUSED BY GRADIENT VARIANCE AMPLIFICATION

We now present compelling evidence that *variance in stochastic gradients* during training is responsible for training instability, because **gradient variance is amplified through the sensitivity of the rollout rewards to fluctuations in network parameters**. In Figure 2, we visualize evolution of both $\ell_{\mathrm{BC}}(\pi_{\widehat{\boldsymbol{\theta}}})$ and $J_H(\pi_{\widehat{\boldsymbol{\theta}}})$ over training for a variety of potential algorithmic interventions. We find that neither changing the model architecture and scale (1st row) nor standard regularization techniques (2nd row) ameliorate the training instabilities observed. We do see, however, that aggressively decaying the learning rate and increasing the batch size (3rd row) significantly reduces oscillations (at least when measuring mean rewards), at the expense of substantially slowing down training. Thus, we conclude that fluctuations from stochasticity in the gradients are to blame for oscillations in rollout rewards, and term this phenomenon *gradient variance amplification* (GVA). To summarize:

(R2) **GVA arises from algorithmic suboptimality rather than an information-theoretic limit.** Even with "infinite" training data (i.e., fresh trajectories with i.i.d. initial conditions at each training step), rollout oscillations persist.

(R3) **Training oscillations are *not* mitigated by many standard approaches to regularization**, including architectural interventions and increased regularization. On the other hand, **oscillations are ameliorated by variance reduction techniques**, such as large batch sizes, learning rate decay, and iterate averaging.

Appendix C shows that (R2) and (R3) remain true across environments and model architectures. In addition, we find that training instability is not the result of inadequate network architecture; we observe oscillations across model scales, and for both MLP and Transformer architectures.

While Figure 2 shows that it is possible to quell GVA using small learning rates or large batch sizes, this may not always be practical, as both interventions can incur steep computational costs.[3] Even worse, the success of continuous optimization in deep learning depends on non-convex feature learning mechanisms (Chizat et al., 2019), and too small a learning rate or too large a batch size can have deleterious effects on generalization.[4] Thus, it is vital to seek interventions that are holistically compatible with existing deep learning pipelines. Among these, Figure 2 highlights that

---

[3]As another unsatisfactory compromise, we also find that shallower models are less susceptible to GVA.

[4]We refer to some theoretical and empirical accounts in Appendix B.

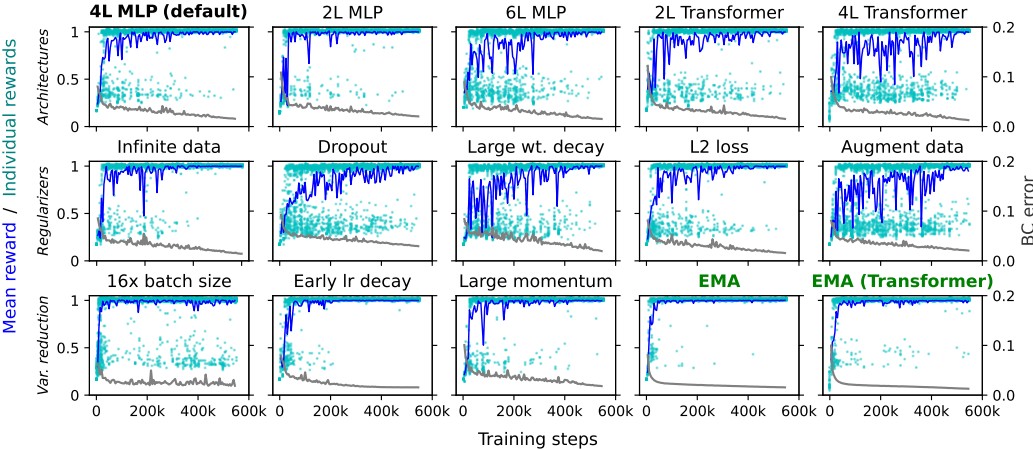

Figure 2: Highlights from a large suite of experiments, suggesting an algorithmic (rather than statistical) origin of reward oscillations. All plots use the 4-layer MLP architecture unless otherwise specified. Blue curves show mean rewards over 20 initial conditions, while teal dots show disaggregated per-episode rewards (such that each point represents the rollout reward of a fixed initial condition of the policy at the current iterate). These oscillations persist across dataset sizes, architectures, model scales, and choices of regularizers, and diminish toward the end of training as the learning rate decays to 0. They are most strongly mitigated by **variance reduction strategies**. Here, we opt for direct visualizations, providing a qualitative demonstration of GVA and its mitigations. We accompany these with quantitative comparisons in Appendix C.1.2.

a large momentum coefficient is mildly helpful, but taking an exponential moving average (EMA) of iterates (Polyak & Juditsky, 1992) is *extremely* effective. This motivates us to take a closer look at the latter in Section 4 through another suite of experiments.

## 3.3 UNDERSTANDING GVA: MISMATCH BETWEEN BC LOSS AND ROLLOUT REWARD

The disparity between behavior cloning loss $\ell_{\mathrm{BC}}(\pi_{\widehat{\boldsymbol{\theta}}})$ and rollout reward $J_H(\pi_{\widehat{\boldsymbol{\theta}}})$ has long been appreciated in the imitation learning literature, and is understood to be caused by *error amplification*, the process by which mildly erroneous predictions, when fed repeatedly through feedback loops, result in highly suboptimal performance (Chen & Hazan, 2021; Wang et al., 2020a). More precisely, for given $\ell_{\mathrm{BC}}$ and $J_H$ as well as a policy $\pi_{\boldsymbol{\theta}^\star}$ and $\delta > 0$, we define the *error amplification constant* at scale $\delta$ to be the maximal value (with respect to $\boldsymbol{\theta}$) for $J_H(\pi_{\boldsymbol{\theta}^\star}) - J_H(\pi_{\boldsymbol{\theta}})$ such that $\ell_{\mathrm{BC}}(\pi_{\boldsymbol{\theta}}) - \ell_{\mathrm{BC}}(\pi_{\boldsymbol{\theta}^\star}) < \delta$. The following proposition provides a simple theoretical illustration for how small fluctuations in BC loss can be drastically amplified by feedback between imperfectly-imitated policies and system dynamics.

**Proposition 3.1** (Example of exponential error amplification). *Let $\mathcal{B}_\delta$ denote the set of $\delta$-Lipschitz functions $\Delta : \mathcal{S} \to \mathcal{A}$ with $\Delta(\boldsymbol{0}) = \boldsymbol{0}$. For any $\delta > 0$, there exists a deterministic MDP with horizon $H$ and an expert policy $\pi_{\boldsymbol{\theta}^\star}$ such that the dynamics are Lipschitz in both state and action and $\pi_{\boldsymbol{\theta}^\star}$ is Lipschitz in the state, and such that*

$$\sup_{\Delta \in \mathcal{B}_\delta} \left\{ J_H(\pi_{\boldsymbol{\theta}^\star}) - J_H(\pi_{\boldsymbol{\theta}^\star} + \Delta) \right\} \geq \Omega(H) \cdot \left( e^{\Omega(H\delta)} - 1 \right),$$

*yet $\sup_{\Delta \in \mathcal{B}_\delta} \ell_{\mathrm{BC}}(\pi_{\boldsymbol{\theta}^\star} + \Delta) \leq \mathcal{O}\left( H \cdot \delta^2 \right)$, where $\ell_{\mathrm{BC}}$ is the $\ell_2$ loss. Thus, the error amplification constant is* exponential *in the time horizon.*

**Working model for GVA.** Proposition 3.1 shows that even when $\ell_{\mathrm{BC}}$ is uniformly small in a neighborhood around $\pi_{\boldsymbol{\theta}^\star}$, the rollout loss can be *exponentially large* in the same neighborhood. At the same time, there are good subsets of parameter space that do not experience this worst-case error amplification in our construction. We therefore hypothesize that, when stochastic optimization converges to a small neighborhood around zero-BC error models, it bounces between low-BC error

**Walker2d-v4: stabilizing effects of EMA (iterate averaging)**

Figure 3: Iterate averaging significantly mitigates GVA-induced reward oscillations, **without needing to change the learning rate schedule or batch size**. These improvements hold across architectures, dataset sizes, and *some* tasks. *Column 2, bottom:* Algorithmic instabilities are more pronounced at smaller sample sizes; thus, **stabilization can lead to improved sample efficiency**. *Column 3:* We recommend updating the EMA at every iterate, with an initial burn-in phase, and with a tuned $\gamma^{(t)} = t^{-\alpha}$ decay, to avoid divergence or slower progress. *Columns 4-5:* We verify that the benefits of EMA are not exclusive to the `Walker2d-v4` task; for some other tasks (including the higher-dimensional `Humanoid-v4`), oscillations are more benign.

models that experience large error amplification, and those that do not. To recapitulate: *GVA is the phenomenon in which gradient stochasticity leads to optimization trajectories repeatedly visiting regions of parameter space with catastrophic error amplification.* Because our MuJoCo environments involve nonlinear contact dynamics (while the example in Proposition 3.1 is linear), oscillations in Figure 1 are even more chaotic than this example may suggest. We elaborate on this point further by studying the advantages of EMA on a discontinuous "cliff loss" problem in Section 4.3.

## 4 MITIGATING GVA: STABILIZERS FOR UNSTABLE OPTIMIZERS

In Section 3.2, we isolated GVA as the primary cause of observed instabilities in BC (cf. Fig. 1) and identified iterate averaging with EMA (Polyak & Juditsky, 1992) as a promising remedy. In this section, we conduct an in-depth investigation of EMA as a mitigation. We start in continuous control (Section 4.1), and find EMA works almost unreasonably well at reducing GVA in the experimental testbed described in the prequel. Next, moving beyond continuous control (Section 4.2), we observe analogous effects in autoregressive language generation. In both settings, we find iterate averaging works so well as to **eliminate the need for full learning rate decay**; this leads us to recommend a conceptual reframing of EMA as a *stabilizer* for training neural networks, akin to (and interacting with) conventional optimizers and schedulers. We conclude (Section 4.3) by exploring the extent to which intuition on benefits of iterate averaging from the theory of stochastic convex optimization applies in our empirical settings.

### 4.1 THE OUTSIZED BENEFIT OF ITERATE AVERAGING

We recall the definition of the EMA method for iterate averaging (Polyak & Juditsky, 1992). Given an optimization trajectory $(\boldsymbol{\theta}^{(t)})_{0 \leq t} \subset \mathbb{R}^d$ and a sequence $(\gamma_t)_{1 \leq t} \subset [0, 1]$, the EMA iterates $(\widetilde{\boldsymbol{\theta}}_\gamma^{(t)})$ are[5]

$$\widetilde{\boldsymbol{\theta}}_\gamma^{(0)} = \boldsymbol{\theta}^{(0)}, \quad \text{and} \quad \widetilde{\boldsymbol{\theta}}_\gamma^{(t+1)} = (1 - \gamma_t) \cdot \widetilde{\boldsymbol{\theta}}_\gamma^{(t)} + \gamma_t \cdot \boldsymbol{\theta}^{(t+1)}. \tag{4.1}$$

Many prior works have detailed the benefits of iterate averaging in stochastic convex optimization and beyond (see Appendix B). Here, we investigate its effect on GVA. We begin by considering the

---

[5]Common heuristics include updating the EMA only after an initial "burn-in", and annealing $\gamma$ with a polynomial decay: $\gamma^{(t)} = \max(t^{-\alpha}, \gamma_{\min})$. It is also customary to use $\beta^{(t)}$ to denote $1 - \gamma^{(t)}$.

same MuJoCo framework as in Section 3. In Figure 3, we produce similar plots to those in Section 3, but this time juxtapose the vanilla trained models with an EMA of their iterates (further results and details are deferred to Appendix C). We observe the following:

(R4) **EMA iterate averaging strongly mitigates rollout oscillations**. *In every setup we consider, across a variety of architectures and environments, EMA significantly reduces the oscillations in rollout reward; in no instance does it hurt performance.*

We provide quantitative comparisons for a wide range of interventions in Figures 8 to 11.

## 4.2 AUTOREGRESSIVE SEQUENCE MODELS AND LANGUAGE GENERATION

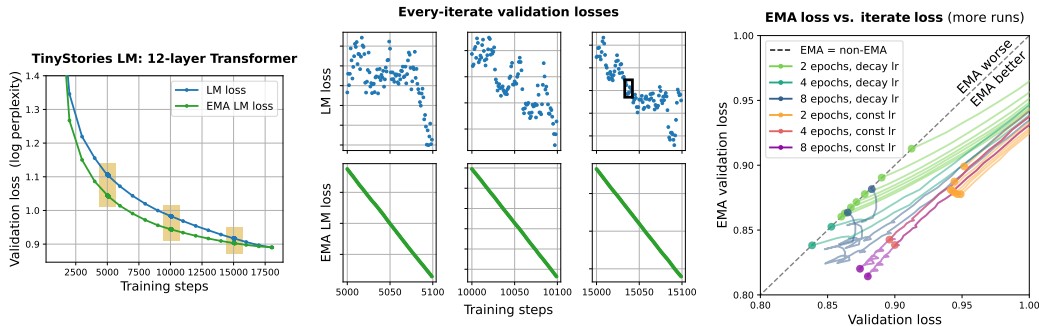

[...] *One day, Tim and Spot were playing near the pit when they saw a big ball. They both wanted to play with...*

**15027-15035:** ...the ball, but they didn't want to share. Tim said, "I want the ball!" Spot barked, "No, I want the ball!" They started to fight over the ball. Tim's mom saw them fighting and said, "Stop fighting! You can both play with the ball if you share." Tim and Spot stopped fighting and [...]

**15036-15037:** ...it. Tim said, "Let's share the ball, Spot!" Spot barked happily, and they both played with the ball together. As they played, a little girl named Sue came by. She saw Tim and Spot sharing the ball and wanted to join them. Tim said, "Yes, you can play with us!" So, Tim, [...]

**15038-15039:** ...it. Tim said, "I want the ball!" Spot barked and wagged his tail. They both ran to the ball and started to play. As they played, the ball went into the pit. Tim and Spot looked at each other, not knowing what to do. Then, a friendly frog jumped out of the pit with the ball [...]

**15040-15049:** ...the ball, but they didn't want to share. Tim said, "I want the ball!" Spot barked, "No, I want the ball!" They started to fight over the ball. As they fought, the ball rolled into the pit. Tim and Spot stopped fighting and looked at the pit. They were sad because they couldn't [...]

Figure 4: **GVA in natural language generation**, with 270M-parameter Transformer models trained on TinyStories. *(Top row) Left:* Validation loss curves with and without EMA. *Center:* Zooming in on *(left)*, evaluations at every update demonstrate small per-iterate loss fluctuations, which are even smaller if EMA is applied; note that the green "lines" are also scatter plots. *Right:* Training paths in (model loss, EMA loss) space. EMA enables training without learning rate decay; this mitigates overfitting, resulting in the lowest-perplexity model. *(Bottom)* Examples of autoregressively generated text (with argmax decoding), where nearby training iterates can bifurcate. See Appendix C.2 for full results, including quantitative evaluations of these *"butterfly effects"* in generation.

We posit that GVA is a generic phenomenon that can manifest in disparate settings: whenever a model's predictions are applied within a (marginally stable or unstable) feedback loop, the closed-loop dynamics can amplify small fluctuations in a deleterious manner. A natural and timely setting with this structure—which complements continuous control–is autoregressive language modeling. Here, a network's parameters $\boldsymbol{\theta}$ are optimized on a 1-step prediction loss, which takes the role of $\ell_{\mathrm{BC}}(\pi_{\boldsymbol{\theta}})$. The network $\pi_{\boldsymbol{\theta}}$ is then used to generate a sequence of symbols $w_{1:H}$ by iteratively rolling out $\pi_{\boldsymbol{\theta}} : w_{1:h} \mapsto w_{h+1}$. Such models have been paradigm-shattering in NLP, code synthesis, and beyond. Motivated by the similarity of this pipeline to behavior cloning,[6] we perform a smaller set of analogous experiments on language generation. Our findings here parallel our findings for continuous control, and show (i) the presence of GVA, and (ii) substantial benefits of iterate averaging. In more detail, we train 270M-parameter 12-layer Transformer models on the TinyStories dataset (Eldan & Li, 2023), which serves as an inexpensive surrogate for a full-scale pretraining pipeline. Highlights are shown in Fig. 4, while Appendix C.2 provides full documentation, including larger-scale training runs with a non-synthetic corpus (Wikipedia). We summarize our findings below:

---

[6]Many works have investigated GPT-style pretraining through the lens of offline IL (Chang et al., 2023a). There are many degrees of freedom in evaluating performance; thus, we do not commit to a canonical notion of reward and measure GVA-induced oscillations via disagreements in long-horizon rollouts.

(R5) Autoregressive LMs exhibit significant rollout oscillations throughout training. **EMA stabilizes the trajectory, accelerates training, and improves generalization**, complementing (and potentially obviating) standard practices in learning rate annealing.

## 4.3 To what extent does convex theory explain the benefits of EMA?

We close by providing mathematical intuition as to why iterate averaging with EMA can reduce the oscillations caused by GVA. As discussed in Section 3.3, oscillations can occur when there is a disparity between the BC loss $\ell_{\mathrm{BC}}(\pi_{\widehat{\theta}})$ on which we train and the rollout reward function $J(\pi_{\widehat{\theta}})$ on which we evaluate. To study this phenomenon, we a consider simple, horizon-one behavior cloning problem with a single action determined by the model parameter $\theta$. We take the *training loss* to be a quadratic $\ell_{\mathrm{BC}}(\theta) = \frac{1}{2} \cdot \|\theta - \mu\|^2$, and the *rollout reward* $J(\cdot)$ to be

$$J(\theta) = \begin{cases} -\|\theta - \mu\|^2, & \|\theta - \mu\| \leq \epsilon \\ -C, & \text{otherwise} \end{cases}, \tag{4.2}$$

where $C \gg \epsilon^2 > 0$ are constants. Here the training loss is convex, but rollout reward is not; the latter exhibits a "cliff," dropping sharply from $-\epsilon^2$ to $-C$ once $\|\theta - \mu\| > \epsilon$. The pair $(\ell_{\mathrm{BC}}, J)$ may be thought of as a discontinuous, horizon-one analogue of the example in Proposition 3.1, illustrating the contrast between extreme sensitivity of reward and insensitivity of the loss to the parameter of interest. The reward function encapsulates discontinuities arising in control tasks from, e.g., contract forces. In the MuJoCo walker, "cliff"-type behavior may come from an expert policy close to overbalancing the agent, with the learner's policy falling down if the parameter is "over the cliff."

We analyze SGD iterates $\theta^{(t+1)} = \theta^{(t+1)} - \eta(\theta^{(t)} - \mu + \mathbf{w})$, where $\eta > 0$ is a constant step size and $\mathbf{w} \sim \mathcal{N}(0, \mathbf{I})$. This corresponds to SGD on a noisy version of the BC loss given by $\tilde{\ell}_{\mathrm{BC}}(\theta) := \mathbb{E}[\|\theta_t - \mathbf{u} + \mathbf{w}\|^2]$, which satisfies $\mathbb{E}_{\mathbf{w}}[\tilde{\ell}_{\mathrm{BC}}(\theta)] = \ell_{\mathrm{BC}}(\theta) + \text{constant}$. We show that applying EMA to the resulting iterates achieves substantially higher rollout reward than vanilla SGD.

**Proposition 4.1** (Informal version of Proposition D.6). *Consider the setting in Eq. (4.2) for parameters $C \gg \epsilon^2 > 0$ in dimension one, and let $\theta^{(T)}$ denote the SGD iterate with learning rate $\eta > 0$ as described above. Let $\tilde{\theta}_{\gamma}^{(T)}$ denote the EMA iterate (4.1) with fixed parameter $\gamma_t \equiv \gamma \leq \eta$ satisfying $\gamma \gg 1/T$. Then, $\mathbb{E}[\ell_{\mathrm{BC}}(\theta^{(T)})]$ scales as $\Theta(\eta)$, while $\mathbb{E}[\ell_{\mathrm{BC}}(\tilde{\theta}_{\gamma}^{(T)})]$ scales as $\Theta(\gamma) \leq \eta$. In particular, when $\eta > c_1 \epsilon$, and $\gamma \log(C/\gamma) \leq c_2 \epsilon$, for absolute constants $c_1, c_2 > 0$, we find that*

$$\mathbb{E}[J(\theta^{\star}) - J(\theta^{(T)})] \geq \frac{C}{2}, \quad \text{but} \quad \mathbb{E}[J(\theta^{\star}) - J(\tilde{\theta}_{\gamma}^{(T)})] \leq \mathcal{O}(\gamma).$$

This proposition holds, which shows that the rollout performance for EMA can be arbitrarily small relative to that of SGD, holds even in the regime where SGD is initialized at $\theta^{(0)} = \mu$ (so that both $\theta^{(T)}$ and $\theta_{\gamma}^{(T)}$ are unbiased estimates of $\mu$), and thus highlights that EMA can reduce the variance that arises from accumulation of SGD noise.[7] Notice that Proposition 4.1 requires $\eta \geq \gamma \gg 1/T$, which is above the optimal step size of $\eta_t = 1/t$.[8] Indeed, in Appendix D we show that EMA, with the parameters we find empirically successful, only benefits optimization *above* these aggressively-decayed theoretically optimal learning rate schedules. Thus, we conclude that **convex theory reveals the variance-reducing benefit of *either* learning rate decay or EMA, but does not suggest which one is better.** We defer further theoretical results to Appendix D, and present an empirical study of a system motivated by the cliff loss in Appendix C.5; in particular, our analysis provides a simple example where GVA provably occurs, both theoretically and empirically.

The above example reveals the difference between the **statistical and algorithmic difficulties** of BC: with enough data, the empirical risk minimizer (sample mean) $\widehat{\theta}$ of BC loss exhibits $\ell_{\mathrm{BC}}(\theta) \sim 1/T \ll \epsilon$, which ensures $J_H$ is small; on the other hand, with minibatch SGD and too large a learning rate, there is a noise floor on how close the non-EMA'd iterate $\widehat{\theta}$ will be to $\theta^{\star}$, ensuring that $J_H$ is large.

---

[7]We compare to similar findings (Sandler et al., 2023) in Appendix B.
[8]Note that the $\eta_t = \frac{1}{t}$ step size schedule gives the sample mean, which is the maximum likelihood estimator for our objective.

ACKNOWLEDGMENTS

We are extremely grateful to Jonathan Chang for illuminating discussions at multiple stages of this project. We thank Daniel Pfrommer for independent verifications of GVA: that it is benign for the 2D quadcopter (a canonical smooth nonlinear control environment), and less so when training diffusion models to imitate expert behavior on the PushT environment. We also thank Mojan Javaheripi, Piero Kauffmann, and Yin Tat Lee for helpful discussions about learning rate schedules, and Sadhika Malladi for helpful NLP references. AB greatfully acknowledges support from the National Science Foundation Graduate Research Fellowship under Grant No. 1122374.

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

Contents of Appendix

## A  Discussion and conclusion

We investigated the causes of instabilities in training neural networks in tasks involving feedback loops, and empirically demonstrated that these instabilities often arise as a result of poor algorithmic choices rather than information-theoretic barriers. We hypothesize that the oscillations are caused by the combination of fractal reward landscapes and noise floor the stochastic gradients, and can be ameliorated by variance reduction—either during training, or *post hoc* (via a stabilizer such as EMA). We now list some directions for future inquiry, containing theoretical and empirical mysteries.

**Unstable optimizers & trajectory stabilization.**  Despite many attempts to design faster-converging optimizers for deep learning, there has been no clear successor to vanilla second-moment preconditioning (Duchi et al., 2011; Kingma & Ba, 2014) with a well-tuned learning rate schedule (Agarwal et al., 2020; Schmidt et al., 2021; Kaddour et al., 2023). By decoupling the roles of optimizers and stabilizers, we open a new avenue for innovation in this space. When training deep neural networks in the presence of feedback loops, our findings suggest: **always try EMA whenever it is affordable**, and **train at higher learning rates without decaying to 0**. Investigations of nuanced downsides, larger-scale settings, and new optimizer-stabilizer algorithms are left for future work.

**Representation learning in RL.** While this work focused on fully offline data, online RL is also known to suffer from serious training instability, often thought to be caused by the statistical challenges of interactive learning. On the other hand, GVA appears to be less pronounced in this setting,[9] and the interaction between GVA and training with "better" data remains mysterious.

On a related note, the inductive biases of iterate averaging for representation learning remain poorly understood. Many algorithms for representation learning benefit from EMA (Grill et al., 2020; Yaz et al., 2018); to what extent can this be explained by a phenomenon related to GVA?

Finally, we observe that our results are consistent with the notion that SGD noise is only harmful, but this cannot be entirely true due to the fact that sufficiently large batch sizes hurt training; we leave a better understanding of when and why SGD noise is helpful for future work.

## B    ADDITIONAL RELATED WORK

In this section, we provide an extended discussion of related work. We split our discussion into two sections, beginning with our primary focus on stochastic optimization (especially as applied to deep learning) and ending with works relevant to one of our empirical domains, reinforcement learning and control.

### B.1    STOCHASTIC OPTIMIZATION AND DEEP LEARNING

**Stochastic optimization in deep learning.**    Minibatch SGD (Robbins & Monro, 1951) is the workhorse of deep learning, and its nuances are explained by mechanisms far outside the purview of stochastic convex optimization. Batch sizes and learning rates are known to control variance (Smith et al., 2017); however, not all variance reduction techniques work in practice (Defazio & Bottou, 2019). Nonetheless, analyses of SGD in simplified settings have proven to be useful (Mandt et al., 2017; Malladi et al., 2022; Li et al., 2019a; 2020). The theoretical literature contains numerous accounts of why finite-sample gradients might be beneficial, e.g., posterior inference (Mandt et al., 2017) and saddle point escape (Jin et al., 2017). On the other hand, we show that this stochasticity can lead to catastrophic error amplification in regimes of practical interest.

**Iterate averaging.**    It is known that returning the last iterate of even deterministic gradient descent is suboptimal, even when the function being optimized is convex (Harvey et al., 2019) (but is non-smooth). Which averaging scheme is optimal depends on whether the underlying function is strongly convex or not, and we discuss the various tradeoffs in Appendix D.3.1. Importantly, there exists memory-efficient averaging schemes (storing at most 2 optimization iterates) which attain optimal rates (Lacoste-Julien et al., 2012). The optimality of EMA for stochastic optimization has not been studied in the literature, and indeed, we show in Appendix D.3.1 EMA is suboptimal – even for deterministic optimization via gradient descent – whenever the averaging parameter $\gamma$ satisfies $\gamma \sim T^{-\beta}$ with $\beta < 1$. This theoretically suboptimal regime is precisely the one in which we observe strong mitigation of GVA in our experiments. In fact, the case of $\beta = 1$, which is analyzed in Lacoste-Julien et al. (2012) in the convex case, is *empirically suboptimal* and leads to a drastic reduction in performance early in training, which is then washed out in later iterations. The stabilizing behavior of iterate averaging of machine learning in practice is well-known (e.g., Bottou (2010)), although it has received less emphasis in modern deep learning than other optimizer hyperparameters.

In addition to theory, iterate averaging in general, and EMA in particular, is extremely common in practice (Izmailov et al., 2018; Kaddour, 2022; Kaddour et al., 2023; Izmailov et al., 2018; Busbridge et al., 2023; He et al., 2019). In contradistinction to these works, the present paper is not purporting to examine iterate averaging in and of itself, but rather is exploring the effects of iterate averaging in the presence of GVA.

**Non-convex side effects of variance reduction techniques.**    Here, we expand on the discussion from Section 3.1, concerning why decreasing the learning rate and increasing the batch size are not satisfactory interventions for GVA:

---

[9]In a small anecdotal exploration, we found that even minimal online interventions to improve data coverage such as DART (Laskey et al., 2017) appear to reduce oscillations significantly.

- *Learning rate:* It is understood in practice that large learning rates promote faster and more robust representation learning in neural networks; theoretical understanding is an active frontier of research, which requires analyzing the trajectory of SGD in the *rich* (data-adapted) regime: see (Li & Yuan, 2017; Li et al., 2019b; Chizat et al., 2019; Barak et al., 2022; Damian et al., 2022; Telgarsky, 2022; Andriushchenko et al., 2023) and references therein.[10]

- *Batch size:* It is also well-understood in practice that the mechanisms enabling successful neural network training break down at extremely large batch sizes. This is especially relevant in the quest for massively parallel training pipelines (Goyal et al., 2017). One way to analyze this theoretically is to note that smaller-batch SGD can more effectively memorize individual examples (Abbe & Sandon, 2020; Abbe et al., 2021), often a desired behavior.

Of greatest relevance to our investigation of iterate averaging, Sandler et al. (2023) presents some related empirical and theoretical results in the context of supervised learning and explores an equivalence between learning rate decay and EMA. Their empirical results focus on pure supervised learning and examine the relationships between training trajectories with different learning rates and how iterate averaging affects this, while we primarily focus on the phenomenon of GVA and the effect of iterate averaging therein.

**SDE limits and analysis of training in deep learning.** Taking the continuous time limit of adaptive gradient optimization procedures and modeling the result as the solution to a stochastic differential equation has seen a lot of recent interest; see Mandt et al. (2017); Cheng et al. (2020); Li et al. (2020); Kunin et al. (2020); Li et al. (2019a); Malladi et al. (2022) and the references therein. Of greatest relevance to our paper is Busbridge et al. (2023), which proves the convergence of constant rate SGD along with constant rate EMA to a system consisting of an SDE and an ODE. Proving rigorous convergence guarantees in this limit is beyond the scope of this work; instead, for one of the results in Appendix D, we take the continuous time limit as given and precisely compute in several toy settings the benefits that EMA has.

**Error amplification and closed-loop effects in language generation.** The deep RL and deep NLP literatures are highly connected, and use common vocabulary to describe considerations involving feedback loops (e.g. training on the BC/autoregressive loss is known as *teacher forcing* (Arora et al., 2022)). Long-range error amplification stemming from autoregression has been noted in prior work (Holtzman et al., 2019; Braverman et al., 2020) (sometimes called *exposure bias*), motivating the design of decoding heuristics.

## B.2 STABILITY IN REINFORCEMENT LEARNING AND CONTROL

**Instabilities of online RL.** Many works have empirically observed instabilities in online RL, often arising as a result of distribution shift (Kirkpatrick et al., 2017; Henderson et al., 2018; Machado et al., 2018; Packer et al., 2018; Zhang et al., 2018; Engstrom et al., 2019; Andrychowicz et al., 2020). On the theoretical side, most existing work on online RL explores structural assumptions (e.g., Bellman rank) that allow carefully designed algorithms to control distribution shift (Russo & Van Roy, 2013; Jiang et al., 2017; Sun et al., 2019; Wang et al., 2020b; Du et al., 2021; Jin et al., 2021; Foster et al., 2021a), but does not directly address the behavior cloning setting we consider, in which instability arises despite the absence of distribution shift.

**Instabilities of offline RL.** While many works have empirically observed instabilities in online RL (Kirkpatrick et al., 2017; Henderson et al., 2018; Machado et al., 2018; Packer et al., 2018; Zhang et al., 2018; Engstrom et al., 2019; Andrychowicz et al., 2020), it is generally understood that these instabilities arise as a result of the distribution shift caused by the complicated dependence structure of interactive data. In contrast, we observe instabilities in the offline setting, where the data is fixed and the dependence structure is significantly simpler. In offline RL, too, many works have observed instabilities (Fujimoto et al., 2019; Kumar et al., 2019), although several theoretical works suggest that the source of these instabilities may be information theoretic limits as opposed to algorithmic issues (Wang et al., 2020a; Zanette, 2021; Foster et al., 2021b).

---

[10]There is also a regime in which *small* learning rates promote generalization (Bousquet & Elisseeff, 2002; Hardt et al., 2016), but this prevails predominantly in online and convex problems.

In particular, Wang et al. (2021) find that conducting offline RL using pre-trained representations often suffers from catastrophic error amplification. The authors there apply known planning algorithms on top of pre-trained representations and find that even in the presence of mild distribution shift, severe degradation in rollout reward occurs. Dong et al. (2020) observe fractality in continuous MDPs as a generic phenomenon; like Wang et al. (2021), they are primarily interested in *learning and planning* as opposed to imitation.

**(In-)stability in behavior cloning.** Behavior cloning is known to suffer from compounding error, a phenomemon in which the imitators's errors accumulate over time and place the learner in states not observed during training. As we confirm in this paper, these instabilities are readily mitigated by forcing the expert to demonstrate corrections from imperfect demonstrations, either by forcing experts to correct from noisy states (e.g. DART (Laskey et al., 2017)), or by sequential iteraction with the environment (e.g. DAGGER (Ross & Bagnell, 2010)); both interventions require online interaction with the expert and thus lies outside the scope of this paper. Direct data augmentation of the expert trajectories via noising is popular in practice (Ke et al., 2021), but does not mitigate training oscillations.

More recent theoretical work (Tu et al., 2022; Pfrommer et al., 2022; Havens & Hu, 2021) has connected behavior cloning to the theory of incremental stability of the induced closed loop dynamics (Angeli, 2002). Leveraging this stability often requires stronger oracles, such as the ability to compute first- or zero- order gradients of the expert policy (Pfrommer et al., 2022). Recently, Block et al. (2023a) proposed a combination of probabilistic and control-theoretic stability to facilitate imitation of *non-smooth* expert trajectories in smooth dynamical systems, assuming access to a type of local stabilization oracle.

**Comparison to Emmons et al. (2021); Chang et al. (2023b).** These works investigate the fact that $\ell_{\mathrm{BC}}$ on a validation set is a poor proxy for $J_H$. In Emmons et al. (2021), through a suite of experiments in a number of domains motivated by robotics, the authors probe how far applying only supervised learning can get in producing a high reward policy. They focus primarily on architectural and regularization interventions including depth and width of their imitator policies as well as the effects of weight decay and dropout on the resulting policies, noting the weak relationship between reward and $\ell_{\mathrm{BC}}$ *at the end of training* in all of these examples. In contrast to this work, we focus on causes and mitigations of *instabilities during training* as well as the more general phenomenon applied beyond continuous control tasks.

Chang et al. (2023b) finds empirically that $\ell_{\mathrm{BC}}$ evaluated on a held out validation set is an extremely poor empirical proxy for rollout rewards in many common benchmarks *during a single training run*, but were not primarily interested in the instability itself, and thus did not further investigate this phenomenon.

**The role of stability in learning to control.** Considerations of stability (in a dynamical sense, rather than a training sense) have been most visible in the recent literature on learning to control linear systems. While stability is not always essential to estimation in control systems (Simchowitz et al., 2018; Sarkar & Rakhlin, 2019; Ghai et al., 2020), much of the contemporary research on learning to control linear systems has assumed access to an initial stabilizing control policy (Fazel et al., 2018; Simchowitz & Foster, 2020; Dean et al., 2018). Indeed, without access to such a controller, certain adaptive control settings require regret scaling exponentially in problem dimension (Chen & Hazan, 2021), until the system can be estimated to sufficient accuracy that a stabilizing controller can be synthesized (in some circumstance, however, these effects can be circumvented if one can construct simulators with reward discounting (Perdomo et al., 2021)). Access to an initial stabilizing controller forms the basis for both classical (Youla et al., 1976) and more recent (Agarwal et al., 2019; Simchowitz et al., 2020; Simchowitz, 2020) convex control parameterizations.

Stability appears more implicitly in the recent literature on nonlinear control. For example, assuming that the output of the dynamical map is bounded and that the system exhibits process noise amounts to a form of marginal stability in Wasserstein and total variation distances (see e.g. Kakade et al. (2020)). Lastly, stabilizability of linearizations of nonsmooth dynamics have emerged as an important desideratum for optimization-based and data-driven trajectory optimization in smooth nonlinear systems (Pfrommer et al., 2023; Westenbroek et al., 2021).

**Stiffness in contact and hybrid dynamics.** System stability is at least as bad as system "stiffness", or largest Lipschitz constant dynamics, which can be arbitrarily large in contact and hybrid dynamical systems even if one does not observe the sort of exponential blow up witnessed in unstable linear dynamics. At the limit, dynamical discontinuities induce "one-step" instability. Contending with systems stiffness has been a long-standing challenge in the field of robotics, where contact systems are modeled non-smoothly (Van Der Schaft & Schumacher, 2007). Recent work has explored the use of stochastic smoothing to circumvented these challenges in trajectory optimization of a known stiff system (Parmas et al., 2018; Suh et al., 2022). Further work has shown that stochastic smoothing makes sequential prediction and planning problems tractable for piecewise affine systems (Block & Simchowitz, 2022; Block et al., 2023b;c). Whether the dynamics of the noiseless system are well-approximated by those of the same system with process noise added remains an open question.

## C  FULL EXPERIMENTAL RESULTS

In this section, we provide additional experimental results as well as full details for the experiments included in the main text. The appendix is organized as follows:

- In Appendix C.1 by providing a detailed description of the experimental setup we use in our empirical MuJoCo results.
- In Appendix C.1.1, we provide an extended discussion of visualizations related to Figure 1. We continue in Appendix C.1.2 by detailing our large suite of experiments investigating the effects that various algorithmic interventions have on GVA, both qualitatively and quantitatively. These correspond to the findings highlighted in Figure 2 and Figure 3 in the main paper.
- In Appendix C.1.3 we provide several additional visualizations, including a number of reward landscapes similar to Figure 1 *(right)* to ensure the robustness of our conclusion that the reward landscape is extremely sensitive to the parameter. We also demonstrate that the successful Transformer imitators learn *non-Markovian* policies; unlike the MLPs, the Transformers observe a history (of length 32 in our experiments) of many past states and actions; in Appendix C.1.3 we show that the Transformers are attending to previous states and not just the most recent one; this corroborates findings in Janner et al. (2021); Shafiullah et al. (2022).
- In Appendix C.2, we give additional details and results relating to our autoregressive language generation experiments, described in Section 4.2.
- In Appendix C.4, we present synthetic experiments on 2-dimensional linear dynamical systems, which serve as accompaniments to our theoretical vignettes. We show empirically that GVA does not occur in marginally stable linear systems, and show that a natural dynamical systems analogue of the cliff loss described in Section 4.3 indeed exhibits the predicted GVA in Appendix C.5; we also observe that, as predicted by the theory, EMA mitigates the oscillations.

All of our experiments are implemented in PyTorch (Paszke et al., 2019), and we use the `x-transformers`[11] implementation of the Transformer architecture. All of our MuJoCo tasks are in Gymnasium (Towers et al., 2023). For our LQR experiments, we implement the dynamics directly in PyTorch.

### C.1  MUJOCO CONTINUOUS CONTROL ENVIRONMENTS

In all of our MuJoCo experiments, we run the following pipeline:

1. Train an expert using online reinforcement learning.
2. Collect expert trajectories by rolling out the (deterministic) expert policy 1000 times.
3. Split the collected trajectories into a training set of size $N = 900$ and a validation set of size 100 trajectories.

---

[11] https://github.com/lucidrains/x-transformers

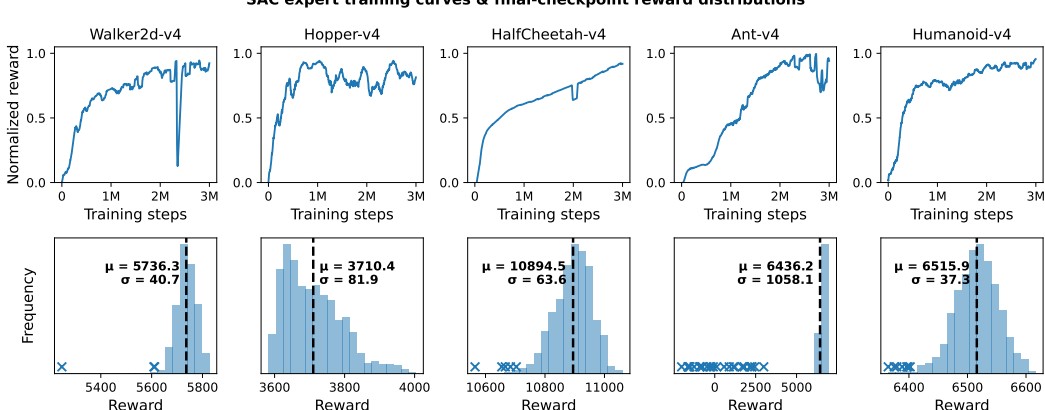

Figure 5: Training curves of the SAC experts for the various MuJoCo continuous control agents, along with final-iterate mean rewards. *Top:* Online reinforcement learning training curves; these exhibit training instabilities, but not of the same nature as those encountered in our offline behavior cloning settings. *Bottom:* Unnormalized reward distributions (through out the rest of this paper, we divide by these means). Outliers are marked by × symbols.

4. Train an agent to imitate the expert from these $N$ offline trajectories, using standard deep learning pipelines to optimize the behavior cloning loss $\ell_{\mathrm{BC}}$.

5. Evaluate the behavior cloner by rolling out the learned policy and evaluating loss $\ell_{\mathrm{BC}}$ on the validation set.

We now discuss some of the items above in more detail.

**Training the expert.** All of our experts are trained using the Soft Actor Critic (SAC) algorithm (Haarnoja et al., 2018) as implemented by `stable-baselines3` (Raffin et al., 2021) with the default parameters for the environment under consideration. We train our experts for 3 million steps; training curves and reward histograms (over the randomness of environment initial conditions) are shown in Figure 5.

**Collecting expert trajectories.** While the SAC online RL algorithm results in a stochastic policy, we are interested in deterministic experts. Thus, we derandomize the expert's outputs, via taking the mean of the action distribution of the randomized SAC agent. We roll out trajectories of length 1000 for every environment. Interestingly, the presence of extreme outliers (`Ant`) and high-dimensional states (`Humanoid`) appear to be uncorrelated with the magnitude of GVA (which is very low for both of these tasks).

**Training the behavior cloner.** This is the step that our work is interested in and is where all of our interventions occur. For every environment, unless specified otherwise, we train the cloner for 20 epochs. Our default hyperparameters are included in Table 1, although each intervention below changes at least one of them.

**Evaluating the behavior cloner.** For most training runs, we collect a checkpoint every 5000 gradient steps, with the exception of the run in Figure 1, where we collect checkpoints every gradient step between iterates 40K and 50K in order to have a more zoomed-in view of the oscillations. For each checkpoint, we compute $\ell_{\mathrm{BC}}$ on the validation set as well as fixing 20 random seeds and, for each seed, sampling a trajectory beginning from the seed-induced initial condition and rolling out the learned policy. Note that the only randomness here is in the initial condition of the agent, as determined by the seed.

**Choice of canonical environment.** Overall, we choose to focus our experiments on `Walker-v4`, for the following reasons: (1) reward oscillations are significantly more benign for cloners of `Ant`

and `Humanoid` agents; (2) we could not get a `HalfCheetah` behavior cloner to reach the maximum reward. We choose to focus our intensive hyperparameter sweeps on `Walker` (rather than `Walker` and `Hopper`) to keep the number of runs manageable.

| Name | Value |
|---|---|
| Optimizer | AdamW |
| Learning rate | $3 \times 10^{-4}$ |
| Epochs | 20 |
| Batch size | 32 |
| $\beta_1$ (momentum) | 0.9 |
| $\beta_2$ | 0.999 |
| Weight decay | 0.1 |
| *(MLP)* Number of layers | 4 |
| *(MLP)* Width | 1024 |
| *(MLP)* Activation | ReLU |
| *(Transformer)* Number of layers | 4 |
| *(Transformer)* Number of heads | 4 |
| *(Transformer)* Embedding dimension | 1024 |
| *(Transformer)* Positional encoding | sinusoidal |
| *(Transformer)* History length | 32 |

Table 1: Table of hyperparameters for behavior cloning.

### C.1.1 REWARD SENSITIVITY, PER-ITERATE OSCILLATIONS, AND LANDSCAPES

Recall the first finding from the main paper:

(R1) **The reward landscape is highly sensitive to small changes in policy parameters**.

Figure 6 provides a closer look at this landscape, via additional visualizations and discussion for the example of GVA from Figure 1:

(a) **Per-iterate fluctuations:** Zooming into the training trajectory, we see extremely large fluctuations at the level of individual gradient descent iterates (blue dots). Rewards can vary by $> 50\%$ of the total maximum reward. Meanwhile, the behavior cloning loss varies at the scale of $< 1\%$ of the range of $\ell_{\mathrm{BC}}$.

(b) **Weak correlation between $J_H$ and $\ell_{\mathrm{BC}}$:** For many pairs of iterates, the behavior cloning objective and rollout rewards are negatively correlated. Note that cloners achieve high rewards without converging to perfect action recovery (red star). Representation is clearly not the bottleneck: the expert agent is a 2-layer MLP with hidden dimension 128, which is realizable by all of these networks.

(c) **Exhaustive evaluation of cross-sections:** For two checkpoints $\boldsymbol{\theta}$ (iterations 115000 and 120000), selected to exemplify negative progress as training progresses, we evaluate neighborhoods of these policies in 2-dimensional cross-sections of the reward landscape. We compute stochastic gradients $\mathbf{g}_1, \mathbf{g}_2$ of $\ell_{\mathrm{BC}}$, for two randomly selected minibatches of size 32. For a choice of step size $\eta = 3 \times 10^{-4}$ (coinciding with the Adam optimizer's learning rate), we evaluate rewards and BC losses for the lattice of policies $\{\boldsymbol{\theta} + i \cdot \eta \mathbf{g}_1 + j \cdot \eta \mathbf{g}_2\}_{i,j \in \{-100,\dots,100\}}$. These rewards are averaged over 20 random seeds determining the initial conditions of the environment, which are *not* re-randomized between evaluations. Thus, we conclude that these fluctuations arise from extremely high sensitivity to initial conditions. Aside from providing a qualitative visualization that GVA is evident, this evaluation provides a lower bound of $2 \times 10^4$ on the Lipschitz constant of $\boldsymbol{\theta} \mapsto J_H(\pi_{\boldsymbol{\theta}})$ (choosing the $\ell_2$ norm for $\boldsymbol{\theta}$), even restricted to non-adversarial stochastic gradient directions. The loss landscape (depicted in Figure 1c, and omitted here) is nearly linear, and much smoother (Lipschitz constant $\approx 50$).

In addition, in Figure 7 we repeat Figure 6(a) but with each mean reward disaggregated such that each point represents the rollout reward of a fixed initial condition of the policy at the current iterate. The points are colored by initial condition so as to illustrate that it is not the case that some initial conditions are significantly more difficult than others.

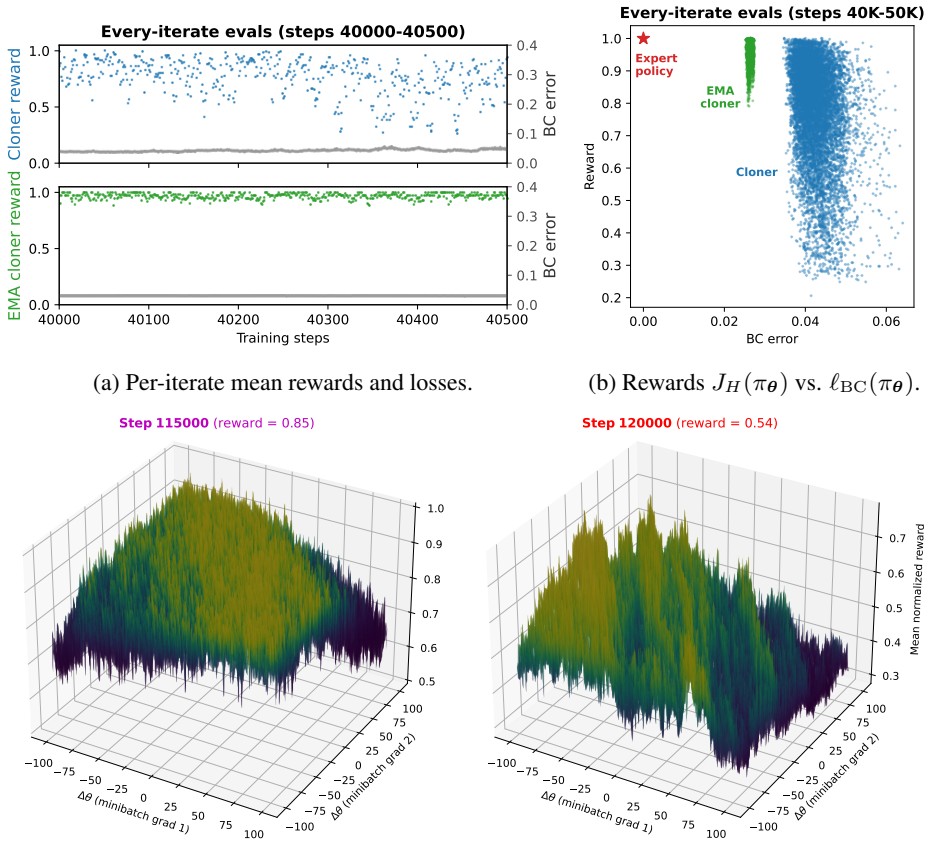

(a) Per-iterate mean rewards and losses.

(b) Rewards $J_H(\pi_{\boldsymbol{\theta}})$ vs. $\ell_{\mathrm{BC}}(\pi_{\boldsymbol{\theta}})$.

(c) 2-dimensional cross-sections $J_H(\pi_{\boldsymbol{\theta}} + i \cdot \eta \mathbf{g}_1 + j \cdot \eta \mathbf{g}_2)$ of the reward landscape.

Figure 6: Additional visualizations for the experiments highlighted in Figure 1. (a) Rewards $J_H(\pi_{\boldsymbol{\theta}})$ and $\ell_{\mathrm{BC}}(\pi_{\boldsymbol{\theta}})$ plotted at *every* training iteration in a small range. (b) Scatter plot of $J_H(\pi_{\boldsymbol{\theta}})$ against $\ell_{\mathrm{BC}}(\pi_{\boldsymbol{\theta}})$ for the full range of every-iterate evaluations displayed in Figure 1. (c) Reward landscapes are shown around good (*left*) and bad (*right*) iterates. Note the extremely high Lipschitz constant (local sensitivity) of the rollout reward in terms of policy parameters; furthermore, there exist good and bad policies in very close proximity along stochastic gradient directions.

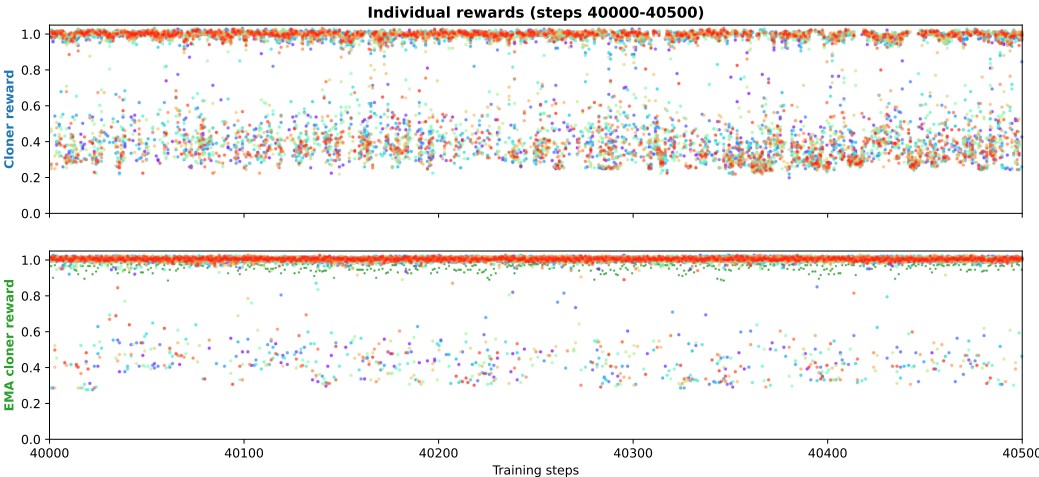

Figure 7: Disaggregated version of Figure 6(a). We visualize rollout rewards of *each rollout* for each iterate in the experiment in Figure 1 with points colored by environment random seed (which only determine initial conditions). We exhibit both the non-EMA (top) and the EMA (bottom) runs.

### C.1.2 Effects of algorithmic interventions

Below, we list the exhaustive range of algorithmic interventions probed for in the `Walker2d-v4` environment. We begin with the suite of experiments focused on diagnosing GVA (highlighted in Figure 2):

(1a) **Architectures:** For both {MLP, Transformer} architectures, we vary the model scale parameters: depths $\{2, 4, 6\}$ and embedding dimensions $\{128, 256, 512, 1024\}$. For all other experiments below, we test interventions on the 4-layer 1024-dimensional MLPs and Transformers.

(2) **Regularization controls:** We consider varied feedforward dropout coefficients ($\{0, 0.1, \ldots, 0.5\}$) for both MLPs and Transformers, and embedding dropout coefficients in the same range for Transformers. We also try weight decays $\{0.01, 0.3\}$, losses $\{\ell_1, \ell_2\}$, and data augmentation scales $\{0.005, 0.01, 0.02\}$.

(3) **Optimizer configurations:** We vary the momentum and adaptive preconditioner parameters ($\beta_1, \beta_2$) of AdamW, noting that these are known to modulate training stability and generalization. We vary $\beta_1 \in \{0.7, 0.9, 0.95, 0.99\}$ and $\beta_2 \in \{0.9, 0.99, 0.999, 0.9999\}$. $\beta_2 = 0.9999$ leads to many non-convergent runs, and are omitted from Figure 9.

(4) **Variance reduction controls:** We vary the power $\alpha$ in the learning rate schedule $\eta_t \propto (1 - \frac{t}{T})^\alpha$, for $\alpha \in \{0, 0.5, 1, 1.5, 2, 2.5, 3\}$; note that $\alpha = 0$ corresponds to removing the learning rate decay. We also try batch sizes of 128 and 512 via $2\times$, $4\times$ and $16\times$ gradient accumulation with the default batch size of 32.

Next, we list the training runs selected to investigate the scope and hyperparameter sensitivity of EMA (highlighted in Figure 3):

(1b) **Architectures:** For all of the architecture choices in (G1) above, we evaluate EMA cloners.

(5) **EMA parameter ablations:** To check the robustness and hyperparameter sensitivities of EMA, we perform a search in the hyperparameter space of update frequencies, burn-in durations, and annealing powers. Throughout these sweeps, we clip the annealing coefficient to a minimum of $\gamma_{\min} = 1 \times 10^{-4}$. Although this can be harmful, the optimal choice of this parameter depends on the number of training steps; we recommend setting a low $\gamma_{\min}$ and tuning the annealing power instead.

(6a,b) **Ablations of sample size and learning rate decay:** We downsample the 900 training trajectories, instead training agents on $\{10, 20, 50, 100, 200, 500\}$ demonstrations. In all of these settings, we remove learning rate decay, and find that this improves generalization (as long as EMA is applied). We evaluate both non-EMA and EMA models.

(7a,b) **Additional environments:** We perform a limited set of analogous comparisons between EMA and non-EMA cloners for other continuous control tasks provided by `gymnasium`. We evaluate both non-EMA and EMA models.

**Quantitative evaluations.** Stochastic optimization in deep learning is fundamentally non-stationary, in the sense that representations change over the course of the training trajectory. Thus, there is no canonical choice of summary statistic which losslessly captures the magnitude of GVA. Nonetheless, to provide a digestible comparison of algorithmic interventions and their effect on GVA, we record a number of basic quantitative metrics:

- **Maximum and final rollout rewards** $J_{\max}, J_{\mathrm{final}}$. These serve as indicators of whether the BC training procedure ever arrives at good policies, modulo stability considerations. GVA then entails a lack of stable convergence to these iterates, in terms of rollout performance. We similarly record the **minimum and final behavior cloning objectives** $\ell_{\min}, \ell_{\mathrm{final}}$.

- **Reward statistics** $\mu_{\mathrm{mid}}, \mathrm{range}_{\mathrm{mid}}$ **from iterates in the middle 50% of training.** We track the mean and range, to summarize the amplitude of GVA-induced reward oscillations through the course of training.

- **Indicators of early success and negative progress** $t_{\mathrm{early}}, t_{\mathrm{worse}}$. We record the **earliest** iteration (as a percentage of the total number of training steps) at which the model visits a checkpoint whose mean reward is $\geq 95\%$ of the best one in its training trajectory. We also

| | (1a) Architectures, no EMA | | | | | | | | (1b) + EMA | | | | | | |
| | $J_{max}$ | $J_{final}$ | $\ell_{min}$ | $\ell_{final}$ | $\mu_{mid}$ | $range_{mid}$ | $t_{early}$ | $t_{worse}$ | $J_{max}$ | $J_{final}$ | $\ell_{min}$ | $\ell_{final}$ | $\mu_{mid}$ | $range_{mid}$ | $t_{early}$ | $t_{worse}$ |
|---|---|---|---|---|---|---|---|---|---|---|---|---|---|---|---|---|
| 2L 128d MLP | 0.84 | 0.66 | 0.075 | 0.075 | 0.49 | 0.61 | 0.66 | - | 0.83 | 0.73 | 0.075 | 0.075 | 0.50 | 0.52 | 0.57 | - |
| 4L 128d MLP | 1.00 | 0.97 | 0.054 | 0.054 | 0.82 | 0.64 | 0.39 | 0.99 | 1.00 | 0.98 | 0.054 | 0.054 | 0.90 | 0.60 | 0.46 | 0.94 |
| 6L 128d MLP | 1.00 | 1.00 | 0.051 | 0.051 | 0.77 | 0.55 | 0.33 | 0.99 | 1.00 | 0.97 | 0.050 | 0.050 | 0.86 | 0.24 | 0.24 | 0.96 |
| 2L 256d MLP | 1.00 | 1.00 | 0.050 | 0.050 | 0.91 | 0.44 | 0.27 | 0.96 | 1.00 | 1.00 | 0.050 | 0.050 | 0.96 | 0.13 | 0.18 | 0.79 |
| 4L 256d MLP | 1.01 | 1.00 | 0.034 | 0.034 | 0.98 | 0.13 | 0.18 | 0.71 | 1.00 | 1.00 | 0.033 | 0.033 | 0.99 | 0.03 | 0.07 | 0.11 |
| 6L 256d MLP | 1.00 | 1.00 | 0.030 | 0.030 | 0.94 | 0.34 | 0.14 | 0.72 | 1.00 | 1.00 | 0.030 | 0.030 | 0.99 | 0.06 | 0.09 | 0.39 |
| 2L 512d MLP | 1.00 | 1.00 | 0.031 | 0.031 | 0.98 | 0.20 | 0.10 | 0.90 | 1.00 | 1.00 | 0.031 | 0.031 | 1.00 | 0.05 | 0.07 | 0.28 |
| 4L 512d MLP | 1.00 | 1.00 | 0.021 | 0.021 | 0.98 | 0.32 | 0.08 | 0.66 | 1.00 | 1.00 | 0.021 | 0.021 | 1.00 | 0.03 | 0.07 | 0.00 |
| 6L 512d MLP | 1.00 | 1.00 | 0.019 | 0.019 | 0.95 | 0.39 | 0.13 | 0.79 | 1.00 | 1.00 | 0.018 | 0.018 | 1.00 | 0.03 | 0.06 | 0.00 |
| 2L 1024d MLP | 1.01 | 1.00 | 0.020 | 0.021 | 0.98 | 0.17 | 0.06 | 0.62 | 1.00 | 1.00 | 0.020 | 0.020 | 1.00 | 0.03 | 0.06 | 0.14 |
| 4L 1024d MLP | 1.00 | 1.00 | 0.016 | 0.016 | 0.95 | 0.19 | 0.14 | 0.60 | 1.00 | 1.00 | 0.016 | 0.016 | 1.00 | 0.03 | 0.06 | 0.07 |
| 6L 1024d MLP | 1.00 | 1.00 | 0.015 | 0.015 | 0.89 | 0.49 | 0.17 | 0.89 | 1.00 | 1.00 | 0.015 | 0.015 | 0.99 | 0.06 | 0.11 | 0.43 |
| 2L 128d Tf. | 1.00 | 0.99 | 0.033 | 0.033 | 0.92 | 0.29 | 0.08 | 0.94 | 1.00 | 1.00 | 0.032 | 0.032 | 0.99 | 0.05 | 0.06 | 0.56 |
| 4L 128d Tf. | 1.00 | 1.00 | 0.027 | 0.027 | 0.90 | 0.59 | 0.10 | 0.94 | 1.00 | 1.00 | 0.026 | 0.026 | 0.99 | 0.08 | 0.06 | 0.45 |
| 6L 128d Tf. | 1.00 | 1.00 | 0.025 | 0.025 | 0.87 | 0.50 | 0.10 | 0.89 | 1.00 | 1.00 | 0.024 | 0.024 | 0.98 | 0.09 | 0.06 | 0.59 |
| 2L 256d Tf. | 1.01 | 1.00 | 0.024 | 0.024 | 0.95 | 0.48 | 0.10 | 0.94 | 1.00 | 1.00 | 0.024 | 0.024 | 0.99 | 0.06 | 0.05 | 0.36 |
| 4L 256d Tf. | 1.00 | 1.00 | 0.019 | 0.019 | 0.92 | 0.51 | 0.15 | 0.92 | 1.00 | 0.99 | 0.019 | 0.019 | 0.99 | 0.05 | 0.07 | 0.45 |
| 6L 256d Tf. | 1.00 | 1.00 | 0.017 | 0.017 | 0.91 | 0.39 | 0.13 | 0.83 | 1.00 | 1.00 | 0.017 | 0.017 | 0.99 | 0.07 | 0.06 | 0.44 |
| 2L 512d Tf. | 1.00 | 1.00 | 0.019 | 0.019 | 0.93 | 0.41 | 0.10 | 0.90 | 1.00 | 0.97 | 0.018 | 0.018 | 0.99 | 0.05 | 0.06 | 0.43 |
| 4L 512d Tf. | 1.00 | 1.00 | 0.015 | 0.015 | 0.84 | 0.61 | 0.12 | 0.86 | 1.00 | 1.00 | 0.015 | 0.015 | 0.99 | 0.06 | 0.06 | 0.72 |
| 6L 512d Tf. | 1.00 | 1.00 | 0.014 | 0.014 | 0.80 | 0.65 | 0.29 | 0.93 | 1.00 | 1.00 | 0.014 | 0.014 | 0.98 | 0.10 | 0.06 | 0.57 |
| 2L 1024d Tf. | 1.00 | 1.00 | 0.016 | 0.016 | 0.93 | 0.35 | 0.09 | 0.83 | 1.00 | 1.00 | 0.015 | 0.015 | 0.99 | 0.06 | 0.07 | 0.33 |
| 4L 1024d Tf. | 1.00 | 1.00 | 0.013 | 0.013 | 0.84 | 0.56 | 0.17 | 0.95 | 1.00 | 1.00 | 0.013 | 0.013 | 0.98 | 0.10 | 0.07 | 0.60 |
| 6L 1024d Tf. | 1.00 | 1.00 | 0.013 | 0.013 | 0.82 | 0.57 | 0.16 | 0.91 | 1.00 | 0.99 | 0.012 | 0.012 | 0.98 | 0.08 | 0.06 | 0.59 |

Figure 8: **Groups (1a) and (1b):** Non-EMA vs EMA rollout evaluations for a variety of architectures. The stabilizing effect of EMA is robust across these architectures and scales. This is most clearly seen in the range$_{\mathrm{mid}}$ column of **(1a)**, and robustly confirms our empirical result **(R1)**. Looking at the analogous column in **(1b)**, we confirm **(R4)**, that EMA uniformly stabilizes the training of these architectures.

record the **latest** iteration where the reward dips below this value. (In the table, $-$ denotes the absence of such an iteration.)

For all runs, we report the median of these statistics over 3 random seeds. Using these quantitative results, we next restate and discuss the relevant empirical findings from the main paper, and discuss these in greater depth in the figure captions. Results are shown in Figures 8, 9, 10, and 11.

(R2) **GVA arises from algorithmic suboptimality rather than an information-theoretic limit.** Even with "infinite" training data (i.e. fresh trajectories with i.i.d. initial conditions at each training step), rollout oscillations persist.

(R3) **Training oscillations are *not* mitigated by many standard approaches to regularization**, including architectural interventions and increased regularization. On the other hand, **oscillations are ameliorated by variance reduction techniques**, such as large batch sizes, learning rate decay, and iterate averaging.

(R4) **EMA iterate averaging strongly mitigates rollout oscillations**, and its performance is robust across a variety of architectures and environments.

We close with a final visualization: Figure 12 shows a scatter plot of $J_H$ vs. $\ell_{\mathrm{BC}}$ for the final iterates of *all* experimental conditions considered in this paper, analogous to Figure 6(b), showing that a generalization gap remains vs. the true (deterministic) expert policy, and that rollout reward can be uncorrelated with this gap in the regimes studied in this work.

### C.1.3 MISCELLANEOUS VISUALIZATIONS

**Additional reward and loss landscapes.** We provide additional (lower-resolution) heatmaps around neighboorhoods of various cloner policies in Figure 13. In addition to stochastic gradient directions, we visualize cross-sections of these landscapes in random directions, like the main figures in (Li et al., 2018).[12] This confirms the generality of our finding in **(R1)**, concerning the

---

[12]Note that Li et al. (2018) choose random directions because the loss landscape looks too flat in stochastic gradient directions. This provides further supporting evidence that our finding **(R1)** is generic: while the loss

**(2) Regularizers**

| | $J_{max}$ | $J_{final}$ | $\ell_{min}$ | $\ell_{final}$ | $\mu_{mid}$ | range$_{mid}$ | $t_{early}$ | $t_{worse}$ |
|---|---|---|---|---|---|---|---|---|
| (4L 1024d MLP) | 1.00 | 1.00 | 0.016 | 0.016 | 0.95 | 0.19 | 0.14 | 0.60 |
| infinite data | 1.00 | 1.00 | 0.014 | 0.014 | 0.96 | 0.53 | 0.11 | 0.51 |
| L1 loss | 1.00 | 1.00 | 0.015 | 0.015 | 0.96 | 0.21 | 0.09 | 0.76 |
| L2 loss | 1.00 | 1.00 | 0.016 | 0.016 | 0.96 | 0.24 | 0.09 | 0.62 |
| data aug. 0.005 | 1.00 | 1.00 | 0.021 | 0.021 | 0.85 | 0.64 | 0.23 | 0.99 |
| data aug. 0.01 | 0.85 | 0.77 | 0.026 | 0.026 | 0.59 | 0.47 | 0.36 | - |
| data aug. 0.05 | 0.38 | 0.33 | 0.058 | 0.058 | 0.31 | 0.12 | 0.12 | - |
| wt. decay 0.01 | 1.00 | 1.00 | 0.014 | 0.014 | 0.98 | 0.17 | 0.08 | 0.72 |
| wt. decay 0.2 | 1.00 | 1.00 | 0.018 | 0.018 | 0.91 | 0.46 | 0.20 | 0.80 |
| wt. decay 0.3 | 1.00 | 1.00 | 0.020 | 0.020 | 0.88 | 0.55 | 0.25 | 0.88 |
| dropout 0.1 | 1.00 | 0.99 | 0.023 | 0.023 | 0.92 | 0.29 | 0.20 | 0.97 |
| dropout 0.2 | 1.00 | 1.00 | 0.030 | 0.030 | 0.86 | 0.37 | 0.43 | 0.94 |
| dropout 0.3 | 1.00 | 0.97 | 0.036 | 0.036 | 0.83 | 0.50 | 0.52 | 0.95 |
| dropout 0.4 | 0.97 | 0.90 | 0.043 | 0.043 | 0.74 | 0.48 | 0.73 | - |
| dropout 0.5 | 0.94 | 0.91 | 0.052 | 0.052 | 0.65 | 0.57 | 0.70 | 0.99 |
| (4L 1024d Tf.) | 1.00 | 1.00 | 0.013 | 0.013 | 0.84 | 0.56 | 0.17 | 0.95 |
| infinite data | 1.00 | 1.00 | 0.012 | 0.012 | 0.92 | 0.50 | 0.12 | 0.76 |
| L1 loss | 1.00 | 1.00 | 0.012 | 0.012 | 0.91 | 0.43 | 0.11 | 0.91 |
| L2 loss | 1.00 | 1.00 | 0.013 | 0.013 | 0.85 | 0.61 | 0.10 | 0.92 |
| data aug. 0.005 | 1.00 | 0.95 | 0.016 | 0.016 | 0.71 | 0.62 | 0.49 | 0.99 |
| data aug. 0.01 | 0.86 | 0.80 | 0.019 | 0.019 | 0.56 | 0.44 | 0.83 | 0.99 |
| data aug. 0.05 | 0.41 | 0.38 | 0.031 | 0.031 | 0.35 | 0.14 | 0.18 | - |
| wt. decay 0.01 | 1.00 | 1.00 | 0.011 | 0.011 | 0.98 | 0.17 | 0.15 | 0.61 |
| wt. decay 0.2 | 1.00 | 1.00 | 0.017 | 0.017 | 0.72 | 0.63 | 0.72 | 0.98 |
| wt. decay 0.3 | 1.00 | 1.00 | 0.021 | 0.021 | 0.65 | 0.73 | 0.61 | 0.95 |
| ff drop 0.1 | 1.00 | 1.00 | 0.017 | 0.017 | 0.84 | 0.52 | 0.27 | 0.97 |
| ff drop 0.2 | 1.00 | 0.94 | 0.020 | 0.020 | 0.82 | 0.61 | 0.17 | - |
| ff drop 0.3 | 1.00 | 0.78 | 0.023 | 0.023 | 0.80 | 0.66 | 0.26 | - |
| ff drop 0.4 | 0.99 | 0.72 | 0.026 | 0.026 | 0.72 | 0.67 | 0.23 | - |
| ff drop 0.5 | 0.97 | 0.60 | 0.029 | 0.029 | 0.67 | 0.66 | 0.36 | - |
| emb. drop 0.1 | 0.87 | 0.82 | 0.020 | 0.020 | 0.60 | 0.53 | 0.13 | - |
| emb. drop 0.2 | 0.71 | 0.55 | 0.030 | 0.031 | 0.45 | 0.37 | 0.29 | - |
| emb. drop 0.3 | 0.52 | 0.23 | 0.042 | 0.052 | 0.35 | 0.28 | 0.10 | - |
| emb. drop 0.4 | 0.42 | 0.20 | 0.055 | 0.105 | 0.26 | 0.14 | 0.08 | - |
| emb. drop 0.5 | 0.33 | 0.19 | 0.072 | 0.276 | 0.22 | 0.05 | 0.07 | - |

**(3) Optimization**

| | $J_{max}$ | $J_{final}$ | $\ell_{min}$ | $\ell_{final}$ | $\mu_{mid}$ | range$_{mid}$ | $t_{early}$ | $t_{worse}$ |
|---|---|---|---|---|---|---|---|---|
| (4L 1024d MLP) | 1.00 | 1.00 | 0.016 | 0.016 | 0.95 | 0.19 | 0.14 | 0.60 |
| $\beta_1,\beta_2=0.7,0.9$ | 1.00 | 1.00 | 0.016 | 0.016 | 0.89 | 0.54 | 0.28 | 0.79 |
| $\beta_1,\beta_2=0.7,0.99$ | 1.00 | 1.00 | 0.016 | 0.016 | 0.87 | 0.54 | 0.29 | 0.89 |
| $\beta_1,\beta_2=0.7,0.999$ | 1.00 | 1.00 | 0.016 | 0.016 | 0.89 | 0.44 | 0.38 | 0.80 |
| $\beta_1,\beta_2=0.9,0.9$ | 1.00 | 1.00 | 0.016 | 0.016 | 0.95 | 0.33 | 0.15 | 0.65 |
| $\beta_1,\beta_2=0.9,0.99$ | 1.00 | 1.00 | 0.016 | 0.016 | 0.94 | 0.25 | 0.13 | 0.68 |
| $\beta_1,\beta_2=0.9,0.999$ | 1.00 | 1.00 | 0.016 | 0.016 | 0.94 | 0.31 | 0.15 | 0.69 |
| $\beta_1,\beta_2=0.95,0.9$ | 1.00 | 1.00 | 0.017 | 0.017 | 0.97 | 0.24 | 0.08 | 0.66 |
| $\beta_1,\beta_2=0.95,0.99$ | 1.00 | 1.00 | 0.016 | 0.016 | 0.96 | 0.51 | 0.12 | 0.64 |
| $\beta_1,\beta_2=0.95,0.999$ | 1.00 | 1.00 | 0.016 | 0.016 | 0.95 | 0.34 | 0.09 | 0.66 |
| $\beta_1,\beta_2=0.99,0.9$ | - | - | - | - | - | - | - | - |
| $\beta_1,\beta_2=0.99,0.99$ | 1.01 | 1.00 | 0.019 | 0.019 | 0.97 | 0.13 | 0.09 | 0.58 |
| $\beta_1,\beta_2=0.99,0.999$ | 1.00 | 1.00 | 0.018 | 0.018 | 0.97 | 0.22 | 0.11 | 0.72 |
| (4L 1024d MLP) | 1.00 | 1.00 | 0.013 | 0.013 | 0.84 | 0.56 | 0.17 | 0.95 |
| $\beta_1,\beta_2=0.7,0.9$ | 1.00 | 1.00 | 0.013 | 0.013 | 0.77 | 0.57 | 0.60 | 0.91 |
| $\beta_1,\beta_2=0.7,0.99$ | 1.00 | 0.97 | 0.013 | 0.013 | 0.79 | 0.63 | 0.26 | 0.95 |
| $\beta_1,\beta_2=0.7,0.999$ | 1.00 | 1.00 | 0.013 | 0.013 | 0.84 | 0.67 | 0.18 | 0.91 |
| $\beta_1,\beta_2=0.9,0.9$ | 1.00 | 1.00 | 0.013 | 0.013 | 0.87 | 0.44 | 0.23 | 0.92 |
| $\beta_1,\beta_2=0.9,0.99$ | 1.00 | 1.00 | 0.013 | 0.013 | 0.84 | 0.56 | 0.17 | 0.94 |
| $\beta_1,\beta_2=0.9,0.999$ | 1.00 | 1.00 | 0.013 | 0.013 | 0.86 | 0.54 | 0.33 | 0.94 |
| $\beta_1,\beta_2=0.95,0.9$ | 1.00 | 1.00 | 0.013 | 0.013 | 0.91 | 0.41 | 0.12 | 0.82 |
| $\beta_1,\beta_2=0.95,0.99$ | 1.00 | 1.00 | 0.013 | 0.013 | 0.87 | 0.62 | 0.17 | 0.94 |
| $\beta_1,\beta_2=0.95,0.999$ | 1.00 | 1.00 | 0.013 | 0.013 | 0.87 | 0.49 | 0.10 | 0.90 |
| $\beta_1,\beta_2=0.99,0.9$ | 1.00 | 1.00 | 0.014 | 0.014 | 0.84 | 0.65 | 0.30 | 0.88 |
| $\beta_1,\beta_2=0.99,0.99$ | 1.00 | 1.00 | 0.014 | 0.014 | 0.91 | 0.53 | 0.12 | 0.93 |
| $\beta_1,\beta_2=0.99,0.999$ | 1.00 | 1.00 | 0.013 | 0.013 | 0.93 | 0.32 | 0.14 | 0.91 |

**(4) Variance reduction**

| | $J_{max}$ | $J_{final}$ | $\ell_{min}$ | $\ell_{final}$ | $\mu_{mid}$ | range$_{mid}$ | $t_{early}$ | $t_{worse}$ |
|---|---|---|---|---|---|---|---|---|
| (4L 1024d MLP) | 1.00 | 1.00 | 0.016 | 0.016 | 0.95 | 0.19 | 0.14 | 0.60 |
| $\eta^{0.5}$ lr decay | 1.00 | 1.00 | 0.018 | 0.018 | 0.91 | 0.38 | 0.08 | 0.89 |
| $\eta^{1.5}$ lr decay | 1.00 | 0.97 | 0.015 | 0.015 | 0.96 | 0.34 | 0.12 | 0.60 |
| $\eta^{2}$ lr decay | 1.00 | 1.00 | 0.015 | 0.015 | 0.96 | 0.16 | 0.08 | 0.56 |
| $\eta^{2.5}$ lr decay | 1.00 | 1.00 | 0.015 | 0.015 | 0.98 | 0.17 | 0.07 | 0.53 |
| $\eta^{3}$ lr decay | 1.00 | 1.00 | 0.015 | 0.016 | 0.99 | 0.09 | 0.12 | 0.32 |
| 2x batch size | 1.00 | 0.97 | 0.025 | 0.028 | 0.92 | 0.44 | 0.16 | 0.99 |
| 4x batch size | 1.00 | 1.00 | 0.021 | 0.023 | 0.96 | 0.22 | 0.06 | 0.96 |
| 16x batch size | 1.01 | 1.00 | 0.019 | 0.022 | 0.98 | 0.13 | 0.05 | 0.94 |
| MLP + EMA | 1.00 | 1.00 | 0.016 | 0.016 | 1.00 | 0.03 | 0.06 | 0.07 |
| (4L 1024d Tf.) | 1.00 | 1.00 | 0.013 | 0.013 | 0.84 | 0.56 | 0.17 | 0.95 |
| $\eta^{0.5}$ lr decay | 1.00 | 1.00 | 0.016 | 0.016 | 0.81 | 0.62 | 0.16 | 0.98 |
| $\eta^{1.5}$ lr decay | 1.00 | 1.00 | 0.012 | 0.012 | 0.91 | 0.49 | 0.07 | 0.83 |
| $\eta^{2}$ lr decay | 1.00 | 1.00 | 0.012 | 0.012 | 0.93 | 0.44 | 0.23 | 0.71 |
| $\eta^{2.5}$ lr decay | 1.00 | 1.00 | 0.012 | 0.012 | 0.95 | 0.49 | 0.17 | 0.62 |
| $\eta^{3}$ lr decay | 1.00 | 1.00 | 0.012 | 0.012 | 0.97 | 0.31 | 0.12 | 0.59 |
| 2x batch size | 0.99 | 0.76 | 0.024 | 0.032 | 0.80 | 0.58 | 0.09 | - |
| 4x batch size | 1.00 | 0.87 | 0.020 | 0.022 | 0.85 | 0.45 | 0.07 | - |
| 16x batch size | 1.00 | 0.97 | 0.015 | 0.020 | 0.93 | 0.37 | 0.06 | 0.98 |
| Tf. + EMA | 1.00 | 1.00 | 0.013 | 0.013 | 0.98 | 0.10 | 0.07 | 0.60 |

**(5) EMA parameters**

| | $J_{max}$ | $J_{final}$ | $\ell_{min}$ | $\ell_{final}$ | $\mu_{mid}$ | range$_{mid}$ | $t_{early}$ | $t_{worse}$ |
|---|---|---|---|---|---|---|---|---|
| (4L 1024d MLP) | 1.00 | 1.00 | 0.016 | 0.016 | 0.95 | 0.19 | 0.14 | 0.60 |
| (MLP + EMA) | 1.00 | 1.00 | 0.016 | 0.016 | 1.00 | 0.03 | 0.06 | 0.07 |
| freq. 2 | 1.00 | 1.00 | 0.016 | 0.016 | 0.99 | 0.03 | 0.06 | 0.14 |
| freq. 5 | 1.00 | 1.00 | 0.016 | 0.016 | 1.00 | 0.03 | 0.06 | 0.25 |
| freq. 10 | 1.00 | 1.00 | 0.018 | 0.018 | 0.99 | 0.09 | 0.17 | 0.41 |
| freq. 20 | 1.00 | 1.00 | 0.023 | 0.023 | 0.84 | 0.37 | 0.50 | 0.81 |
| freq. 50 | 0.53 | 0.41 | 0.089 | 0.221 | 0.26 | 0.09 | 0.93 | - |
| freq. 100 | 0.35 | 0.22 | 0.077 | 0.604 | 0.21 | 0.01 | 0.02 | - |
| burn-in 0 | 1.00 | 1.00 | 0.016 | 0.016 | 0.99 | 0.03 | 0.06 | 0.12 |
| burn-in 50 | 1.00 | 1.00 | 0.016 | 0.016 | 1.00 | 0.03 | 0.07 | 0.12 |
| burn-in 100 | 1.00 | 1.00 | 0.016 | 0.016 | 1.00 | 0.03 | 0.06 | 0.12 |
| burn-in 1000 | 1.00 | 1.00 | 0.016 | 0.016 | 1.00 | 0.03 | 0.07 | 0.12 |
| burn-in 10000 | 1.00 | 1.00 | 0.016 | 0.016 | 0.99 | 0.04 | 0.06 | 0.12 |
| annealing $\alpha=1$ | 1.00 | 1.00 | 0.016 | 0.016 | 0.99 | 0.03 | 0.14 | 0.26 |
| no burn-in, $\alpha=1$ | 1.00 | 1.00 | 0.016 | 0.016 | 0.99 | 0.04 | 0.15 | 0.26 |
| (4L 1024d Tf.) | 1.00 | 1.00 | 0.013 | 0.013 | 0.84 | 0.56 | 0.17 | 0.95 |
| (Tf. + EMA) | 1.00 | 1.00 | 0.013 | 0.013 | 0.98 | 0.10 | 0.07 | 0.60 |
| freq. 2 | 1.00 | 0.98 | 0.013 | 0.013 | 0.98 | 0.09 | 0.07 | 0.57 |
| freq. 5 | 1.00 | 1.00 | 0.014 | 0.014 | 0.96 | 0.13 | 0.13 | 0.67 |
| freq. 10 | 0.99 | 0.99 | 0.023 | 0.023 | 0.84 | 0.33 | 0.63 | 0.96 |
| freq. 20 | 0.86 | 0.62 | 0.058 | 0.076 | 0.57 | 0.51 | 0.71 | - |
| freq. 50 | 0.42 | 0.20 | 0.087 | 0.272 | 0.21 | 0.05 | 0.01 | - |
| freq. 100 | 0.31 | 0.21 | 0.091 | 0.713 | 0.22 | 0.05 | 0.01 | - |
| burn-in 0 | 1.00 | 1.00 | 0.013 | 0.013 | 0.98 | 0.07 | 0.06 | 0.89 |
| burn-in 50 | 1.00 | 1.00 | 0.013 | 0.013 | 0.98 | 0.08 | 0.06 | 0.57 |
| burn-in 100 | 1.00 | 1.00 | 0.013 | 0.013 | 0.98 | 0.07 | 0.07 | 0.57 |
| burn-in 1000 | 1.00 | 1.00 | 0.013 | 0.013 | 0.98 | 0.10 | 0.06 | 0.57 |
| burn-in 10000 | 1.00 | 1.00 | 0.013 | 0.013 | 0.98 | 0.05 | 0.06 | 0.87 |
| annealing $\alpha=1$ | 1.00 | 1.00 | 0.013 | 0.013 | 0.98 | 0.09 | 0.24 | 0.58 |
| no burn-in, $\alpha=1$ | 1.00 | 1.00 | 0.013 | 0.013 | 0.98 | 0.10 | 0.23 | 0.58 |

Figure 9: **Groups (2) through (5):** Effects of various algorithmic choices: regularization, optimizer hyperparameters, variance reduction, and EMA hyperparameters. **(2)** Standard regularizers do not reduce the oscillations (and often worsen them) **(R2, R3)**. **(3)** Large momentum has a stabilizing effect on the rewards, presumably via variance reduction **(R4)**. However, it can result in other (well-known) optimization instabilities, leading to non-convergence. **(4)** Variance reduction techniques *do* work **(R4)**; among these, EMA is the most robust (see especially the range$_{\mathrm{mid}}$ column, measuring oscillation amplitude). **(5)** EMA tends to fail if updated too infrequently; an initial burn-in period is helpful for excluding poor initial iterates. EMA with suboptimal annealing ($\alpha$, so that $\gamma^{(t)} = t^{-\alpha}$) can delay learning (see $t_{\mathrm{early}}$ column), but still works.

| | (6a) Dataset size & lr decay ablations, no EMA | | | | | | | | (6b) + EMA | | | | | | | |
|---|---|---|---|---|---|---|---|---|---|---|---|---|---|---|---|---|
| | $J_{max}$ | $J_{final}$ | $\ell_{min}$ | $\ell_{final}$ | $\mu_{mid}$ | range$_{mid}$ | $t_{early}$ | $t_{worse}$ | $J_{max}$ | $J_{final}$ | $\ell_{min}$ | $\ell_{final}$ | $\mu_{mid}$ | range$_{mid}$ | $t_{early}$ | $t_{worse}$ |
| (4L 1024d MLP) | 1.00 | 1.00 | 0.016 | 0.016 | 0.95 | 0.19 | 0.14 | 0.60 | 1.00 | 1.00 | 0.016 | 0.016 | 1.00 | 0.03 | 0.06 | 0.07 |
| 10 trajectories | 0.37 | 0.32 | 0.082 | 0.085 | 0.30 | 0.08 | 0.13 | - | 0.35 | 0.32 | 0.080 | 0.085 | 0.31 | 0.06 | 0.08 | - |
| 20 trajectories | 0.52 | 0.47 | 0.060 | 0.062 | 0.38 | 0.19 | 0.57 | - | 0.54 | 0.44 | 0.060 | 0.062 | 0.42 | 0.14 | 0.78 | - |
| 50 trajectories | 0.72 | 0.55 | 0.041 | 0.041 | 0.46 | 0.40 | 0.53 | - | 0.76 | 0.63 | 0.041 | 0.041 | 0.58 | 0.24 | 0.79 | - |
| 100 trajectories | 0.95 | 0.80 | 0.030 | 0.030 | 0.65 | 0.54 | 0.82 | - | 0.94 | 0.83 | 0.030 | 0.030 | 0.79 | 0.21 | 0.27 | - |
| 200 trajectories | 1.00 | 1.00 | 0.023 | 0.023 | 0.80 | 0.52 | 0.49 | 0.99 | 1.00 | 0.96 | 0.023 | 0.023 | 0.93 | 0.14 | 0.16 | 0.99 |
| 500 trajectories | 1.00 | 1.00 | 0.018 | 0.018 | 0.92 | 0.26 | 0.25 | 0.83 | 1.00 | 1.00 | 0.018 | 0.018 | 0.99 | 0.05 | 0.10 | 0.32 |
| 10 traj. + const lr | 0.35 | 0.26 | 0.082 | 0.100 | 0.28 | 0.10 | 0.06 | - | 0.36 | 0.28 | 0.081 | 0.096 | 0.29 | 0.06 | 0.02 | - |
| 20 traj. + const lr | 0.44 | 0.27 | 0.061 | 0.078 | 0.32 | 0.15 | 0.17 | - | 0.45 | 0.36 | 0.060 | 0.070 | 0.34 | 0.10 | 0.08 | - |
| 50 traj. + const lr | 0.62 | 0.47 | 0.044 | 0.050 | 0.41 | 0.35 | 0.40 | - | 0.66 | 0.49 | 0.042 | 0.046 | 0.50 | 0.27 | 0.09 | - |
| 100 traj. + const lr | 0.73 | 0.66 | 0.036 | 0.041 | 0.48 | 0.43 | 0.37 | 0.99 | 0.79 | 0.59 | 0.033 | 0.035 | 0.68 | 0.26 | 0.10 | - |
| 200 traj. + const lr | 0.91 | 0.60 | 0.033 | 0.051 | 0.63 | 0.56 | 0.16 | - | 0.99 | 0.85 | 0.027 | 0.028 | 0.89 | 0.20 | 0.19 | - |
| 500 traj. + const lr | 0.99 | 0.89 | 0.033 | 0.036 | 0.77 | 0.61 | 0.28 | - | 1.00 | 0.98 | 0.023 | 0.023 | 0.98 | 0.09 | 0.10 | 0.90 |
| (4L 1024d Tf. ) | 1.00 | 1.00 | 0.013 | 0.013 | 0.84 | 0.56 | 0.17 | 0.95 | 1.00 | 1.00 | 0.013 | 0.013 | 0.98 | 0.10 | 0.07 | 0.60 |
| 10 trajectories | 0.33 | 0.28 | 0.081 | 0.090 | 0.29 | 0.06 | 0.05 | - | 0.37 | 0.26 | 0.077 | 0.090 | 0.29 | 0.06 | 0.06 | - |
| 20 trajectories | 0.41 | 0.33 | 0.060 | 0.063 | 0.32 | 0.15 | 0.19 | - | 0.46 | 0.31 | 0.056 | 0.063 | 0.33 | 0.10 | 0.07 | - |
| 50 trajectories | 0.62 | 0.48 | 0.041 | 0.041 | 0.41 | 0.27 | 0.21 | - | 0.71 | 0.48 | 0.039 | 0.041 | 0.48 | 0.20 | 0.15 | - |
| 100 trajectories | 0.89 | 0.77 | 0.029 | 0.029 | 0.58 | 0.52 | 0.80 | - | 0.90 | 0.81 | 0.028 | 0.028 | 0.74 | 0.25 | 0.10 | - |
| 200 trajectories | 0.97 | 0.92 | 0.021 | 0.021 | 0.71 | 0.56 | 0.94 | - | 1.00 | 0.93 | 0.021 | 0.021 | 0.88 | 0.20 | 0.41 | - |
| 500 trajectories | 1.00 | 1.00 | 0.015 | 0.015 | 0.85 | 0.57 | 0.22 | 0.92 | 1.00 | 1.00 | 0.015 | 0.015 | 0.98 | 0.07 | 0.13 | 0.87 |
| 10 traj. + const lr | 0.36 | 0.26 | 0.081 | 0.097 | 0.29 | 0.09 | 0.14 | - | 0.35 | 0.28 | 0.077 | 0.093 | 0.30 | 0.06 | 0.03 | - |
| 20 traj. + const lr | 0.39 | 0.32 | 0.060 | 0.068 | 0.31 | 0.12 | 0.16 | - | 0.44 | 0.32 | 0.056 | 0.067 | 0.31 | 0.09 | 0.06 | - |
| 50 traj. + const lr | 0.62 | 0.38 | 0.043 | 0.049 | 0.38 | 0.31 | 0.18 | - | 0.68 | 0.49 | 0.039 | 0.045 | 0.48 | 0.22 | 0.15 | - |
| 100 traj. + const lr | 0.77 | 0.51 | 0.035 | 0.040 | 0.48 | 0.45 | 0.18 | - | 0.86 | 0.61 | 0.030 | 0.033 | 0.65 | 0.27 | 0.12 | - |
| 200 traj. + const lr | 0.84 | 0.60 | 0.031 | 0.033 | 0.61 | 0.50 | 0.14 | - | 0.97 | 0.82 | 0.025 | 0.027 | 0.86 | 0.21 | 0.09 | - |
| 500 traj. + const lr | 0.93 | 0.60 | 0.030 | 0.033 | 0.68 | 0.62 | 0.22 | - | 1.00 | 0.97 | 0.021 | 0.022 | 0.94 | 0.13 | 0.17 | 0.98 |

Figure 10: **Groups (6a) and (6b):** Ablations of dataset size and learning rate decay. When learning from fewer trajectories, optimization instabilities can become even worse. In such regimes, **the stabilizing effect of EMA results in end-to-end improvements in sample efficiency.** Furthermore, EMA can replace learning rate decay. This refines **(R4)**, showing that mitigating the non-statistical phenomenon of GVA can manifest data-efficiency in practice.

| | (7a) Other environments, no EMA | | | | | | | | (7b) + EMA | | | | | | | |
|---|---|---|---|---|---|---|---|---|---|---|---|---|---|---|---|---|
| | $J_{max}$ | $J_{final}$ | $\ell_{min}$ | $\ell_{final}$ | $\mu_{mid}$ | range$_{mid}$ | $t_{early}$ | $t_{worse}$ | $J_{max}$ | $J_{final}$ | $\ell_{min}$ | $\ell_{final}$ | $\mu_{mid}$ | range$_{mid}$ | $t_{early}$ | $t_{worse}$ |
| **Hopper**: MLP | 1.00 | 0.98 | 0.018 | 0.018 | 0.87 | 0.37 | 0.28 | 0.98 | 1.00 | 0.97 | 0.018 | 0.018 | 0.93 | 0.12 | 0.15 | 0.98 |
| Transformer | 0.88 | 0.85 | 0.018 | 0.019 | 0.65 | 0.32 | 0.93 | 1.00 | 0.89 | 0.85 | 0.018 | 0.018 | 0.66 | 0.10 | 0.96 | 0.99 |
| MLP + no lr decay | 0.95 | 0.73 | 0.035 | 0.042 | 0.77 | 0.46 | 0.03 | - | 0.96 | 0.89 | 0.026 | 0.026 | 0.88 | 0.15 | 0.02 | - |
| Tf. + no lr decay | 0.82 | 0.63 | 0.033 | 0.035 | 0.63 | 0.25 | 0.05 | - | 0.83 | 0.61 | 0.026 | 0.027 | 0.62 | 0.09 | 0.01 | - |
| **HalfCheetah**: MLP | 0.67 | 0.60 | 0.011 | 0.011 | 0.39 | 0.29 | 0.97 | - | 0.70 | 0.54 | 0.011 | 0.011 | 0.42 | 0.25 | 0.98 | - |
| Transformer | 0.62 | 0.54 | 0.013 | 0.013 | 0.39 | 0.30 | 0.97 | - | 0.62 | 0.56 | 0.011 | 0.011 | 0.34 | 0.23 | 0.99 | - |
| MLP + no lr decay | 0.50 | 0.27 | 0.030 | 0.046 | 0.36 | 0.32 | 0.26 | - | 0.48 | 0.34 | 0.022 | 0.022 | 0.37 | 0.22 | 0.36 | - |
| Tf. + no lr decay | 0.51 | 0.32 | 0.029 | 0.037 | 0.34 | 0.27 | 0.08 | - | 0.48 | 0.26 | 0.023 | 0.028 | 0.30 | 0.20 | 0.11 | - |
| **Ant**: MLP | 1.04 | 0.97 | 0.051 | 0.052 | 1.00 | 0.15 | 0.03 | - | 1.04 | 1.00 | 0.051 | 0.052 | 1.00 | 0.11 | 0.02 | 0.96 |
| Transformer | 1.04 | 1.03 | 0.050 | 0.050 | 1.00 | 0.15 | 0.03 | 0.99 | 1.04 | 0.99 | 0.049 | 0.050 | 1.00 | 0.16 | 0.02 | 0.98 |
| MLP + no lr decay | 1.04 | 0.97 | 0.075 | 0.083 | 0.99 | 0.14 | 0.02 | - | 1.04 | 1.01 | 0.059 | 0.060 | 1.00 | 0.11 | 0.02 | 0.98 |
| Tf. + no lr decay | 1.04 | 1.04 | 0.072 | 0.076 | 1.00 | 0.12 | 0.04 | 0.98 | 1.04 | 1.03 | 0.060 | 0.061 | 0.99 | 0.13 | 0.02 | 0.99 |
| **Humanoid**: MLP | 1.01 | 1.00 | 0.020 | 0.020 | 0.99 | 0.03 | 0.01 | 0.00 | 1.00 | 1.00 | 0.020 | 0.020 | 1.00 | 0.03 | 0.01 | 0.00 |
| Transformer | 1.01 | 1.00 | 0.018 | 0.018 | 0.99 | 0.03 | 0.01 | 0.00 | 1.00 | 0.99 | 0.018 | 0.018 | 0.99 | 0.02 | 0.01 | 0.00 |
| MLP + no lr decay | 1.01 | 0.99 | 0.060 | 0.069 | 0.99 | 0.05 | 0.01 | 0.00 | 1.00 | 0.99 | 0.043 | 0.043 | 0.99 | 0.03 | 0.01 | 0.00 |
| Tf. + no lr decay | 1.01 | 0.99 | 0.048 | 0.051 | 0.99 | 0.05 | 0.01 | 0.28 | 1.00 | 0.98 | 0.033 | 0.033 | 0.99 | 0.02 | 0.01 | 0.00 |

Figure 11: **Groups (7a) and (7b):** Evaluation of non-EMA and EMA cloned agents on additional tasks, showing a broader scope for **(R4)** than just the `Walker2d` environment. Improvements of an identical nature are observed for `Hopper`. Sometimes, behavior cloning does not fully work (`HalfCheetah`), or the reward oscillations are more benign (`Ant`, `Humanoid`), but **EMA stabilizes training in all of these circumstances**, never resulting in a substantially worse cloned policy.

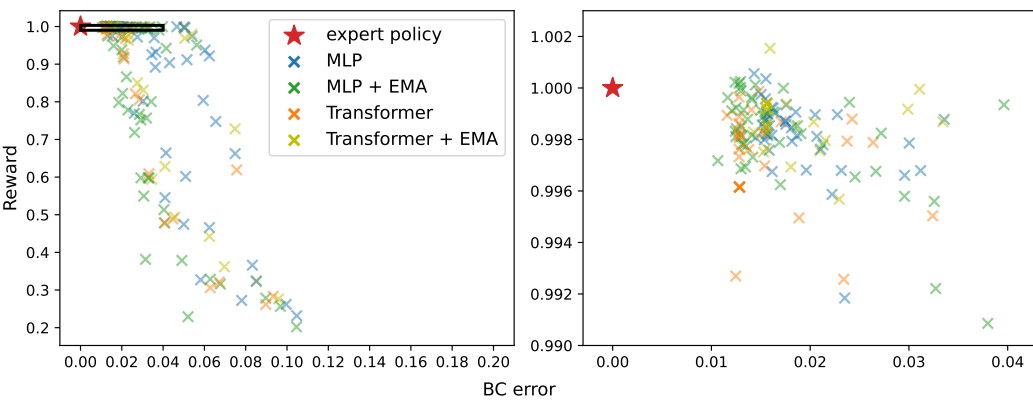

Figure 12: **Do BC agents succeed by perfectly cloning the expert?** Global scatter plot of final-iterate rollout reward $J_H$ vs. 1-step prediction error $\ell_{\mathrm{BC}}$ for *all* `Walker2d` agents trained in this paper. The plot on the right is a zoomed-in region (shown on the left). From (1) the lack of convergence to perfect agreement with the expert policy (including with access to infinite data) and (2) the lack of a strong correlation between the surrogate objective $\ell_{\mathrm{BC}}$ and true objective $J_H$ in the regime of high performance, we illustrate the following conclusion: **although statistical convergence is sufficient for behavior cloning, it is not necessary.** Deep imitation learning succeeds in a regime that is short of perfectly recovering the demonstrator, and successes and failures are instead governed by fine-grained algorithmic choices.

fractality of the reward landscape in regions where the loss is extremely smooth and near-convex. The setup is identical to that described for Figures 1(c) and 6(c), except the following: models are 4-layer 1024-dimensional {MLPs, Transformers} {with, without} EMA; checkpoint indices {1, 2} denote training steps 50000 and 205000; step size increments are varied by $3 \times \{10^{-3}, 10^{-4}, 10^{-5}\}$ (labeled as neighborhood sizes $\{10^{-2}, 10^{-3}, 10^{-4}\}$)

**Non-Markovianity of the Transformer cloner.** Of interest (but somewhat orthogonal to the scope of this paper), our behavior-cloned Transformer agents are *reparameterizations* of the original MLP expert policies. They are non-Markov, in the sense that their architecture allows their actions to depend on a sliding window of state history, rather than only the most recent state. A natural question is whether the internal representations of Transformer agents correspond to the formation of meaningful non-Markov circuits (Elhage et al., 2021; Edelman et al., 2022; Liu et al., 2023b). Preliminary visualizations in Figure 14 suggest that this is indeed the case: attention heads appear to attend to the past, perhaps learning internal "in-context resets" (Liu et al., 2023a). Similar findings appear in the nascent "sequence models for continuous control" literature (Janner et al., 2021; Shafiullah et al., 2022).

## C.2    NATURAL LANGUAGE GENERATION

In this section, we describe the autoregressive language modeling experiments in full detail. Throughout this section, we use the same standard Transformer architecture, with 12 layers, embedding dimension 1536, context length 1024, and sinusoidal position encodings. We use BPE tokenizers (Sennrich et al., 2015) with vocabulary size 2000 for TinyStories (version 2, with some postprocessing to remove Unicode glitches; 598M tokens), and 32000 for English Wikipedia (5.3B tokens, with a random 99:1 train-validation split). We emphasize that the green lines in the bottom row are scatter plots.

The purpose of using the LLM-synthesized TinyStories corpus (Eldan & Li, 2023), in accordance with the authors' intent, is to provide a significantly cheaper proxy for all algorithmic considerations

---

landscape looks smooth, the landscape of the long-horizon rollout rewards can appear to be fractal in the same regions.

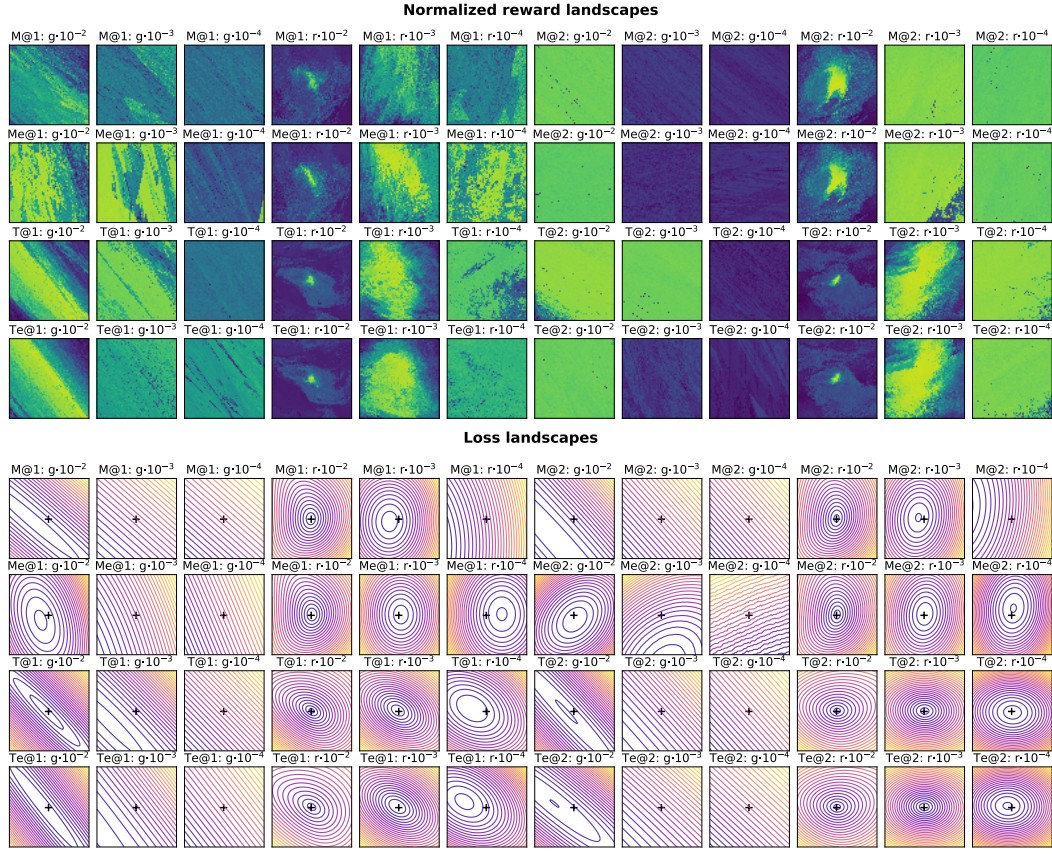

Figure 13: Additional visualizations of reward $J_H(\pi_{\boldsymbol{\theta}})$ and loss $\ell_{\mathrm{BC}}(\pi_{\boldsymbol{\theta}})$ landscapes around neighborhoods of policies. {M, T} = {MLP, Transformer}; e = EMA; {1, 2} = {early, late} checkpoints; {g, r} = {gradient, random} directions. The number (e.g. $10^{-3}$) denotes neighborhood radius.

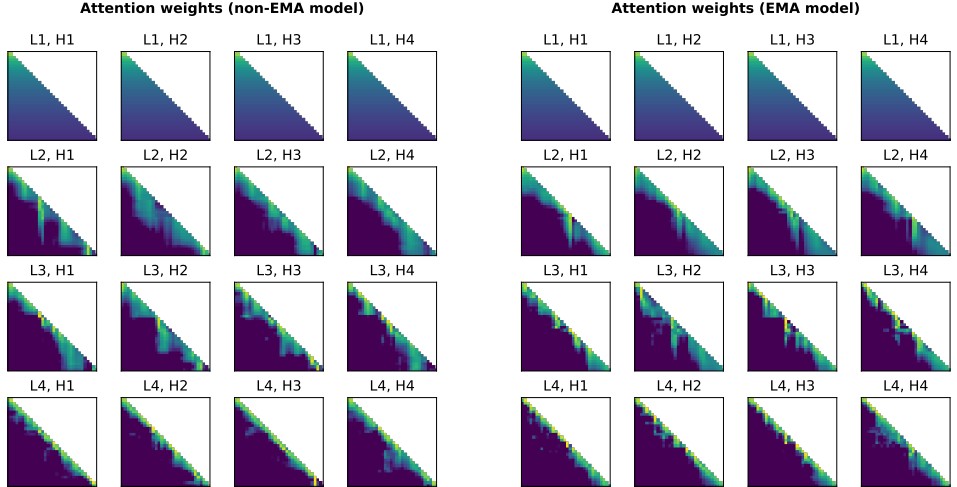

Figure 14: Heatmaps of attention weights of 4-layer, 4-head, 1024-dimensional Transformer cloners, evaluated on their own rollouts (evaluated at step 200, where a non-Markov policy has access to the past 32 states in its context). Cloned agents seem to learn policies which compute sliding-window filters in the first layer, and attend to history in a systematic manner (which we do not attempt to decode). This serves as a cursory check that the policies are indeed non-Markov, a property that is impossible for the MLP-based models to share. Weights are taken from the final checkpoint, and both agents incur normalized rewards of $> 0.99$.

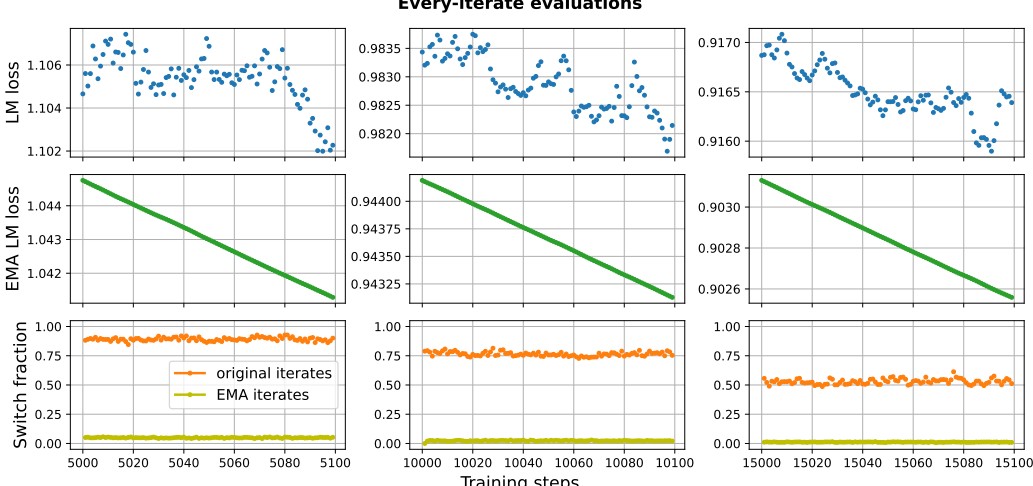

Figure 15: Zoomed-in every-iterate evaluations from the 270M-parameter TinyStories training procedure, enlarged from Figure 4 *(top left)* in the main paper. These evaluation phases occur for 100 iterations each, starting from checkpoints 5000, 10000, and 15000 (the total number of steps is 18062). The first two rows show minuscule fluctuations in the "behavior cloning" (next-token-prediction) loss. The third row measures **rollout disagreement between consecutive gradient iterates**: the fraction of input sequences where the deterministic (argmax) autoregressive generations at checkpoints steps $t$ and $t-1$. As the losses fluctuate, pairs of consecutive checkpoints disagree on >50% of autoregressively-generated deterministic completions. This is only true for <5% of EMA iterates $\widetilde{\boldsymbol{\theta}}_\gamma^{(t)}$.

relevant to the large language model pretraining pipeline. This mitigates the common occurrence of *undertraining* when benchmarking LLM training pipelines, while allowing for carefully controlled experiments at a reasonable computational cost. Nonetheless, the fact that TinyStories is a proxy distribution comprises a limitation of this empirical study. We hope to address this gap in future work (and also encourage interested parties to incorporate EMA without learning rate decay into their LM pipelines).

Each model is trained on a single node with 8 NVIDIA V100 GPUs, with a global batch size of 64 (thus, 65536 tokens), using the AdamW (Loshchilov & Hutter, 2017) optimizer, learning rates $\{3, 5, 8\} \times 10^{-4}$, and weight decay 0.1. For the 2-epoch TinyStories runs, a finer-grained grid of learning rates is used: $\eta = \{2, 3, 4, 5, 7, 8\} \times 10^{-4}$. For Wikipedia, we only use $3 \times 10^{-4}$ (as a quick preliminary check that our qualitative findings remain on natural data). 1000 steps of learning rate warmup are used throughout these experiments. With this hardware setup, the TinyStories training runs take 4 hours per epoch, while a single-epoch Wikipedia training run takes 50 hours.

We restate the empirical finding from the main paper, then discuss each aspect in depth.

(R5) Autoregressive LMs exhibit significant rollout oscillations throughout training. **EMA stabilizes the trajectory, accelerates training, and improves generalization**, complementing (and potentially obviating) standard practices in learning rate annealing.

**Small fluctuations in the "BC" loss.** In the training run with 2 epochs (18K steps) learning rate decay, and learning rate $3 \times 10^{-4}$, we perform zoomed-in evaluations of every iterate $t \in \{5001, \ldots, 5099\} \cup \{10001, \ldots, 10099\} \cup \{15001, \ldots, 15099\}$. Figure 4 *(top and middle rows)* shows that there are small fluctuations in terms of the validation log loss. Strikingly, upon performing the same evaluations on the EMA iterates, progress is extremely smooth in time (nearly indistinguishable from linear). Note that this is *not* to illustrate GVA– in this setting, the next-token-prediction log loss is $\ell_{\mathrm{BC}}$. Indeed, these fluctuations are minuscule.

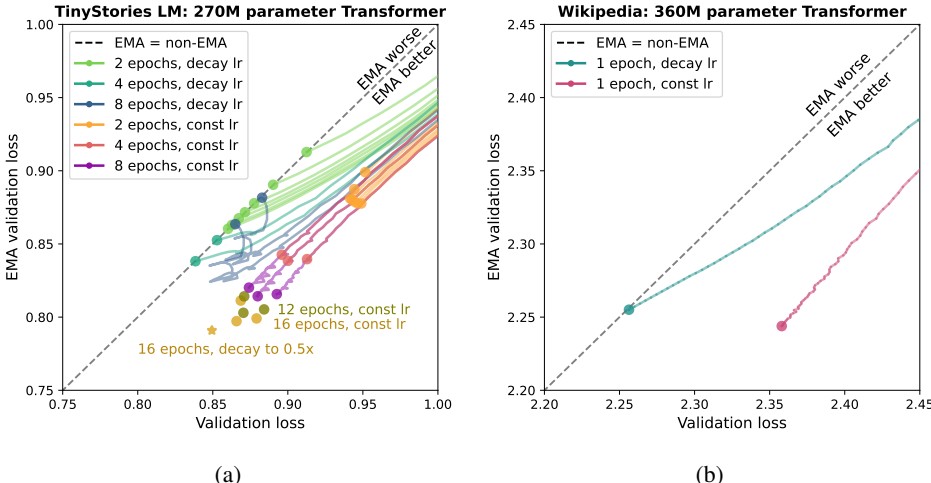

Figure 16: Training paths of autoregressive LMs in (model loss, EMA loss) space, for (a) TinyStories and (b) English Wikipedia language models. In both cases, we find that with a suitable $\beta$ schedule (1000-iteration burn-in and $\gamma \propto t^{-0.67}$ annealing), **EMA never hurts validation perplexity**. With linear learning rate annealing (down to 0), the optimizer and EMA iterates arrive at the same final validation loss. (In a limited set of experiments, findings are identical with cosine decay). On the other hand, training at a constant learning rate, **EMA iterates are better, even though the optimizer iterates are worse**. Training pipelines which decouple optimization from stabilization can reduce overfitting and produce lower-loss models.

**Large rollout oscillations due to GVA.** Figure 4 *(bottom)* provides a quick check that while local oscillations in the loss are small, the generations of these models diverge over long-horizon autoregressive rollouts. Figure 4 *(third row)* makes this quantitative: for 1000 examples from the TinyStories validation set, we take 4 prefixes, of length {10%, 25%, 50%, 75%}, and query each of the 300 model checkpoints for their argmax completion. For each checkpoint $t$ in the intervals above, we record a *switching cost*: the fraction of prefixes ("in-distribution prompts") for which generations from $\theta^{t-1}$ and $\theta^t$ differ; these are the orange and yellow dots per iterate. The mean switching costs on the respective intervals are {89.2%, 76.5%, 53.0%} for the optimizer iterates, and {4.9%, 2.3%, 1.1%} for the EMA iterates. In short, these numbers can be interpreted as **probabilities that the greedy step-wise MLE sequence of consecutive LM training iterates disagree over long rollouts,** conditioned on in-distribution prefixes.

**Improvements in validation loss.** In the continuous control settings, we do not pay much attention to analyzing the effects of trajectory smoothing on the behavior cloning loss, because obtaining a high-reward policy is the clear primary objective. In language generation, $\ell_{\mathrm{BC}}$ has a special role as the main quantity of interest in text *compression* (as opposed to closed-loop generation): namely, it is the statistical language model's log-perplexity. Figure 16a shows that on TinyStories, in terms of validation set perplexity, the EMA iterates are **always** at least as good as the optimizer iterates. Furthermore, beyond 4 training epochs, when models overfit when the learning rate is decayed to 0. without decay, **models do not overfit and keep improving**, even up to 16 training epochs. We find that a *partial* annealing strategy (decaying linearly from $8 \times 10^{-4}$ to $4 \times 10^{-4}$) produces the best of all models, with a token perplexity of 2.21 (compared to 2.31 without EMA). While the margin of overall improvement for 1 epoch over the larger Wikipedia dataset (Figure 16b) is smaller (9.54 vs. 9.39 token perplexity), the finding that EMA never hurts is upheld.

**Examples of divergent generations.** Below, we provide some examples of argmax completions from these TinyStories language models. This serves as a quick check that (1) the autoregressive generations produced by these language models are qualitatively fluent in a restricted linguistic domain, and (2) minibatch noise-induced fluctuations induce distinct autoregressive generations. The robust benchmarking of end-to-end generation quality presents methodological ambiguities; resolving them is outside the scope of this paper.

**Prompt:** *Once upon a time, in a warm and sunny place, there was a big pit. A little boy*

Continuation from data: *named Tom liked to play near the pit. One day, Tom lost his red ball. He was very sad. Tom asked his friend, Sam, to help him search for the ball. They looked high and low, but they could not find the ball. Tom said, "I think my ball fell into the pit." Sam and Tom went close to the pit. They were scared, but they wanted to find the red ball. They looked into the pit, but it was too dark to see. Tom said, "We must go in and search for my ball." They went into the pit to search. It was dark and scary. They could not find the ball. They tried to get out, but the pit was too deep. Tom and Sam were stuck in the pit. They called for help, but no one could hear them. They were sad and scared, and they never got out of the pit.*

**All models agree:** named Tim and his dog, Spot, liked to play near the pit. They would run, jump, and laugh all day long. One day, Tim and Spot were playing near the pit when they saw a big ball. [...]

**15027-15035:** [...] They both wanted to play with the ball, but they didn't want to share. Tim said, "I want the ball!" Spot barked, "No, I want the ball!" They started to fight over the ball. Tim's mom saw them fighting and said, "Stop fighting! You can both play with the ball if you share." Tim and Spot stopped fighting and started to play together. They had so much fun playing with the ball near the warm pit. From that day on, they always shared their toys and played happily together.

**15036-15037:** [...] They both wanted to play with it. Tim said, "Let's share the ball, Spot!" Spot barked happily, and they both played with the ball together. As they played, a little girl named Sue came by. She saw Tim and Spot sharing the ball and wanted to join them. Tim said, "Yes, you can play with us!" So, Tim, Spot, and Sue played together near the warm pit, and they all had a great time. They learned that sharing is fun and makes everyone happy.

**15038-15039:** [...] They both wanted to play with it. Tim said, "I want the ball!" Spot barked and wagged his tail. They both ran to the ball and started to play. As they played, the ball went into the pit. Tim and Spot looked at each other, not knowing what to do. Then, a friendly frog jumped out of the pit with the ball in its mouth. The frog said, "Here is your ball, Tim and Spot! I saw you playing and wanted to join in too!" Tim and Spot were so happy and thanked the frog. They all played together and had a great day.

**15040-15049:** [...] They both wanted to play with the ball, but they didn't want to share. Tim said, "I want the ball!" Spot barked, "No, I want the ball!" They started to fight over the ball. As they fought, the ball rolled into the pit. Tim and Spot stopped fighting and looked at the pit. They were sad because they couldn't play with the ball anymore. They learned that fighting was not good and they should share their toys.

**15000-15099 (EMA):** [...] Tim said, "Spot, let's play with the ball!" They played with the ball, but it was too big for them. They tried to push it, but it would not move. Then, Tim had an idea. He said, "Spot, let's reverse and push the ball back to where we found it." They pushed the ball back and it started to roll. They were so happy and played with the ball all day long.

Figure 17: Argmax-decoded generations from a sequence of consecutive training checkpoints (15027-15049), seeded by a prefix of examples from the TinyStories dataset. These are the full generations corresponding to Figure 4 *(bottom)*. Note that we do not attempt to evaluate generation quality systematically in this work; we only note that (1) the argmax generations oscillate between semantically distinct modes, and (2) the EMA iterates, aside from having better losses, switch their rollout argmax trajectories (conditioned on in-distribution prompt prefixes) far less frequently.

We close with a few remarks on larger-scale experiments, in light of the flurry of interest in understanding and improving the training of large language models (LLMs).

- **Prior work.** Our preliminary NLP experiments are certainly not the first to note the benefits of averaging language models (Kaddour, 2022; Wortsman et al., 2022; Sanyal et al., 2023; Sandler et al., 2023). In contradistinction to these works, our work contributes a set of *controlled* experiments to isolate the phenomenon of GVA.

- **Iterate averaging at the LLM scale.** Towards understanding considerations which may appear at the scale of our study, it is difficult to perform completely analogous experiments with open-source pretrained models. Model releases which target scientific analysis (e.g. Sellam et al. (2021); Biderman et al. (2023)) only publish a small number of intermediate checkpoints (if at all), so that frequent EMA cannot be emulated from these artifacts.[13] Thus, we encourage interested parties to **explore trajectory stabilization at the scale of frontier LLMs**, and **publish both EMA-filtered and frequently-saved checkpoints** for evaluation and analysis.

- **Preliminary recommendations.** We suggest updating the EMA at every iteration, setting $\gamma = 10^{-4}$ ($\beta = 0.9999$), employing a burn-in that is roughly the same length as the learning rate warmup, and tuning the annealing parameter in the range $0.5 \leq \alpha \leq 1$ (larger $\alpha$ acts like a smaller learning rate). We also note (based on limited, informal tests) that the stabilizing benefits of EMA appear to remain when **finetuning** pretrained LLMs; in this use case, we recommend setting burn-in to $0$, and carefully tuning the annealing parameter based on the finetuning dataset size and number of passes over the data.

## C.3 LINEAR SYSTEMS

In order to generate intuition about the cause and mitigation of GVA, we study a simple linear system as well as a simple 2-piece affine system. We recall the setting of the Linear Quadratic Regulator (LQR) (Kalman, 1960). Let $\mathbf{x}_t \in \mathbb{R}^{d_\mathbf{x}}$ and $\mathbf{u}_t \in \mathbb{R}^{d_\mathbf{u}}$ be state and control vectors and suppose that

$$\mathbf{x}_{t+1} = \mathbf{A}\mathbf{x}_t + \mathbf{B}\mathbf{u}_t + \mathbf{w}_t, \qquad \mathbf{w}_t \sim \mathcal{N}\left(\mathbf{0}, \sigma^2 \mathbf{I}\right), \tag{C.1}$$

where $\mathbf{A} \in \mathbb{R}^{d_\mathbf{x} \times d_\mathbf{x}}$ and $\mathbf{B} \in \mathbb{R}^{d_\mathbf{x} \times d_\mathbf{u}}$. For a horizon $H$, and policy $\pi_\mathbf{K} : \mathbf{x} \mapsto \mathbf{K}\mathbf{x}$ with $\mathbf{K} \in \mathbb{R}^{d_\mathbf{u} \times d_\mathbf{x}}$, the reward is given by

$$J_H(\pi_\mathbf{K}) = \mathbb{E}\left[\sum_{t=0}^{H-1} r(\mathbf{x}_t, \mathbf{u}_t)\right] = -\sum_{t=0}^{H-1} \mathbf{x}_0^\top \left(\left(\mathbf{A} + \mathbf{B}\mathbf{K}\right)^t\right)^\top (\mathbf{Q} + \mathbf{K}\mathbf{R}) \left(\mathbf{A} + \mathbf{B}\mathbf{K}\right)^t \mathbf{x}_0 \tag{C.2}$$

$$r(\mathbf{x}_t, \mathbf{u}_t) = -\mathbf{x}_t^\top \mathbf{Q}\mathbf{x}_t - \mathbf{u}_t^\top \mathbf{R}\mathbf{u}_t,$$

where $\mathbf{Q}, \mathbf{R}$ are positive semi-definite matrices.

**Remark C.1.** In most treatments of LQR, $J_H$ is taken to be positive and thus corresponds to a control *cost* rather than a reward. In order to remain consistent with the rest of the paper, we take $J_H$ to be negative and thus a reward.

In the infinite horizon case, it is known that an optimal policy $\mathbf{K}^\star$ can be found by solving the Discrete Algebraic Riccati Equation (DARE) (Kalman, 1960):

$$\mathbf{K}^\star = -\left(\mathbf{R} + \mathbf{B}^\top \mathbf{S}\mathbf{B}\right)^{-1}(\mathbf{B}\mathbf{S}\mathbf{A}),$$

$$\mathbf{S} = \mathbf{Q} + \mathbf{A}^\top \mathbf{S}\mathbf{A} - \mathbf{A}^\top \mathbf{S}\mathbf{B}\left(\mathbf{R} + \mathbf{B}^\top \mathbf{S}\mathbf{B}\right)^{-1}\mathbf{B}^\top \mathbf{S}\mathbf{A}.$$

In this section, when the system matrices $(\mathbf{A}, \mathbf{B}, \mathbf{Q}, \mathbf{R})$ are clear from context, we reserve $\mathbf{K}^\star$ for this optimal policy. The closed-loop system is called *marginally unstable* if $\|\mathbf{A} + \mathbf{B}\mathbf{K}^\star\|_{\text{op}} = 1$ and *marginally stable* if $\|\mathbf{A} + \mathbf{B}\mathbf{K}^\star\|_{\text{op}} < 1$. Stability is a helpful condition for analysis as it implies that trajectories converge toward zero and thus we do not expect GVA to appear. Marginal stability does not cause convergence, but as we show in Appendix E.2, when the imitator is sufficiently close to the Ricatti policy, we also do not expect to see GVA. In both experiments, we follow a similar pipeline as in the MuJoCo experiments as described in Appendix C.1, with the main difference that we hard-code our expert. We now describe our two experiments in detail. In our experiments, we let $d_\mathbf{x} = d_\mathbf{u} = 2$ for ease of visualization of trajectories.

---

[13]Nor would it be practical to store every training iterate.

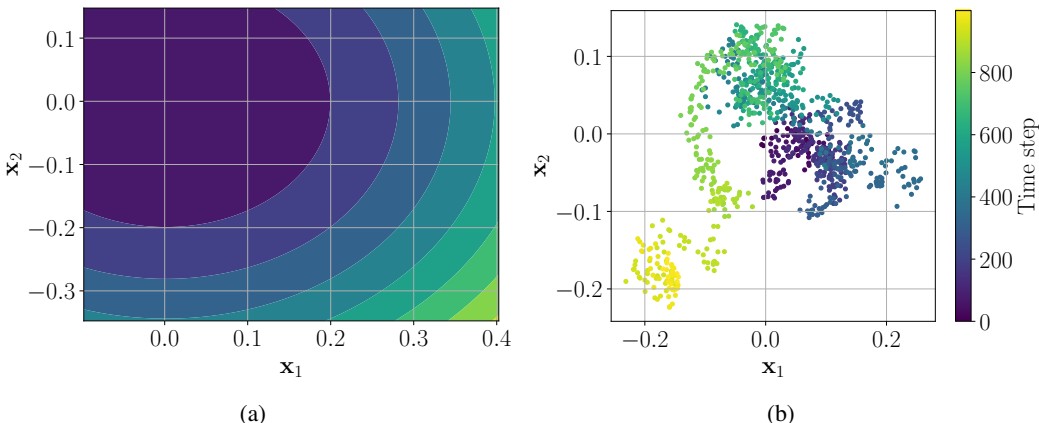

(a)                                                 (b)

Figure 18: Depiction of marginally stable LQR setting. In (a) we show a contour plot of the reward (for fixed action $\mathbf{u}$), which is simply a quadratic and in (b) we show a sample expert trajectory in state space. Each point is the state at a given time step, with the color corresponding to the time.

## C.4 LQR WITH MARGINALLY STABLE DYNAMICS

In our first experiment, we let

$$\mathbf{A} = \begin{bmatrix} 1.0025 & 0 \\ 0 & 1.0025 \end{bmatrix}, \qquad \mathbf{B} = \begin{bmatrix} -0.0043 & -0.0026 \\ -0.0026 & -0.0043 \end{bmatrix}, \qquad \mathbf{Q} = \mathbf{R} = \mathbf{I}.$$

To generate these matrices, we chose a small $\alpha > 0$ and let $\epsilon = \frac{\alpha}{H}$. We then sampled a rotation $\mathbf{O}$ uniformly at random from the orthogonal group. We let $\mathbf{A} = (1 + \epsilon)\mathbf{I}$ and $\mathbf{B} = -\epsilon\mathbf{O}$. Note that this system is clearly marginally stable. The rotation is included to make the learning problem slightly more challenging for the MLP. We let $\mathbf{K}^\star$ be the Riccati policy of this marginally stable system,

$$\mathbf{K} = \begin{bmatrix} 1.3867 & 0.8250 \\ 0.8250 & -1.3867 \end{bmatrix}.$$

Steps (2-5) of the pipeline in Appendix C.1 are identical to the MuJoCo experiments. A visualization of the reward landscape, the expert policy, and a sample expert trajectory can be found in Figure 18. As demonstrated theoretically in Appendix E.2, we expect little to no oscillation in this setting, and indeed we observe this. In particular, we consider two imitator function classes:

- **Linear.** This is simply linear regression with dependent features, where we optimize over the class of functions $\left\{\mathbf{x} \mapsto \mathbf{K}\mathbf{x} | \mathbf{K} \in \mathbb{R}^{d_{\mathbf{x}} \times d_{\mathbf{u}}}\right\}$.

- **MLP.** We also investigate the effect of introducing nonconvexity into the optimization by optimizing over depth 2 neural networks with the dimension of the hidden layer being 32. For optimization, we use a AdamW with a learning rate of .0003 and default parameters. We also use a linear decay with a warmup of 50 steps.

We present the result of this experiment in Figure 19. As expected, we do not see GVA in this setting; indeed, both the linear imitator and the MLP are able to recover the expert to a sufficiently high degree of accuracy so as to obviate the lack of stability, as predicted by Proposition D.2. We also exhibit a scatter plot comparing $\ell_{\mathrm{BC}}$ and $J$ in Figure 20 to emphasize the lack of GVA in this setting.

## C.5 LQR WITH CLIFF

Unsurprisingly, we did not see GVA in the previous experiment. Indeed, we predict GVA to occur when the rollout dynamics function is highly unstable and possibly discontinuous. To demonstrate that this is indeed the case even with extremely simple dynamics, we consider a system motivated by a spring falling off a cliff. Again we suppose that $d_{\mathbf{x}} = d_{\mathbf{u}} = 2$ and think of $\mathbf{x}_1$ as a position

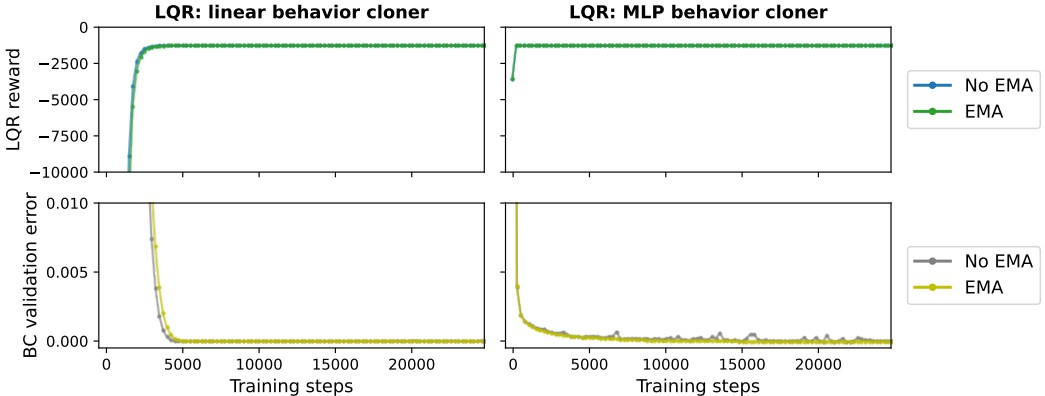

Figure 19: Training curves of imitator in marginally stable LQR setting. We show both linear imitators (left) and MLP imitators (right), comparing the reward curves (top) with the $\ell_{\mathrm{BC}}$ curves on a validation set (bottom). We show the results both of EMA and of no EMA.

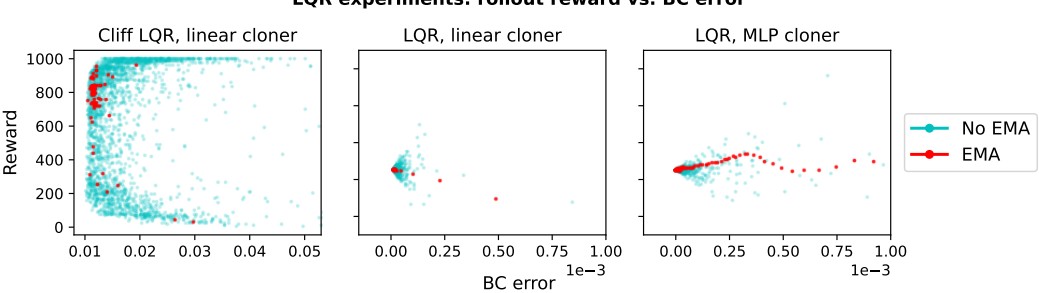

Figure 20: Scatter plots comparing $\ell_{\mathrm{BC}}$ to the reward $J$ at different checkpoints of a single run in the LQR experiments. We show the results both of EMA and of no EMA. We exhibit a linear imitator of LQR with a cliff setting (left) as well as a linear (center) and MLP (right) imitator of the marginally stable LQR setting.

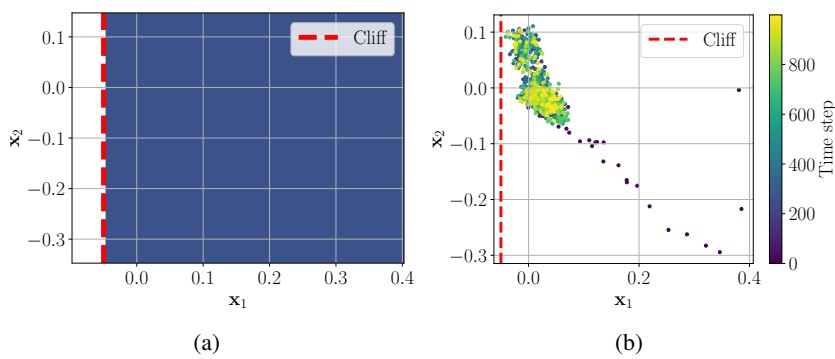

Figure 21: Depiction of the LQR with a cliff setting. In (a) we show a contour plot of the reward landscape; note that the reward is negative infinity to the left of the cliff, marked in red. In (b) we show an example expert trajectory in state space. Each point is the state at a given time step, with the color corresponding to the time. Also marked is the cliff in red.

Figure 22: Training curves of linear imitator in LQR with a cliff. We show scatter plots of both the reward curves (top) and the $\ell_{\mathrm{BC}}$ curves on a validation set (bottom). We show the results both of EMA and of no EMA.

coordinate in a one-dimensional space and $\mathbf{x}_2$ as a velocity coordinate. We let

$$\mathbf{A} = \exp\left(\eta \cdot \begin{bmatrix} 0 & 1 \\ -1 & 0 \end{bmatrix}\right), \qquad \mathbf{B} = \begin{bmatrix} 0 \\ 1 \end{bmatrix}, \qquad \mathbf{Q} = \mathbf{R} = \mathbf{I}.$$

for a time parameter $\eta$ that we set to be $0.1$ in our experiments. As it stands, this system acts as a discrete time approximation of a spring oscillating in one dimension, where the control is such that the learner can only affect the velocity directly (perhaps by applying some small force). To this system, we introduce a 'cliff' parameter $\kappa < 0$ such that if $\mathbf{x}_1 < \kappa$, then the agent 'falls off a cliff' and the episode ends. We modify the reward function by letting

$$r(\mathbf{x}, \mathbf{u}) = \begin{cases} 1 & \text{if } \mathbf{x}_1 > \kappa \\ -\infty & \text{otherwise.} \end{cases}$$

in our experiments, we let $\kappa = -0.05$. For a visualization of the reward, see Figure 21 (a). We again use the Ricatti policy $\mathbf{K}^\star$ (with $\mathbf{Q} = \mathbf{R} = \mathbf{I}$) as the expert. Note that this policy does not take into account the infinite negative reward incurred by falling off the cliff and thus is not necessarily optimal in the above setting.

Here we only train a linear policy and examine the effect that EMA has on training. We use a large, constant learning rate of 0.3 and apply EMA as a post hoc filter, making this a direct MDP analogue of the situation considered in Proposition D.5 in Appendix D. We display the results of this experiment in Figure 22. Note that GVA clearly occurs in this setting, as predicted by Proposition D.5, although we observe that the . Furthermore, we see that EMA is able to mitigate the oscillations and allow the agent to converge to an at least mediocre policy. We emphasize that the expert policy here is suboptimal because the Riccati policy does not take into account the infinite negative reward incurred by falling off the cliff and thus it is not surprising that the EMA'd imitator does not perform optimally. We also include a scatater plot of $\ell_{\mathrm{BC}}$ vs. $J$ in Figure 20 to emphasize the fact that, even though there remains some small oscillation in $\ell_{\mathrm{BC}}$ due to the large constant learning rate, GVA is still very much observable in the complete lack of relationship between the two quantities.

# D   THEORETICAL ANALYSIS OF TOY EXAMPLES: DETAILS AND ADDITIONAL RESULTS

In this section, we provide details and formal statements for the theoretical vignettes in Section 3 and Section 4, as well as additional results.

We begin in Appendix D.1, which studies error amplification in linear dynamical systems. Proposition D.1, which is a formal version of Proposition 3.1, provides a linear example of extreme sensitivity of rollout reward to parameters despite the lack of sensitivity in the training loss. We proceed by recalling that for (marginally) stable linear systems, this is not an issue, which is in line with our empirical results in Appendix C.4. We then shift our focus to exploring the effect of EMA in several toy examples.

- In Appendix D.2, we introduce and analyze a variant of the cliff loss example studied empirically in Appendix C.5, and demonstrate in Proposition D.3 that, if iterates are distributed as Gaussians, then the cliff loss can be characterized in terms of the BC loss. For the sake of simplicty, we assume Gaussianity in our iterates when conducting computations.

- Next, in Appendix D.2.1, we consider the effect of SGD on square loss and its interaction with EMA; Proposition D.5 demonstrates that the benefits of EMA can be heavily dependent on the choice of learning rate. We then use this result to give further consequences for the cliff loss. This constitutes a formal version of Proposition 4.1. In Proposition D.7, we provide a similar analysis in the continuous time limit for a variety of more complicated learning rate schedules, with the caveat that this result is restricted to the setting where EMA is started after a sufficiently large warmup period so as to saturate the population gradient.

- Finally, in Appendix D.3, we consider the extent to which convex theory explains the empirical benefits of EMA. We begin by surveying a number of alternative iterate averaging schemes in Appendix D.3.1 and demonstrate in Theorem 2 that EMA is provably *suboptimal* for stochastic convex optimization when the *EMA parameter* is chosen with the scaling that we find necessary in practice. These results apply to the *step size* choices which are optimal for the convex optimization tasks. To summarize, while the large learning rates we find empirically effective cannot be explained by the theory, conditional on using these mathematically suboptimal rates, stochastic convex optimization does provide useful intuition.

All proofs are deferred to Appendix E.

## D.1 ERROR AMPLIFICATION IN LINEAR DYNAMICAL SYSTEMS

In this section, we study the error amplification and the GVA phenomenon in linear dynamical systems. We begin with a formal version of Proposition 3.1, which demonstrates that, even with a Lipschitz dynamical system, the imitation loss may have to quite small in order to guarantee that the rollout reward is close to optimal. This is the origin of GVA: small perturbations in parameters that do not appear in BC the loss landscape lead to enormous changes in the rollout reward.

**Proposition D.1** (GVA in linear dynamical systems). *For any $\epsilon > 0$ and $d \in \mathbb{N}$, there exists a linear function $f : \mathbb{R}^d \times \mathbb{R}^d \to \mathbb{R}^d$ and linear policy $\pi_{\boldsymbol{\theta}^\star} : \mathbb{R}^d \to \mathbb{R}^d$ such that the following statements hold:*

(a) *There exists an MDP $\mathcal{M}$ with* deterministic *transitions such that $\mathbf{x}_{h+1} = f(\mathbf{x}_h, \mathbf{u}_h)$ for all $t$.*

(b) *The functions $\mathbf{x} \mapsto f(\mathbf{x}, \pi_{\boldsymbol{\theta}^\star}(\mathbf{x}))$, $(\mathbf{x}, \mathbf{u}) \mapsto f(\mathbf{x}, \mathbf{u})$, and $\mathbf{x} \mapsto \pi_{\boldsymbol{\theta}^\star}(\mathbf{x})$ are globally $1 - \epsilon$, $\sqrt{1 + c^2}$, and $\epsilon/c$ Lipschitz, respectively.*

(c) *Let the reward function $r(\mathbf{x}, \mathbf{u}) = -\|\mathbf{x}\|^2$ be quadratic, so that $J_H(\pi_{\boldsymbol{\theta}}) = -\mathbb{E}\left[\sum_{t=1}^H \|\mathbf{x}_t\|^2\right]$, and let $\mathcal{B}_{\epsilon'}$ denote the set of $\epsilon'$-Lipschitz linear functions $\Delta : \mathbb{R}^d \to \mathbb{R}^d$ satisfying $\Delta(\mathbf{0}) = \mathbf{0}$. If $\delta = c\epsilon' - \epsilon$. Then,*

$$\sup_{\Delta \in \mathcal{B}_{\epsilon'}} \{J_H(\pi_{\boldsymbol{\theta}^\star}) - J_H(\pi_{\boldsymbol{\theta}^\star} + \Delta)\} \geq Cd \cdot \begin{cases} \min\left(H, |\frac{1}{\delta}|\right), & \delta \leq 0, \\ He^{CH\delta}, & \text{otherwise} \end{cases}$$

*for a universal constant $C > 0$.*

(d) *Let imitation loss be defined as*

$$\ell_{\mathrm{BC}}(\pi_{\boldsymbol{\theta}}) = \sum_{t=1}^H \|\pi_{\boldsymbol{\theta}}(\mathbf{x}_t) - \pi_{\boldsymbol{\theta}^\star}(\mathbf{x}_t)\|^2.$$

*Then*

$$\sup_{\Delta \in \mathcal{B}_{\epsilon'}} \ell_{\mathrm{BC}}(\pi_{\boldsymbol{\theta}}) \leq H \left\| \mathbf{x}_1 \right\|^2 \cdot (\epsilon')^2.$$

*(e) Moreover, there exists a subset $\tilde{\mathcal{B}}_\delta \subset \mathcal{B}_\delta$ such that, if $\epsilon$ is sufficiently small, then for $\Delta \in \tilde{\mathcal{B}}'_\delta$, $\ell_{\mathrm{BC}}(\pi_{\boldsymbol{\theta}^\star} + \Delta) \geq H\delta^2$ and yet, $J_H(\pi_{\boldsymbol{\theta}^\star}) - J_H(\pi_{\boldsymbol{\theta}^\star} + \Delta) \leq 0$.*

Note that Proposition D.1 provides an existence result, providing a simple example where GVA may occur. In effect, (d) ensures that small perturbations around the expert $\pi_{\boldsymbol{\theta}^\star}$ result in small changes to the next step prediction loss $\ell_{\mathrm{BC}}$, while (c) shows that such small changes lead to exponential blowup in the rollout reward of the policies $J_H$; statements (a) and (b) demonstrate that such a phenomenon can even occur in otherwise well-behaved (in the sense of being Lipschitz) systems.

We now complement Proposition D.1 with a negative example, showing that if a linear system is sufficiently stable, then GVA does not occur. For more intuition and a number of references relevant to these linear examples, see Simchowitz et al. (2020); Hazan & Singh (2022). To state the result, recall the dynamics given in Eq. (C.1), as well as the LQR loss given in Eq. (C.2). We consider linear policies in which there is some $\mathbf{K} \in \mathbb{R}^{\mathsf{d_x} \times \mathsf{d_u}}$ such that $\mathbf{u}_t = \pi_{\boldsymbol{\theta}^\star}(\mathbf{x}_t) = K\mathbf{x}_t$, which implies that the expected dynamics are given by a linear function: $\mathbb{E}\left[\mathbf{x}_H\right] = (\mathbf{A} + \mathbf{B}K)^H \mathbf{x}_0$. We have the following result.

**Proposition D.2** (GVA does not occur in sufficiently stable linear systems). *Consider the linear dynamical system in Eq. (C.1) and suppose that the expert policy given by $\pi_{\boldsymbol{\theta}^\star}(\mathbf{x}_t) = \mathbf{K}^\star \mathbf{x}_t$ is such that $\left\| \mathbf{A} + \mathbf{B}\mathbf{K} \right\|_{\mathrm{op}} \leq 1$, i.e., the closed-loop dynamics are marginally stable. Suppose that $\widehat{\mathbf{K}}$ is an imitator policy trained so that $\left\| \mathbf{K} - \widehat{\mathbf{K}} \right\|_{\mathrm{op}} \leq \frac{\epsilon}{H \|\mathbf{B}\|_{\mathrm{op}}}$ for $\epsilon \leq 1$ and $\left\| \mathbf{Q} + \widehat{\mathbf{K}}\mathbf{R} \right\|_{\mathrm{op}} \leq C$. Then*

$$J_H(\pi_{\widehat{\mathbf{K}}}) - \inf_{\left\| \mathbf{K} - \widehat{\mathbf{K}} \right\|_{\mathrm{op}} \leq \frac{\epsilon}{H\|\mathbf{B}\|_{\mathrm{op}}}} J_H(\pi_{\mathbf{K}}) \leq 100 \left( CH^2 + H \left\| \mathbf{R} \right\|_{\mathrm{op}} \right) \left\| \mathbf{x}_0 \right\|^2 \epsilon.$$

*If $\ell_{\mathrm{BC}}(\pi_{\mathbf{K}}) = \mathbb{E}\left[ \sum_{h=1}^H \left\| (\mathbf{K} - \mathbf{K}^\star) \mathbf{x}_t \right\|^2 \right]$, then*

$$\sup_{\left\| \mathbf{K} - \widehat{\mathbf{K}} \right\|_{\mathrm{op}} \leq \epsilon} \ell_{\mathrm{BC}}(\pi_{\mathbf{K}}) \leq CH \left( \left\| \mathbf{x}_0 \right\|^2 + \mathsf{d_x} \right) \epsilon^2.$$

*Furthermore, if there exists some $\delta > 0$ such that $\left\| \mathbf{A} + \mathbf{B}\mathbf{K} \right\|_{\mathrm{op}} \leq 1 - \delta$, then as soon as $\left\| \mathbf{K} - \widehat{\mathbf{K}} \right\|_{\mathrm{op}} \leq \frac{\delta}{2\|\mathbf{B}\|_{\mathrm{op}}}$, we have*

$$J_H(\pi_{\widehat{\mathbf{K}}}) - \inf_{\left\| \mathbf{K} - \mathbf{k} \right\|_{\mathrm{op}} \leq \frac{\delta}{2\|\mathbf{B}\|_{\mathrm{op}}}} J_H(\pi_{\mathbf{K}}) \leq 100 \left( CH^2 + H \left\| \mathbf{R} \right\|_{\mathrm{op}} \right) \left\| \mathbf{x}_0 \right\|^2 \epsilon.$$

*Note that in both cases, the rollout error $J_H$ does not grow significantly more quickly than $\ell_{\mathrm{BC}}$, in contrast to Proposition D.1, suggesting that GVA will not occur.*

Note that in the data regimes we consider empirically for the LQR setting, the assumption that $\left\| \widehat{\mathbf{K}} - \mathbf{K} \right\|_{\mathrm{op}} \lesssim \frac{1}{H}$ is reasonable, and SGD will converge relatively quickly to this regime (Nesterov et al., 2018). Thus, for this setting, we do not expect GVA to persist in training, even for marginally stable systems. It follows that the conclusion of this proposition is in line with our empirical results for LQR Appendix C.4.

## D.2 BENEFITS OF EMA FOR THE CLIFF LOSS

In this section, we state a formal version of Proposition 4.1, which shows that EMA can reduce the variance of SGD for the cliff loss in Section 4.3. Recall that we consider a simple parameter estimation task, where we minimize the "behavior cloning" loss

$$\ell_{\mathrm{BC}}(\boldsymbol{\theta}) = \frac{1}{2} \|\boldsymbol{\theta} - \boldsymbol{\mu}\|^2, \tag{D.1}$$

for some mean vector $\boldsymbol{\mu} \in \mathbb{R}^d$, while the "reward" function is given by a cliff loss:

$$J(\boldsymbol{\theta}) = \begin{cases} -\|\boldsymbol{\theta} - \boldsymbol{\mu}\|^2 & \|\boldsymbol{\theta} - \boldsymbol{\mu}\| \leq \epsilon \\ -C & \text{otherwise} \end{cases}, \tag{D.2}$$

where $1 \geq \epsilon > 0$ is a fixed scale parameter. This cliff loss is a simplified version of the "LQR-with-a-cliff" dynamical system given in Appendix C.5, corresponding to trajectories of length $H = 1$, which simplifies calculations. We begin by showing that despite its simplicity, this setting is nonetheless nontrivial: when $\boldsymbol{\theta}$ is drawn from a Gaussian distribution, the expected cliff loss indeed exhibits cliff-like behavior in the regime where $\mathbb{E}[\ell_{\mathrm{BC}}(\boldsymbol{\theta})] = \Theta(\epsilon^2)$. To state the result, let $\boldsymbol{\theta}^\star = \boldsymbol{\mu}$ denote the optimal parameter. Note that the Gaussian assumption is used mainly for the sake of simplicity, as it allows for exact calculations; in addition in the continuous time limit of SGD as applied to $\ell_{\mathrm{BC}}$, the iterates approach a Gaussian process (Mandt et al., 2017), a fact which we use in the sequel to apply this result.

**Proposition D.3.** *Let $\boldsymbol{\theta} \sim \mathcal{N}(\boldsymbol{\mu}', \Sigma)$ be a Gaussian random vector in $\mathbb{R}^d$. Then, there exists universal constants $c_1, c_2, c_3 > 0$ such that the following hold. First,*

$$J(\boldsymbol{\theta}^\star) - \mathbb{E}[J(\boldsymbol{\theta})] \geq \frac{C}{2}, \quad \text{whenever} \quad \mathbb{E}[\ell_{\mathrm{BC}}(\boldsymbol{\theta})] \geq c_1 \epsilon^2. \tag{D.3}$$

*On the other hand, suppose that $\mathbb{E}[\ell_{\mathrm{BC}}(\boldsymbol{\theta})] \leq \frac{\epsilon^2}{8}$. Then,*

$$J(\boldsymbol{\theta}^\star) - \mathbb{E}[J(\boldsymbol{\theta})] \leq 2\mathbb{E}[\ell_{\mathrm{BC}}(\boldsymbol{\theta})] + C \cdot c_2 \exp\left(-\frac{c_3 \epsilon^2}{2\mathbb{E}[\ell_{\mathrm{BC}}(\boldsymbol{\theta})]}\right).$$

*In particular, if $\epsilon^2 \geq c_3^{-1}\mathbb{E}[\ell_{\mathrm{BC}}(\boldsymbol{\theta})]\log(c_2 C/\mathbb{E}[\ell_{\mathrm{BC}}(\boldsymbol{\theta})])$, then,*

$$J(\boldsymbol{\theta}^\star) - \mathbb{E}[J(\boldsymbol{\theta})] \leq 3\mathbb{E}[\ell_{\mathrm{BC}}(\boldsymbol{\theta})].$$

In the sequel, we show (via Proposition D.3) how small improvements in BC loss can translate to major improvements in the cliff loss, and how EMA can induce these improvements.

We shall also show how the SGD iterates will exhibit large probabilistic fluctuations in their cliff loss, *even if* the expected cliff loss is large. To do so, we require the following.

**Lemma D.4.** *Let $\mathbf{z}_1 \sim \mathcal{N}(0, \sigma^2)$. Then,*

$$\mathbb{P}[|\mathbf{z}_1| \leq \sigma\epsilon] \geq \epsilon\sqrt{\frac{2}{\pi}} e^{-\epsilon^2/2}.$$

*In particular, for $\epsilon \leq 1$, $\mathbb{P}[|\mathbf{z}_1| \leq \sigma\epsilon] \geq \epsilon/3$.*

*Proof.* By rescaling, we can assume $\sigma = 1$. We then bound

$$\mathbb{P}[|\mathbf{z}_1| \leq \epsilon] = \int_{-\epsilon}^{\epsilon} \frac{e^{-u^2/2}}{\sqrt{2\pi}} du \geq 2\epsilon\sqrt{\frac{1}{2\pi}} e^{-\epsilon^2/2} = \epsilon\sqrt{\frac{2}{\pi}} e^{-\epsilon^2/2}.$$

$\square$

### D.2.1 Analysis of stochastic gradient descent on square loss and cliff Loss

In this section, we study the benefits of EMA for the square-loss $\ell_{\mathrm{BC}}$ defined in (D.1) just above. This objective function is equivalent to estimation of the mean parameter $\boldsymbol{\mu}$. We demonstrate mathematically that in this problem, iterate averaging effectively decreases the learning rate. Because rapid $O(1/t)$ learning decay is optimal, we find that for larger learning rates, EMA yields a benefit. Note that a similar qualitative observation in a more restricted setting was made in Sandler et al. (2023). While this setting is too simple to observe interesting fractal behavior that we observe in GVA on more sophisticated imitation learning tasks , it does provide some theoretical intuition for the empirical finding that aggressively decaying the learning rate and iterate averaging result in similar qualitative behavior.

We consider stochastic gradient updates on the $\ell_{\mathrm{BC}}$ defined in (D.1) defined as follows,

$$\mathbf{y}^{(t)} = \boldsymbol{\mu} + \mathbf{w}^{(t)}, \quad \boldsymbol{\theta}^{(0)} = \mathbf{0}, \quad \boldsymbol{\theta}^{(t+1)} = \boldsymbol{\theta}^{(t)} - \eta_t \mathbf{y}_t$$
$$\tilde{\boldsymbol{\theta}}_\gamma^{(t)} = \gamma_t \boldsymbol{\theta}^{(t)} + (1 - \gamma_t)\tilde{\boldsymbol{\theta}}_\gamma^{(t-1)}, \quad \bar{\boldsymbol{\theta}}^{(0)} = \tilde{\boldsymbol{\theta}}_\gamma^{(0)}, \tag{D.4}$$

where $\mathbf{w}^{(t)}$ is a scaled, isotropic Gaussian and $\eta_t, \gamma_t > 0$ are the learning rate and EMA parameters, respectively. Because $\mathbf{w}^{(t)}$ is isotropic, the problem tensorizes across coordinates. Hence, the following result considers only the dimension one case.

**Proposition D.5.** *Consider the process in* (D.4) *for dimension* $d = 1$, *with constant learning rate and EMA parameters* $\gamma_t \equiv \gamma$ *and* $\eta_t \equiv \eta$. *Let* $b = |\boldsymbol{\theta}^{(0)} - \boldsymbol{\mu}|$, *and* $\mathbf{w}_t \overset{\text{i.i.d}}{\sim} \mathcal{N}(0, \sigma^2)$. *Then*

$$2\mathbb{E}[\ell_{\mathrm{BC}}(\tilde{\boldsymbol{\theta}}_\gamma^{(t)})] = \mathbb{E}[(\tilde{\boldsymbol{\theta}}_\gamma^{(t)} - \boldsymbol{\mu})^2] \le 2b^2(1-\gamma)^{2T} + \begin{cases} 4\sigma^2\eta + 4b^2(1-\eta)^{2t} & \gamma \ge 2\eta, \\ 16\sigma^2\eta + 32b^2(1-\frac{\eta}{4})^{2t} & \frac{\gamma}{2} \le \eta \le 2\gamma, \\ 4\sigma^2\gamma + 4b^2\frac{\gamma^2}{\eta^2}(1-\gamma)^{2t} & \eta \ge 2\gamma. \end{cases}$$

*Similarly, the BC loss is lower bounded as*

$$2\mathbb{E}[\ell_{\mathrm{BC}}(\tilde{\boldsymbol{\theta}}_\gamma^{(t)})] = \mathbb{E}[(\tilde{\boldsymbol{\delta}}_\gamma^{(T)})^2] \ge b^2(1-\gamma)^{2T} + \frac{1}{4}\left(\begin{cases} \sigma^2\eta + b^2(1-\eta)^{2(T-1)} & \gamma \ge \eta \\ \sigma^2\gamma + b^2\frac{\gamma^2}{\eta^2}(1-\gamma)^{2(T-1)} & \eta \ge \gamma \end{cases}\right),$$

*Moreover, in the "no-EMA" setting in which* $\gamma = 1$, *we have*

$$2\mathbb{E}[\ell_{\mathrm{BC}}(\boldsymbol{\theta}^{(t)})] = \eta\mathbb{E}[(\boldsymbol{\theta}^{(t)} - \boldsymbol{\mu})^2] = \sigma^2\left(\frac{1 - ((1-\eta)^{2t})}{2 - \eta}\right) + b^2(1-\eta)^{2t}.$$

To understand when this result reveals benefits of EMA, let us first consider the regime where $\eta, \gamma \gg 1/t$. Here, the $b^2$ term capturing transient dependence on initial condition can be neglected. We then observe that, as long as $\gamma \gtrsim \eta$, EMA with parameter $\gamma$ increases the BC loss by at most a constant factor relative to no EMA. On the other hand, when $\gamma \ll \eta$, the BC loss $\mathbb{E}[\ell_{\mathrm{BC}}(\tilde{\boldsymbol{\theta}}_\gamma^{(t)})]$ for EMA scales linearly in $\gamma$, while the no-EMA BC loss $\mathbb{E}[\ell_{\mathrm{BC}}(\boldsymbol{\theta}^{(t)})]$ is linear in $\eta \gg \gamma$. In other words, when the step size is large, a small EMA parameter significantly aids variance reduction. This is intuitively clear, as smaller EMA parameters correspond to averages over longer windows.

**Consequences for the cliff loss.** By combining Proposition D.5 with Proposition D.3 and Lemma D.4, we now state that EMA is beneficial for the cliff loss setting considered in Appendix D.2 discrete-time SGD, complementing the results for continuous-time SGD in the prequel. This amounts to a discrete time analogue of Proposition E.3.

**Proposition D.6.** *Consider the cliff loss* $J$ *with parameter* $\epsilon > 0$, *consider the iterates produced in Proposition D.5 with* $\sigma^2 = 1$ *and* $b \le 1$. *Then, there exists constants* $c_1, c_2, \cdots > 0$ *such that, if the step size* $\eta$ *and EMA parameter* $\gamma$ *satisfy*

$$\eta \ge c_1\epsilon^2 \ge c_2\epsilon^2 \ge \gamma, \quad (1-\gamma)^{2T} \le \gamma,$$

*then*

*(a) It holds that*

$$J(\boldsymbol{\theta}^\star) - \mathbb{E}[J(\boldsymbol{\theta}^{(T)})] \ge \frac{C}{2}, \quad yet \quad J(\boldsymbol{\theta}^\star) - \mathbb{E}[J(\tilde{\boldsymbol{\theta}}_\gamma^{(T)})] \le c_3\left(\gamma + Ce^{-c_4\epsilon^2/\gamma}\right).$$

*(b) Moreover,* $c_5\eta \le \mathbb{E}[\ell_{\mathrm{BC}}(\boldsymbol{\theta}^{(t)})] \le c_6\eta$ *and* $c_5\gamma \le \mathbb{E}[\ell_{\mathrm{BC}}(\tilde{\boldsymbol{\theta}}_\gamma^{(t)})] \le c_6\gamma$

*(c) If* $\epsilon^2 \ge c_7\gamma\log(C/\gamma)$, *then it holds that*

$$J(\boldsymbol{\theta}^\star) - \mathbb{E}[J(\boldsymbol{\theta}^{(T)})] \ge \frac{C}{2}, \quad yet \quad J(\boldsymbol{\theta}^\star) - \mathbb{E}[J(\tilde{\boldsymbol{\theta}}_\gamma^{(T)})] \le c_8\gamma.$$

*(d) Lastly, even though* $J(\boldsymbol{\theta}^\star) - \mathbb{E}[J(\boldsymbol{\theta}^{(T)})] \ge \frac{C}{2}$, *we have that*

$$\mathbb{P}[J(\boldsymbol{\theta}^\star) - \mathbb{E}[J(\boldsymbol{\theta}^{(T)})] \le \gamma] \ge c_9\gamma/\eta.$$

This proposition reveals that an *logarithmic* difference in the magnitude of $\gamma$ and $\eta$ can lead to $\Omega(C)$-differences in cliff loss. Since $C$ is an arbitrarily large problem parameter, and since $\gamma$ and $\eta$ are proportional to the magnitudes of the respective BC losses due to Proposition D.5, this shows how minor differences in BC loss due to EMA translate into substantial differences in evaluation performance.

Despite these separations in *expected rollout reward*, part (d) of the proposition states that with probability $\Omega(\gamma/\eta)$, the SGD iterate has performance comparable to that of the EMA'd iterate. This reproduces the a second feature of GVA: that test rollout performance of the SGD iterate is not uniformly suboptimal, and some SGD iterates can indeed have loss comparable to that of the EMA'd weights.

### D.2.2   EMA FOR CLIFF LOSS IN THE CONTINUOUS LIMIT

In what follows we analyze the effect of EMA on SGD for the cliff loss. To simplify calculations, we analyze SGD in the continuous-time limit (Mandt et al., 2017; Malladi et al., 2022; Busbridge et al., 2023). In particular, we consider the limit as the learning rate tends to zero at an appropriate rate such that the iterate trajectory becomes a continuous semi-martingale, following e.g. Busbridge et al. (2023); see Appendix B for further references that consider similar limits.

Formally, as in Section 4, we consider a fixed sequence of positive learning rates $(\eta_t)_{k \geq 0}$, as well as an adaptive sequence of (potentially stochastic) estimates $(g^{(t)})_{k \geq 0} \subset \mathbb{R}^d$ of the gradient of $\ell_{\mathrm{BC}}(\boldsymbol{\theta})$. The discrete time iterates $\boldsymbol{\theta}^{(t)} \in \mathbb{R}^d$ are defined recursively, with the initial iterate $\boldsymbol{\theta}^{(0)}$ fixed and subsequent iterates given by

$$\boldsymbol{\theta}^{(t+1)} = \boldsymbol{\theta}^{(t)} - \eta_t \cdot g^{(t)}.$$

Our primary objective of focus is Exponential Moving Average (EMA) of the iterates $\boldsymbol{\theta}^{(t)}$, defined recursively by $\widetilde{\boldsymbol{\theta}}_\gamma^{(0)} = \boldsymbol{\theta}^{(0)}$ and

$$\widetilde{\boldsymbol{\theta}}_\gamma^{(t+1)} = (1 - \gamma_t) \cdot \widetilde{\boldsymbol{\theta}}_\gamma^{(t)} + \gamma_t \cdot \boldsymbol{\theta}^{(t+1)}, \tag{D.5}$$

where $(\gamma_t)_{k \geq 0} \subset [0, 1]$ is a given, non-adaptive sequence of parameters.

**Continuous time limit of SGD.**   To study the continuous time limit of the process above, we take $\eta_t \downarrow 0$ and $K \uparrow \infty$ at a fixed rate such that $\lim_{K \to \infty} \sum_{k=0}^{K} \eta_t \in (0, \infty)$; concrete choices for the sequence $\eta_t$ and stochastic gradient process $g^{(t)}$ are given in the sequel. We then abuse notation by letting $\boldsymbol{\theta}^{(t)}, \widetilde{\boldsymbol{\theta}}_\gamma^{(t)}, \gamma_t$ denote the continuous time analogues of $\boldsymbol{\theta}^{(t)}, \widetilde{\boldsymbol{\theta}}_\gamma^{(t)}$, and $\gamma_t$. With this limit, $\widetilde{\boldsymbol{\theta}}_\gamma^{(t)}$ is the solution to the following differential equation:

$$\mathrm{d}\widetilde{\boldsymbol{\theta}}_\gamma^{(t)} = \gamma_t \cdot \left( \boldsymbol{\theta}^{(t)} - \widetilde{\boldsymbol{\theta}}_\gamma^{(t)} \right) \mathrm{d}t. \tag{D.6}$$

In what follows, we always suppose that a unique solution to (D.6) exists; this is always true by the Picard-Lindelöf theorem (Lindelöf, 1894) if $\gamma_t$ are uniformly bounded and both $\gamma_t$, and $\boldsymbol{\theta}^{(t)}$ are continuous in $t$.[14] In the case where $g_k$ is a stochastic gradient with mean the population gradient and finite second moment of its Euclidean norm, $\eta_t = \eta$ is constant, and $\beta_k = \beta$ is constant, Busbridge et al. (2023, Theorem D.1) shows that as $\eta \downarrow 0$, with the correct scaling, $\tilde{\theta}_\gamma^{(t)}$ is the correct limit of the $\widetilde{\boldsymbol{\theta}}_\gamma^{(t+1)}$. Similar limits with adaptive gradient updates can be found in Busbridge et al. (2023); Malladi et al. (2022). In lieu of re-proving such limits here, we appeal to these results and instead consider a fixed process $\boldsymbol{\theta}^{(t)}$ and compare the behavior of the EMA process $\widetilde{\boldsymbol{\theta}}_\gamma^{(t)}$ to the vanilla SGD process $\boldsymbol{\theta}^{(t)}$ in order to provide intuition as to the effect of EMA on the iterates.

**Benefits of EMA for continuous-time SGD after saturation.**   In practice, EMA is often only applied to the tail iterates of SGD, i.e., after some amount of warmup. If the warmup is sufficiently large, then the iterates may already have approximately found a population-level stationary point, and thus the observed gradients are non-zero only due to the stochasticity therein. Our most basic

---

[14]For background on terminology and results from stochastic calculus, we refer the reader to the excellent exposition of Le Gall (2016).

result, Proposition D.7, abstracts this limit as a driftless stochastic process. More precisely, we assume that

$$\boldsymbol{\theta}^{(t)} = \int_0^t \eta_s \mathrm{d}B_s, \tag{D.7}$$

where $B_s$ is the standard Brownian motion in $\mathbb{R}^d$, which corresonds to the limit when $\mathbb{E}\left[g^{(t)}\right] = 0$. In this simpler setting, we can examine the benefits of EMA with different schedules of learning rates.

**Proposition D.7.** *Suppose that $\eta_t$ is chosen via one of the following schedules:*

*(i) $\eta_t = \eta$ for all $t$ (constant learning rate);*

*(ii) $\eta_t = \eta(1+t)^{-\frac{1}{2}}$ (inverse square root schedule);*

*(iii) $\eta_t = \eta\left(1+t\right)^{-1}$ (inverse schedule);*

*(iv) $\eta_t = 1 - \frac{s}{t}$ (linear decay schedule).*

*Let $\boldsymbol{\theta}^{(t)}$ be as in Eq. (D.7) and $\widetilde{\boldsymbol{\theta}}_\gamma^{(t)}$ as in Eq. (D.6) for fixed $\gamma$. Let the behavior cloning loss $\ell_{\mathrm{BC}}$ and the reward function $J$ be as in Eq. (D.1) and Eq. (D.2) for parameters $C \gg \epsilon > 0$ and $\boldsymbol{\mu} = \mathbf{0}$. If $\boldsymbol{\theta}^{(0)} = 0$, then there exists an $\eta > 0$ sucht that for all sufficiently large $t$, there is some $\gamma > 0$ such that*

$$J(\boldsymbol{\theta}^\star) - \mathbb{E}\left[J\left(\boldsymbol{\theta}^{(t)}\right)\right] \geq \frac{C}{2}, \quad yet \quad J(\boldsymbol{\theta}^\star) - \mathbb{E}\left[J\left(\widetilde{\boldsymbol{\theta}}_\gamma^{(t)}\right)\right] \leq 2\epsilon \ll \frac{C}{2}.$$

*In addition, for the inverse learning rate, we may choose this $\eta$ such that $\max\left(\mathbb{E}\left[\ell_{\mathrm{BC}}\left(\theta^{(t)}\right)\right], \mathbb{E}\left[\ell_{\mathrm{BC}}\left(\tilde{\theta}_\gamma^{(t)}\right)\right]\right) = \mathcal{O}\left(\epsilon\right).$*

This shows that in the continuous-time limit, EMA can substantially improve regret relative to vanilla SGD. While the three learning rate settings above are obviously not exhaustive, they provide a number of classic examples. Constant learning rates are frequently analyzed in theory due to their simplicity, while square root learning rate are typically used in convex optimization to attain the optimal rate of convergence. Linear decay learning rates are common in deep learning and many of our experiments use them. Thus we see that in many natural settings, iterate averaging can help alleviate EMA. In order to broaden the applicability of the result, we also consider an analogue of Proposition D.5 in continuous time, which is similar to Proposition D.7 but allows for drift and thus does not assume we begin EMA in a stationary regime. We state this result in Appendix E.5.2.

## D.3 FOR OPTIMAL CONVEX LEARNING RATES, EMA DOES NOT HELP

In both Propositions E.3 and D.7 we saw the benefits of EMA in convex settings. We now consider several ways in which the convex theory does *not* predict improvement due to EMA, which complements the empirical results in Section 4. We begin by surveying a number of iterate averaging schemes from the literature as applied to stochastic convex optimization in Appendix D.3.1. Then, in Appendix D.3.2 we prove that EMA with the parameters that we find work empirically (see Appendix C) is *provably suboptimal for convex optimization*. This is attributed to the fact that aggressive learning rate decay – far more than is desirable for neural network training – is optimal for convex optimization.

### D.3.1 COMPARISON OF EFFECTS OF ITERATE AVERAGING SCHEMES

In Appendix D.2 we showed the potential benefits of EMA when there exists a discrepancy between the training loss and the reward function. In such settings, both the first and second moments of the distance between the returned parameter and the optimum are relevant to the expected reward. In this section, we survey some prior work and observe that the benefits are often less clear when such a discrepancy does not exist. To do this, we compare various existing iterate averaging schemes and show that some versions of EMA are minimax optimal, while others are not. We focus our discussion on strongly convex but non-smooth stochastic convex optimization. Specifically, we let $\boldsymbol{\theta}_t \in \mathbb{R}^d$,

and consider optimizing a function $F(\boldsymbol{\theta}) : \mathbb{R}^d \to \mathbb{R}$ on a convex domain $\mathcal{K} \subset \mathbb{R}^d$. We assume that $F$ is $\mu$-strongly convex and $L$ Lipschitz w.r.t. the Euclidean norm, and that $\mathrm{diam}(\mathcal{K}) \leq D$. Lastly, we assume that gradients are given according to the following oracle:

**Definition D.1** (Subgradient Oracle). We assume that there exists an oracle $\mathcal{G}(\boldsymbol{\theta})$ and a $G \geq L$ which return a random vector **g** such that (a) $\|\mathbf{g}\| \leq G$ and (b) $\mathbb{E}_{\mathcal{G}(\boldsymbol{\theta})}[\mathbf{g}] \in \partial F(\boldsymbol{\theta})$, where $\partial F(\cdot)$ denotes the convex subgradient. We say $\mathcal{G}(\boldsymbol{\theta})$ is a deterministic oracle if it is deterministic. In this case, we can always take $G = L$.

We consider stochastic optimization algorithms of the form

$$\boldsymbol{\theta}^{(t+1)} = \Pi_\mathcal{K}\left(\boldsymbol{\theta}^{(t)} - \eta_t \mathbf{g}^{(t)}\right), \quad \mathbf{g} \sim \mathcal{G}(\boldsymbol{\theta}^{(t)}) \tag{D.8}$$

$$\tilde{\boldsymbol{\theta}}_\gamma^{(t)} = \gamma_t \boldsymbol{\theta}^{(t)} + (1 - \gamma_t)\tilde{\boldsymbol{\theta}}_\gamma^{(t)}, \tag{D.9}$$

and let $F^\star := \min_{\boldsymbol{\theta} \in \mathcal{K}} F(\boldsymbol{\theta})$. We denote $\gamma_t = 1$, we call this *final iterate gradient descent*, because $\tilde{\boldsymbol{\theta}}_\gamma^{(t)} = \boldsymbol{\theta}_t$. When $\gamma_t = \frac{1}{t}$, we call this *full iterate averaging* because $\tilde{\boldsymbol{\theta}}_\gamma^{(t)} = \frac{1}{t}\sum_{s=1}^t \bar{\boldsymbol{\theta}}_s$, and denote it $\tilde{\boldsymbol{\theta}}^{\mathrm{avg}}$. Finally, when $\gamma_t = \gamma$ is fixed, we call this *fixed exponential moving average*, denoted $\tilde{\boldsymbol{\theta}}^{(\gamma)}$. Lacoste-Julien et al. (2012) show that the averaging parameters $\gamma_t = \frac{2}{t+1}$ are optimal, and we denote the resulting parameter $\bar{\boldsymbol{\theta}}^{\mathrm{LJ}}$ after the authors. Finally, we consider a scheme that *cannot* be represented in the form (D.9): the $\alpha$ suffix averaging analyzied in Rakhlin et al. (2011) and defined to be

$$\tilde{\boldsymbol{\theta}}_{\mathrm{suf},\alpha}^{(t)} = \frac{1}{\lceil \alpha t \rceil} \sum_{s=t-\lceil \alpha t \rceil+1}^t \boldsymbol{\theta}^{(t)}.$$

It is known that no setting of step sizes or EMA weights, and in fact, no stochastic optimization algorithm making at most $T$ queries to the gradient oracle $\mathcal{G}(\boldsymbol{\theta})$, can do better than the following information theoretic optimal rate for

$$\mathbb{E}[F(\tilde{\boldsymbol{\theta}}_T) - F^\star] = \Theta\left(\frac{G^2}{\mu T}\right) \quad \text{(information-theoretic optimal)} \tag{D.10}$$

The following theorem summarizes the performance of these various schemes relative to the information-theoretic optimal benchmark.

**Theorem 1.** *With optimal step size, the final-iterate and suffix averaging schemes described above suffer the following suboptimal lower bounds:*

- *In dimension at least $T$, exists an $F$ which is 1-strongly convex and 3-Lipschitz, and a deterministic gradient oracle such that the final (unaveraged) iterate $\boldsymbol{\theta}_T$ under update (D.8) with the standard step size $\eta_t = 1/t = 1/\mu t$ satisfies*

$$F(\boldsymbol{\theta}^{(T)}) - F^\star \geq \Omega(\log T/T) \text{ with probability one.}$$

  *More generally, the lower bound on $\alpha$-suffix averaging $\tilde{\boldsymbol{\theta}}_{\mathrm{suf},\alpha}$ is $F(\tilde{\boldsymbol{\theta}}_{\mathrm{suf},\alpha}^{(T)}) - F^\star \geq \Omega(\frac{1}{T}\log\frac{1}{\alpha})$, which is suboptimal when $\alpha = o(T)$. This is due to Harvey et al. (2019).*

- *There exists an $F$ which is $O(1)$-strongly convex and $O(1)$-Lipschitz, and a stochastic gradient oracle with $G = O(1)$ such that the average iterate produced by gradient descent with step size $\eta = 1/ct$ for any constant $c = \Omega(1)$ is shown by Rakhlin et al. (2011)*

$$\mathbb{E}[F(\tilde{\boldsymbol{\theta}}_T^{\mathrm{avg}}) - F^\star] \geq c\Omega(\log T/T)$$

*On the other hand, let $\alpha \in (0,1)$ be a constant bounded away from either zero or 1. Then the $\alpha$-suffix average $\tilde{\boldsymbol{\theta}}_{\mathrm{suf},\alpha}^{(t)}$, as well as the Lacoste-Julien average $\bar{\boldsymbol{\theta}}_t^{\mathrm{LJ}}$ satisfy*

$$\max\left\{\mathbb{E}[F(\tilde{\boldsymbol{\theta}}_{\mathrm{suf},\alpha}^{(T)}) - F^\star], \mathbb{E}[F(\bar{\boldsymbol{\theta}}_T^{\mathrm{LJ}}) - F^\star]\right\} \leq \frac{G^2}{T\mu^2}$$

*The guarantee for suffix averaging and $\bar{\boldsymbol{\theta}}^{\mathrm{LJ}}$ are due to Rakhlin et al. (2011) and Lacoste-Julien et al. (2012), respectively.*

We note that the Lacoste-Julien scheme $\bar{\boldsymbol{\theta}}_t^{\text{LJ}}$ is unique among those discussed that attains the information-theoretic optimal rate (D.10) and can also be efficiently computed in a streaming fashion if $T$ is unknown in advance[15]. In the regimes of interest, however, the functions we are minimizing are not necessarily continuous, let alone smooth. In the following section, we demonstrate that if the smoothness assumption is dropped, then EMA becomes supoptimal.

### D.3.2 EMA IS SUBOPTIMAL FOR NONSMOOTH CONVEX GD, PROVIDED THE STEP SIZE SCHEDULE IS OPTIMAL

Given the above discussion of EMA weighting schemes, a natural question to ask is whether the optimal choice of $\gamma_t$ in our empirical results is borne out by our theoretical analysis. We show that this is not the case and thus a convex mental model for the benefits of EMA we see empirically is insufficient, absent a discrepancy between the loss we are optimizing and the reward we care about. Throughout, we let $\mathcal{G}(\boldsymbol{\theta})$ denote a gradient oracle, and again consider updates given in (D.8) and (D.9). Our main result is as follows.

**Theorem 2.** *Let $T \in \mathbb{N}$ be given and let $\mathcal{K}$ denote the unit radius Euclidean ball in $\mathbb{R}^T$. There exists a 1-strongly convex, $\mathcal{O}(1)$-Lipschitz function $F(\boldsymbol{\theta})$ on $\mathcal{K}$ and a deterministic gradient oracle such that, for any $\beta \in (0, 1)$ the updates (D.8) and (D.9) with either fixed $\gamma_t \equiv \gamma = T^{-\beta}$ or time varying $\gamma_t = t^{-\beta}$ suffers a lower bound of*

$$F(\bar{\boldsymbol{\theta}}_T) - F^\star \geq \Omega \left( \frac{\beta \log T}{T} \right).$$

*Moreover, there exists an $\mathcal{O}(1)$-Lipschitz but not strongly convex function $F(\boldsymbol{\theta})$ on $\mathcal{K}$ such that gradient descent update with step size $\eta_t = \frac{1}{\sqrt{t}}$ and EMA with either $\gamma_t \equiv \gamma = T^{-\beta}$ or $\gamma_t = t^{-\beta}$ and $T \geq 2^{\frac{1}{1-\beta}}$ suffers*

$$F(\bar{\boldsymbol{\theta}}_T) - F^\star \geq \Omega \left( \frac{\beta \log T}{\sqrt{T}} \right).$$

*Finally, running EMA with step size $\gamma \leq \frac{c \log T}{T}$ for a sufficiently small universal constant $c$ in either example suffers $F(\bar{\boldsymbol{\theta}}_T) - F^\star \geq \Omega(\frac{1}{T^{1/4}})$. In particular, for all **fixed** EMA parameters $\gamma$,*

$$F(\bar{\boldsymbol{\theta}}_T) - F^\star \geq \begin{cases} \Omega \left( \frac{\log \log T}{T} \right) & \text{strongly convex case} \\ \Omega \left( \frac{\log \log T}{\sqrt{T}} \right) & \text{weakly convex case} \end{cases}$$

We emphasize that Theorem 2 demonstrates that in the regime that we find EMA empirically works, the convex theory is lacking in explanatory power. This is due to the fact that the ideal step size choices for convex optimization decay *far more rapidly* than those in the non-convex optimization of neural networks. By contrast, if the step size decays more slowly than the EMA parameter, we can achieve benefits in convex problems, as shown in Proposition D.5. And it is precisely the slow- or no-step size decay regimes that we empirically evaluate for the nonconvex optimization of deep neural networks.

## E  PROOFS

In this section we provide rigorous proofs of the statements from Appendix D.

### E.1  PROOF OF PROPOSITION D.1

*Proof of Proposition D.1.* Consider the dynamical system $\mathbf{A} = \mathbf{I}_d$, $\mathbf{B} = c\mathbf{I}_d$, and expert policy parameter $\mathbf{K}^\star = -\frac{\epsilon}{c}\mathbf{I}_d$. We let $f(\mathbf{x}, \mathbf{u}) = \mathbf{A}\mathbf{x} + \mathbf{B}\mathbf{u}$ and $\pi_\mathbf{K}(\mathbf{x}) = \mathbf{K}\mathbf{x}$. It is clear that these satisfy the requisite Lipschitz constants. For any other policy $\Delta$ which is at most $\epsilon'$ Lipschitz, we have that

$$\|\mathbf{x}_{t+1}\| \leq \|f(\mathbf{x}_t, \pi(\mathbf{x}_t) + \Delta(\mathbf{x}_t))\| = \|(\mathbf{A} - \mathbf{B}\mathbf{K})\mathbf{x}_t + c\Delta(\mathbf{x}_t)\| \leq (\|(\mathbf{A} - \mathbf{B}\mathbf{K}\| + c\epsilon')\|\mathbf{x}_t\|$$
$$\leq (1 - \epsilon + c\epsilon')\|\mathbf{x}_t\|.$$

---

[15]Observe that if $T$ is known to the learner, then one can implement tail averaging as a special case of full iterate averaging, but only begin the averating at time step $\lceil \alpha T \rceil$.

Defining $\rho := 1 - \epsilon + c\epsilon' = 1 + \delta$, we have

$$J_T(\pi_{\boldsymbol{\theta}^\star}) - J_T(\pi_{\boldsymbol{\theta}^\star} + \Delta) \leq \mathbb{E}\left[\sum_{h=1}^H \|\mathbf{x}_h\|^2\right] \leq \mathbb{E}\left[\sum_{h=1}^H \rho^{2(h-1)}\|\mathbf{x}_1\|^2\right] = \sum_{h=1}^H \rho^{2(h-1)}\mathbb{E}\left[\|\mathbf{x}_1\|^2\right]$$

$$= d\sum_{h=1}^H \rho^{2(h-1)}.$$

Moreover, by selecting $\Delta(\mathbf{x}) = \epsilon'\mathbf{x}$, this upper bound is attained. As $\rho = 1 + \delta$, for $0 \leq \delta \leq \frac{1}{2}$, it is standard to bound $d\sum_{h=1}^H \rho^{2(h-1)} \geq d\frac{H}{2}(1+\delta)^{2(\lfloor\frac{H}{2}\rfloor - 1)} = \Omega(H\exp(\delta H))$, and similarly, $\sum_{h=1}^H \rho^{2(h-1)} \leq H\rho^{2(H-1)} \leq H\exp(\mathcal{O}(H))$. For $\delta \leq 0$, then $\sum_{h=1}^H \rho^{2(h-1)}$ is $\Theta(\min\{H, \frac{1}{\delta}\})$ by a standard computation. The first result follows.

For the second statement, note that

$$\mathbb{E}\left[\left\|\left(\mathbf{K}^\star - \widehat{\mathbf{K}}\right)\mathbf{x}_t\right\|^2\right] \leq \left\|\mathbf{K}^\star - \widehat{\mathbf{K}}\right\|_{\mathrm{op}}^2 \cdot \|\mathbf{x}_t\|^2 \leq \left\|\mathbf{K}^\star - \widehat{\mathbf{K}}\right\|_{\mathrm{op}}^2 \cdot \|\mathbf{x}_1\|^2 \leq \epsilon^2 \|\mathbf{x}_1\|^2,$$

where the second inequality follows from the fact that $\|\mathbf{A} + \mathbf{B}\mathbf{K}\|_{\mathrm{op}} \leq 1$. The result follows by summing over $t$.

Lastly, we show the existence of $\Delta$'s with relatively large imitation cost but that do not suffer exponential error amplification. Define $\Delta_0 := \frac{\epsilon'}{2} \cdot \mathbf{I}$, and define the set $\tilde{\mathcal{B}}_{\epsilon'} = \{\Delta : c\|\Delta_0 - \Delta\| \leq \epsilon'/16\}$. This implies that for every $\Delta \in \tilde{\mathcal{B}}_{\epsilon'}$. Define $\tilde{\Delta} := \frac{1}{c}(\Delta - \Delta_0)$. Then,

$$(\mathbf{A} + \mathbf{B}(\mathbf{K} + \Delta)) = \underbrace{(1 - \epsilon - \epsilon'/2)\mathbf{I}}_{:=\tilde{\mathbf{A}}} + \tilde{\Delta}.$$

Note that if $\|\mathbf{x}_0\| \leq 1$ almost surely, then by the preceding analysis, it is clear that $\mathbb{E}\left[\|\mathbf{x}_h\|^2\right] \leq H$ and thus by construction, if we let $\mathbf{C} = \tilde{\mathbf{A}} + \tilde{\Delta}$, then

$$\mathbb{E}\left[\|\mathbf{x}_H\|^2\right] = \mathbb{E}\left[\left\|\mathbf{C}^H\mathbf{x}_0 + \sum_{s=1}^H \mathbf{C}^{H-s}\mathbf{w}_s\right\|\right] = \|\mathbf{C}\|_{\mathrm{op}}^H \mathbb{E}\left[\|\mathbf{x}_0\|\right] + d \cdot \sum_{s=1}^H \|\mathbf{C}\|_{\mathrm{op}}^{H-s}.$$

If we suppose that $\epsilon \ll H^{-1}$, then the above is $\Omega(H)$; thus $\mathbb{E}\left[\ell_{\mathrm{BC}}(\mathbf{K})\right] \geq \Omega(H\delta^2)$ as desired. On the other hand, because $\|\mathbf{A} + \mathbf{B}(\mathbf{K} + \Delta)\|_{\mathrm{op}} \leq \|\mathbf{A} + \mathbf{B}\mathbf{K}^\star\|_{\mathrm{op}}$, the rollout reward of the perturbed policy is greater than that of the expert. The result follows.

$\square$

### E.2 PROOF OF PROPOSITION D.2

The triangle inequality tells us that

$$\left\|\mathbf{A} + \mathbf{B}\widehat{\mathbf{K}}\right\|_{\mathrm{op}} \leq \|\mathbf{A} + \mathbf{B}\mathbf{K}\|_{\mathrm{op}} + \left\|\mathbf{B}(\mathbf{K} - \widehat{\mathbf{K}})\right\|_{\mathrm{op}} \leq 1 + \frac{\epsilon}{H},$$

by assumption. In particular, for $t \in [H]$, we have that

$$\left\|\mathbf{A} + \mathbf{B}\widehat{\mathbf{K}}\right\|_{\mathrm{op}} \leq e^\epsilon \leq 1 + 2\epsilon$$

for $\epsilon < 1$. For each $t$, we have

$$
\mathbf{x}_0^\top \left( \left( \mathbf{A} + \mathbf{B}\widehat{\mathbf{K}} \right)^t \right)^\top (\mathbf{Q} + \widehat{\mathbf{K}}\mathbf{R}) \left( \mathbf{A} + \mathbf{B}\widehat{\mathbf{K}} \right)^t \mathbf{x}_0
$$

$$
\geq \mathbf{x}_0^\top \left( (\mathbf{A} + \mathbf{B}\mathbf{K})^t \right)^\top (\mathbf{Q} + \mathbf{K}\mathbf{R}) (\mathbf{A} + \mathbf{B}\mathbf{K})^t \mathbf{x}_0
$$

$$
- \left| \mathbf{x}_0^\top \left( \left( \mathbf{A} + \mathbf{B}\widehat{\mathbf{K}} \right)^t - (\mathbf{A} + \mathbf{B}\mathbf{K})^t \right)^\top (\mathbf{Q} + \mathbf{K}\mathbf{R}) (\mathbf{A} + \mathbf{B}\mathbf{K})^t \mathbf{x}_0 \right|
$$

$$
- \left| \mathbf{x}_0^\top \left( \left( \mathbf{A} + \mathbf{B}\widehat{\mathbf{K}} \right)^t \right)^\top ((\widehat{\mathbf{K}} - \mathbf{K})\mathbf{R}) (\mathbf{A} + \mathbf{B}\mathbf{K})^t \mathbf{x}_0 \right|
$$

$$
- \left| \mathbf{x}_0^\top \left( \left( \mathbf{A} + \mathbf{B}\widehat{\mathbf{K}} \right)^t \right)^\top (\mathbf{Q} + \widehat{\mathbf{K}}\mathbf{R}) \left( \left( \mathbf{A} + \mathbf{B}\widehat{\mathbf{K}} \right)^t - (\mathbf{A} + \mathbf{B}\mathbf{K})^t \right) \mathbf{x}_0 \right|
$$

$$
\geq CH\epsilon \|\mathbf{x}_0\|^2 + \|\mathbf{x}_0\|^2 \|\mathbf{R}\|_{\mathrm{op}} (1 + 2\epsilon)^2 \epsilon + CH\epsilon \|\mathbf{x}_0\|^2
$$

$$
\geq 100C \|\mathbf{x}_0\|^2 (H + \|\mathbf{R}\|_{\mathrm{op}})\epsilon.
$$

The first result follows by summing over $t$. The second result follows similarly after observing that $\|\mathbf{A} + \mathbf{B}\mathbf{K}\|_{\mathrm{op}} \leq 1$ for all $\left\| \mathbf{K} - \widehat{\mathbf{K}} \right\|_{\mathrm{op}} \leq \frac{\delta}{2}$. $\qquad\square$

### E.3  CLIFF LOSS FOR GAUSSIAN RANDOM VECTORS (PROOF OF PROPOSITION D.3)

In this section, we prove Proposition D.3, which establishes the direct but discontinuous relationship between the the square BC loss and the cliff loss. Recall our imitation losses and cliff loss for $\boldsymbol{\mu}$ fixed:

$$
\ell_{\mathrm{BC}}(\boldsymbol{\theta}) = \frac{1}{2} \|\boldsymbol{\theta} - \boldsymbol{\mu}\|^2, \quad J(\boldsymbol{\theta}) = \begin{cases} -\|\boldsymbol{\theta} - \boldsymbol{\mu}\|^2 & \|\boldsymbol{\theta} - \boldsymbol{\mu}\| \leq \epsilon \\ -C & \text{otherwise} \end{cases},
$$

The statement (D.3) of Proposition D.3 is a direct consequence of the following lemma:

**Lemma E.1.** *Let $\boldsymbol{\theta}$ be a random Gaussian vector with covariance matrix $\boldsymbol{\Sigma}$. Then there is some constant $c$ such that for all $\epsilon > 0$,*

$$
\inf_{\boldsymbol{\mu}} \mathbb{P} \left( \|\boldsymbol{\theta} - \boldsymbol{\mu}\| > \epsilon \right) \geq 1 - c \cdot \frac{\epsilon}{\sqrt{\mathbb{E}[\|\boldsymbol{\theta} - \boldsymbol{\mu}\|^2]}}.
$$

*Proof.* By the classical Carbery-Wright inequality (Carbery & Wright, 2001, Theorem 8), for any degree $q \in \mathbb{N}$, there is a constant $c > 0$ such that for any nonnegative polynomial $P(\cdot)$ of degree $q$ in a vector-valued log concave random variable $\mathbf{z} \in \mathbb{R}^d$ such that

$$
\mathbb{P} \left( P(\mathbf{z}) \leq \epsilon^q \right) \leq c \cdot \left( \frac{\epsilon^q}{\mathbb{E}\left[ P(\mathbf{z}) \right]} \right)^{1/q}
$$

$$
= c \cdot \frac{\epsilon}{\mathbb{E}\left[ P(\mathbf{z}) \right]^{1/q}}.
$$

Noting that $\boldsymbol{\theta} - \boldsymbol{\mu} \overset{d}{=} \boldsymbol{\mu}' - \boldsymbol{\mu} + \boldsymbol{\Sigma}^{1/2}\mathbf{z}$ where $\mathbf{z} \sim \mathcal{N}(0, \mathbf{I})$, we see that $\|\boldsymbol{\theta} - \boldsymbol{\mu}\|^2$ can be expressed as as a degree 2 polynomial in the log-concave standard Gaussian variable. The result follows. $\qquad\square$

To prove the second part of Proposition D.3, we recall the following corollary of the classical Hanson-Wright inequality:

**Lemma E.2.** *Let $\boldsymbol{\theta}$ be a Gaussian random vector with mean $\boldsymbol{\mu}'$ and covariance $\boldsymbol{\Sigma}$. Then there is a constant $c$ such that for all $t > 0$, it holds that*

$$
\mathbb{P} \left( \|\boldsymbol{\theta} - \boldsymbol{\mu}\|^2 \geq 2 \cdot \|\boldsymbol{\mu} - \boldsymbol{\mu}'\|^2 + 2\mathrm{tr}(\boldsymbol{\Sigma}) + 2t \right) \leq c' e^{-c \min\left( \frac{t^2}{\|\boldsymbol{\Sigma}\|_{\mathrm{F}}^2}, \frac{t}{\|\boldsymbol{\Sigma}\|_{\mathrm{op}}} \right)}
$$

*Proof.* This follows immediately from the classical Hanson-Wright inequality (Rudelson & Vershynin, 2013) by considering the random vector $\boldsymbol{\theta} = \boldsymbol{\mu}' + \Sigma^{1/2}Z$ with $Z$ isotropic Gaussian and applying Young's inequality to upper bound $\|\boldsymbol{\theta} - \boldsymbol{\mu}\|^2 \leq 2\|\boldsymbol{\theta} - \boldsymbol{\mu}'\|^2 + 2\|\boldsymbol{\mu}' - \boldsymbol{\mu}\|^2$. $\qquad\square$

Translated into the language of Proposition D.3, set $\epsilon^2 = 2\|\boldsymbol{\mu} - \boldsymbol{\mu}'\|^2 + 2\mathrm{tr}(\boldsymbol{\Sigma}) + 2t = 4\mathbb{E}[\ell_{\mathrm{BC}}(\boldsymbol{\theta})] + 2t$. Suppose that $\epsilon^2 \geq 8\mathbb{E}[\ell_{\mathrm{BC}}(\boldsymbol{\theta})]$, so that $t = \frac{1}{2}(\epsilon^2 - 8\mathbb{E}[\ell_{\mathrm{BC}}(\boldsymbol{\theta})]) \geq \frac{\epsilon^2}{4}$. Then,

$$\mathbb{P}\left(\|\boldsymbol{\theta} - \boldsymbol{\mu}\|^2 \geq \epsilon^2\right) \leq ce^{-c\min\left(\frac{\epsilon^4}{\|\boldsymbol{\Sigma}\|_{\mathrm{F}}^2}, \frac{\epsilon^2}{\|\boldsymbol{\Sigma}\|_{\mathrm{op}}}\right)}$$

Morever, we can upper bound $\|\boldsymbol{\Sigma}\|_{\mathrm{F}}^2 \leq \mathrm{tr}(\boldsymbol{\Sigma})^2 \leq 4\mathbb{E}[\ell_{\mathrm{BC}}(\boldsymbol{\theta})]^2$ and $\|\boldsymbol{\Sigma}\|_{\mathrm{op}} \leq \mathbb{E}[\ell_{\mathrm{BC}}(\boldsymbol{\theta}) \leq 2\mathbb{E}[\ell_{\mathrm{BC}}(\boldsymbol{\theta})]$. Thus, using $\epsilon^2 \geq 8\mathbb{E}[\ell_{\mathrm{BC}}(\boldsymbol{\theta})]$, $\min\left(\frac{\epsilon^4}{\|\boldsymbol{\Sigma}\|_{\mathrm{F}}^2}, \frac{\epsilon^2}{\|\boldsymbol{\Sigma}\|_{\mathrm{op}}}\right) \geq \frac{\epsilon^2}{2\mathbb{E}[\ell_{\mathrm{BC}}(\boldsymbol{\theta})]}$. By reassigning constants, we conclude that, if $\epsilon^2 \geq 8\mathbb{E}[\ell_{\mathrm{BC}}(\boldsymbol{\theta})]$,

$$\mathbb{P}\left(\|\boldsymbol{\theta} - \boldsymbol{\mu}\|^2 \geq \epsilon^2\right) \leq c_2 e^{-c_3 \frac{\epsilon^2}{2\mathbb{E}[\ell_{\mathrm{BC}}(\boldsymbol{\theta})]}}$$

Finally, we lower bound

$$\mathbb{E}[J(\boldsymbol{\theta})] \geq -C\mathbb{P}\left(\|\boldsymbol{\theta} - \boldsymbol{\mu}\|^2 \geq \epsilon^2\right) + \mathbb{E}[-\ell_{\mathrm{BC}}(\boldsymbol{\theta})] \geq -\mathbb{E}[\ell_{\mathrm{BC}}(\boldsymbol{\theta})] - C \cdot c_2 e^{-c_1 \frac{\epsilon}{2\mathbb{E}[\ell_{\mathrm{BC}}(\boldsymbol{\theta})]}}.$$

This concludes the proof of the second statement of Proposition D.3. $\qquad\square$

### E.4 PROOF OF PROPOSITION D.5

We now turn to the proof of Proposition D.5. Introduce the error $\boldsymbol{\delta}^{(t)} = \boldsymbol{\theta}^{(t)} - \boldsymbol{\mu}$ and $\tilde{\boldsymbol{\delta}}_\gamma^{(t)} = \tilde{\boldsymbol{\theta}}_\gamma^{(t)} - \boldsymbol{\mu}$. We readily compute

$$\boldsymbol{\delta}^{(t)} = (1 - \eta)\boldsymbol{\delta}^{(t-1)} - \eta\mathbf{w}^{(t-1)}, \quad \tilde{\boldsymbol{\delta}}_\gamma^{(t)} = (1 - \gamma)\tilde{\boldsymbol{\delta}}_\gamma^{(t-1)} + \gamma\boldsymbol{\delta}$$

Let's quickly compute the No-EMA setting:

$$\boldsymbol{\delta}^{(t)} = (1 - \eta)^{T-1}\boldsymbol{\delta}^{(1)} + \eta\sum_{i=0}^{T-1}(1 - \eta)^{T-1-i}\mathbf{w}_i$$

and thus, with $|\boldsymbol{\delta}^{(1)}| = r$ and $\mathbb{E}[\mathbf{w}_t^2] = \sigma^2$,

$$\begin{aligned}
\mathbb{E}[(\boldsymbol{\delta}^{(t)})^2] &= b^2(1 - \eta)^{2T} + \sigma^2\sum_{i=0}^{T-1}(1 - \eta)^{2(T-1-t)} \\
&= b^2(1 - \eta)^{2T} + \eta^2\sigma^2\sum_{i=0}^{T-1}(1 - \eta)^{2i} \\
&= \eta^2 b^2(1 - \eta)^{2T} + \sigma^2\frac{1 - ((1 - \eta)^{2T})}{1 - (1 - \eta)^2} \\
&= \eta^2 b^2(1 - \eta)^{2T} + \sigma^2\frac{1 - ((1 - \eta)^{2T})}{\eta(2 - \eta)} \\
&= \eta b^2(1 - \eta)^{2T} + \sigma^2\frac{1 - ((1 - \eta)^{2T})}{2 - \eta}.
\end{aligned}$$

For the EMA setting, defining $\mathbf{z} = (\boldsymbol{\delta}^{(t)}, \tilde{\boldsymbol{\delta}}_\gamma^{(t)})$, we have

$$\mathbf{z}^{(t)} = \underbrace{\begin{bmatrix} (1 - \eta) & 0 \\ \gamma & (1 - \gamma) \end{bmatrix}}_{=\mathbf{A}}\mathbf{z}^{(t-1)} + \underbrace{\begin{bmatrix} -\eta \\ 0 \end{bmatrix}}_{=\mathbf{B}}\mathbf{w}^{(t-1)}$$

Let $\mathbf{e}_1, \mathbf{e}_2$ denote the canonical basis vectors for $\mathbb{R}^2$. Using $\mathbf{z}^{(0)} = (\boldsymbol{\delta}^{(0)}, \tilde{\boldsymbol{\delta}}_\gamma^{(0)}) = (\boldsymbol{\delta}^{(0)}, \boldsymbol{\delta}^{(0)}) = \boldsymbol{\delta}^{(0)}(\mathbf{e}_1 + \mathbf{e}_2)$, $\tilde{\boldsymbol{\delta}}_\gamma^{(t)} = \mathbf{e}_2^\top \mathbf{z}^{(t)}$, and $\mathbf{B}_2 = \eta \mathbf{e}_1$, we compute

$$\tilde{\boldsymbol{\delta}}_\gamma^{(T)} = \sum_{t=0}^{T-1} \eta \mathbf{e}_2^\top \mathbf{A}^{T-(t+1)} \mathbf{B} \mathbf{w}_t \mathbf{e}_1 + \mathbf{e}_2^\top \mathbf{A}^T (\mathbf{e}_1 + \mathbf{e}_2) \boldsymbol{\delta}^{(0)}$$

And thus, by independence of $\mathbf{w}_1, \dots, \mathbf{w}_{T-1} \sim \mathcal{N}(0, \sigma^2)$ and $\boldsymbol{\delta}^{(0)}$,

$$\mathbb{E}[(\tilde{\boldsymbol{\delta}}_\gamma^{(T)})^2] = \sigma^2 \sum_{t=0}^{T-1} (\eta \mathbf{e}_2^\top \mathbf{A}^{T-(t+1)} \mathbf{e}_1)^2 + (\boldsymbol{\delta}^{(0)})^2 (\mathbf{e}_2^\top \mathbf{A}^{T-1}(\mathbf{e}_1 + \mathbf{e}_2))^2$$

$$= \sigma^2 \sum_{t=0}^{T-1} (\eta \mathbf{A}^{T-(t+1)}[2,1])^2 + (\boldsymbol{\delta}^{(0)})^2 (\mathbf{A}^{T-1}[2,1] + \mathbf{A}^{T-1}[2,2])^2$$

$$= \sigma^2 \sum_{t=0}^{T-1} (\eta \mathbf{A}^t[2,1])^2 + (\boldsymbol{\delta}^{(0)})^2 (\mathbf{A}^T[2,1] + \mathbf{A}^T[2,2])^2$$

$$\le \sigma^2 \sum_{t=0}^{T-1} (\eta \mathbf{A}^t[2,1])^2 + 2(\boldsymbol{\delta}^{(0)})^2 (\mathbf{A}^T[2,1]^2 + \mathbf{A}^T[2,2]^2)$$

$$= \sigma^2 \sum_{t=0}^{T-1} (\eta \mathbf{A}^t[2,1])^2 + 2b^2 (\mathbf{A}^T[2,1]^2 + (1-\gamma)^{2(T)}) := V_T + 2b^2(1-\gamma)^{2T} \quad \text{(E.1)}$$

where above, $\mathbf{X}[i,j]$ is the $i,j$-the element of matrix $\mathbf{X}$, and where in the last line, we use that the diagonal elements of a power of triangular matrix are the powers of its diagonals, as well as $(\boldsymbol{\delta}^{(1)})^2 = b^2$.

To compute the powers $\mathbf{A}^t[2,1]$ lets first assume $\gamma \ne \eta$, and moreover, that either $\gamma \ge 2\eta$ or $\eta \ge 2\gamma$. We can address the other cases at the end. Using a formula for diagonal matrix exponentiation with $\gamma \ne \eta$, we obtain

$$\mathbf{A}^t = \begin{bmatrix} (1-\eta)^t & 0 \\ \gamma \frac{(1-\eta)^t - (1-\gamma)^t}{(1-\eta) - (1-\gamma)} & (1-\gamma)^t \end{bmatrix},$$

We then have

$$\eta \mathbf{A}^t[i,j] = \eta \gamma \frac{(1-\eta)^t - (1-\gamma)^t}{(1-\eta) - (1-\gamma)} = \eta \gamma \frac{(1-\eta)^t - (1-\gamma)^t}{\gamma - \eta} \le 2 \begin{cases} \eta(1-\eta)^t & \gamma \ge 2\eta \\ \gamma(1-\gamma)^t & \eta \ge 2\gamma \end{cases} \quad \text{(E.2)}$$

Hence,

$$V_T \le 4\sigma^2 \left( \begin{cases} \eta^2 \sum_{t=0}^{T-2} (1-\eta)^{2t} & \gamma \ge 2\eta \\ \gamma^2 \sum_{t=0}^{T-2} (1-\gamma)^{2t} & \eta \ge 2\gamma \end{cases} \right) + 4b^2 \left( \begin{cases} (1-\eta)^{2(T-1)} & \gamma \ge 2\eta \\ \frac{\gamma^2}{\eta^2}(1-\gamma)^{2(T-1)} & \eta \ge 2\gamma \end{cases} \right)$$

$$\le 4\sigma^2 \left( \begin{cases} \frac{\eta^2}{1-(1-\eta)^2} & \gamma \ge 2\eta \\ \frac{\gamma^2}{1-(1-\gamma)^2} & \eta \ge 2\gamma \end{cases} \right) + 4b^2 \left( \begin{cases} (1-\eta)^{2(T-1)} & \gamma \ge 2\eta \\ \frac{\gamma^2}{\eta^2}(1-\gamma)^{2(T-1)} & \eta \ge 2\gamma \end{cases} \right)$$

$$= 4 \left( \begin{cases} \sigma^2 \frac{\eta}{2-\eta} + b^2(1-\eta)^{2(T-1)} & \gamma \ge 2\eta \\ \sigma^2 \frac{\gamma}{2-\gamma} + b^2 \frac{\gamma^2}{\eta^2}(1-\gamma)^{2(T-1)} & \eta \ge 2\gamma \end{cases} \right)$$

$$= 4 \left( \begin{cases} \sigma^2 \eta + b^2(1-\eta)^{2(T-1)} & \gamma \ge 2\eta \\ \sigma^2 \gamma + b^2 \frac{\gamma^2}{\eta^2}(1-\gamma)^{2(T-1)} & \eta \ge 2\gamma \end{cases} \right)$$

Lets now hand the case $\eta \in [\frac{1}{2}\gamma, 2\gamma]$. In this case, observe that the entries

$$\mathbf{A} = \begin{bmatrix} (1-\eta) & 0 \\ \gamma & (1-\gamma) \end{bmatrix}$$

are non-decreasing in $\eta$. Hence, we can upper bound

$$\mathbf{A}^k \leq (\mathbf{A}')^k \text{ entrywise}, \quad \mathbf{A}' = \begin{bmatrix} (1 - \eta') & 0 \\ \gamma & (1 - \gamma) \end{bmatrix}, \quad \eta' = \frac{\eta}{4}. \tag{E.3}$$

We can now apply the same upper bound with parameters $\gamma, \eta' = \frac{\eta}{4}$. Defining $V_T'$ as corresponding to $V_T$ but with $\eta'$ instead of $V_T$, we have that

$$V_T' \leq 4(\sigma^2 \eta' + b^2(1 - \eta')^{2T}) = \sigma^2 \eta + 4b^2(1 - \frac{\eta}{4})^{2T}$$

Moreover, its easy to see from (E.1) and Eq.E.3 that

$$V_T \leq 16V_T',$$

yielding $V_T \leq 16\sigma^2\eta + 32b^2(1 - \frac{\eta}{4})^{2T}$. We conclude with a lower bound on $\mathbb{E}[(\tilde{\boldsymbol{\delta}}_\gamma^{(T)})^2]$. Repeating the computation from (E.1), and using non-negativity of the entries of the matrices $\mathbf{A}^t$, one can lower bound

$$\mathbb{E}[(\tilde{\boldsymbol{\delta}}_\gamma^{(T)})^2] \geq \sigma^2 \sum_{t=0}^{T-1} (\eta \mathbf{A}^t[2,1])^2 + b^2(\mathbf{A}^T[2,1]^2 + (1 - \gamma)^{2(T)}) := \underline{V}_T + b^2(1 - \gamma)^{2T} \tag{E.4}$$

Following the computation in (E.2), we can also lower bound

$$\eta \mathbf{A}^t[i,j] \geq \begin{cases} \eta(1 - \eta)^t & \gamma > \eta \\ \gamma(1 - \gamma)^t & \eta > \gamma \end{cases}$$

Thus,

$$\underline{V}_T \geq \sigma^2 \left( \begin{cases} \eta^2 \sum_{t=0}^{T-2}(1 - \eta)^{2t} & \gamma > 2\eta \\ \gamma^2 \sum_{t=0}^{T-2}(1 - \gamma)^{2t} & \eta > \gamma \end{cases} \right) + b^2 \left( \begin{cases} (1 - \eta)^{2(T-1)} & \gamma > \eta \\ \frac{\gamma^2}{\eta^2}(1 - \gamma)^{2(T-1)} & \eta > \gamma \end{cases} \right)$$

$$\leq 4\sigma^2 \left( \begin{cases} \frac{\eta^2(1-(1-\eta)^{2(T-1)})}{1-(1-\eta)^2} & \gamma > \eta \\ \frac{\gamma^2(1-(1-\gamma)^{2(T-1)})}{1-(1-\gamma)^2} & \eta > \gamma \end{cases} \right) + b^2 \left( \begin{cases} (1 - \eta)^{2(T-1)} & \gamma > \eta \\ \frac{\gamma^2}{\eta^2}(1 - \gamma)^{2(T-1)} & \eta > \gamma \end{cases} \right)$$

$$\overset{(i)}{\geq} \frac{1}{4} \left( \begin{cases} \sigma^2\eta + b^2(1 - \eta)^{2(T-1)} & \gamma \geq \eta \\ \sigma^2\gamma + b^2\frac{\gamma^2}{\eta^2}(1 - \gamma)^{2(T-1)} & \eta \geq \gamma \end{cases} \right),$$

where in $(i)$ we use that $(1 - (1 - x)^2) \leq 2x$ for $x \leq 1$, and the assumption that $\eta, \gamma \leq 1/2$. Using continuity of $\underline{V}_T$ and our lower bound on it, we directly extend to the $\eta = \gamma$ case. Combining with Eq.E.4,

$$\mathbb{E}[(\tilde{\boldsymbol{\delta}}_\gamma^{(T)})^2] \geq b^2(1 - \gamma)^{2T} + \frac{1}{4} \left( \begin{cases} \sigma^2\eta + b^2(1 - \eta)^{2(T-1)} & \gamma \geq \eta \\ \sigma^2\gamma + b^2\frac{\gamma^2}{\eta^2}(1 - \gamma)^{2(T-1)} & \eta \geq \gamma \end{cases} \right),$$

$\square$

## E.5 EMA FOR CONTINUOUS GAUSSIAN PROCESSES

In this section, we prove Proposition D.7 using stochastic calculuss. We begin by stating our main computation, used in tandem with Proposition D.3 to prove Proposition D.7.

**Theorem 3.** *Let $t \mapsto \gamma_t$ be continuous and nonnegative, and let $\boldsymbol{\theta}^{(t)}$ be (a) Gaussian process with almost surely continuous paths, (b) continuous semi-martingale, and (c) have pointwise finite second moments. Introduce the function*

$$G(t) := \int_0^t \gamma(s)\mathrm{d}s,$$

*and consider the random process $\widetilde{\boldsymbol{\theta}}_\gamma^{(t)}$ defined by*

$$\frac{\mathrm{d}}{\mathrm{d}t}\widetilde{\boldsymbol{\theta}}_\gamma^{(t)} = \gamma_t \cdot \left( \boldsymbol{\theta}^{(t)} - \widetilde{\boldsymbol{\theta}}_\gamma^{(t)} \right) \mathrm{d}t, \quad \widetilde{\boldsymbol{\theta}}_\gamma^{(0)} = \boldsymbol{\theta}^{(0)}.$$

*Then,*

(a) $\widetilde{\boldsymbol{\theta}}_\gamma^{(t)}$ Gaussian for each $t$.

(b) The following identity holds:

$$\mathbb{E}\left[\widetilde{\boldsymbol{\theta}}_\gamma^{(t)}\right] = e^{-G(t)} \cdot \mathbb{E}\left[\boldsymbol{\theta}^{(0)}\right] + \int_0^t e^{G(s)-G(t)}\gamma_s \cdot \mathbb{E}\left[\boldsymbol{\theta}^{(s)}\right] \mathrm{d}s$$

(c) It holds that

$$\mathbb{E}\left[\widetilde{\boldsymbol{\theta}}_\gamma^{(t)} \otimes \widetilde{\boldsymbol{\theta}}_\gamma^{(t)}\right] \preceq e^{-G(t)} \cdot \mathbb{E}\left[\boldsymbol{\theta}^{(0)} \otimes \boldsymbol{\theta}^{(0)}\right] + \int_0^t e^{G(s)-G(t)}\gamma_s \cdot \mathbb{E}\left[\boldsymbol{\theta}^{(s)} \otimes \boldsymbol{\theta}^{(s)}\right] \mathrm{d}s,$$

where the outer product and $\preceq$ denotes the Loewner order.

### E.5.1 ANALYSIS OF THE DRIFTLESS PROCESS

In this section, we study the variance reduction of EMA on the Gaussian drift process

$$\boldsymbol{\theta}^{(t)} = \int_0^t \eta_s \mathrm{d}B_s,$$

To quantify this, define

$$H(s) = \int_0^s \eta_u^2 \mathrm{d}u.$$

**Corollary E.1.** Suppose that $\boldsymbol{\theta}^{(t)} = \int_0^t \eta_s dB_s$, where $\eta_s$ is some deterministic process and $B_s$ is a Brownian motion in $\mathbb{R}^d$. Then $\widetilde{\boldsymbol{\theta}}_\gamma^{(t)}, \boldsymbol{\theta}^{(t)}$ are Gaussian for all $t$, and it holds that

$$\mathbb{E}\left[\widetilde{\boldsymbol{\theta}}_\gamma^{(t)}\right] = 0 \qquad \text{and} \qquad \mathrm{Cov}\left(\widetilde{\boldsymbol{\theta}}_\gamma^{(t)}\right) \preceq \left(\int_0^t e^{G(s)-G(t)}\gamma_s \cdot H(s)\,ds\right) \mathbf{I},$$

where $\mathbf{I}$ is the identity matrix and $H(s) = \int_0^s \eta_u^2 \mathrm{d}u$ as above. On the other hand,

$$\mathrm{Cov}\left(\boldsymbol{\theta}^{(t)}\right) = H(t) \cdot \mathbf{I}.$$

*Proof.* Note that $\boldsymbol{\theta}^{(t)}$ satisfies the conditions of Theorem 3 Furthermore, $\mathbb{E}\left[\boldsymbol{\theta}^{(t)}\right] = 0$ and $\mathrm{Cov}\left(\boldsymbol{\theta}^{(t)}\right) = \left(\int_0^t \eta_s^2\,ds\right)\mathbf{I}$. The result follows. $\qquad\square$

As stated above, Corollary E.1 is a toy model for the setting where EMA is only applied after the training loss has saturated, with $\eta_s$ denoting the continuous analogue of the learning rate in the discrete time setting of (D.5). We now consider several instantiations of this result, all assuming that $\boldsymbol{\theta}^{(0)}$ is deterministic and $\gamma_t = \gamma$ is constant.

**Example E.1** (Constant Learning Rate). We first model the constant learning rate setting where $\eta_s = \eta > 0$ for some fixed $\eta$. We first observe that $H(t) = \eta^2 t$ and thus $\mathrm{Cov}\left(\boldsymbol{\theta}^{(t)}\right) = \eta^2 t \cdot \mathbf{I}$. On the other hand, we have that

$$\mathrm{Cov}\left(\widetilde{\boldsymbol{\theta}}_\gamma^{(t)}\right) \preceq \eta^2 \left(t - \frac{1-e^{-\gamma t}}{\gamma}\right) \mathbf{I} \prec \eta^2 t \cdot \mathbf{I} = \mathrm{Cov}\left(\boldsymbol{\theta}^{(t)}\right).$$

Note that as $\gamma \downarrow 0$, $\mathrm{Cov}\left(\widetilde{\boldsymbol{\theta}}_\gamma^{(t)}\right)$ tendds to zero, while as $\gamma \uparrow \infty$, the covariance tends to $\mathrm{Cov}\left(\boldsymbol{\theta}^{(t)}\right)$.

While it demonstrates that EMA can be effective in variance reduction, the first example is not very realistic, partly because constant learning rates are rarely used in practice. For convex optimization, it is common to let $\eta_t$ scale inversely with the square root of the iteration. In this case, the separation is similarly pronounced:

**Example E.2** (Inverse Square Root Learning Rate). If we let $\eta_t = \eta(1+t)^{-\frac{1}{2}}$, as is commonly done in convex optimization, then we may see that $H(t) = \eta^2 \log(1+t)$ and so $\text{Cov}\left(\boldsymbol{\theta}^{(t)}\right) = \eta^2 \log(1+t) \cdot \mathbf{I}$. To compute the covariance of $\widetilde{\boldsymbol{\theta}}_\gamma^{(t)}$, we apply Jensen's inequality to conclude that

$$
\begin{aligned}
\int_0^t \gamma e^{\gamma(s-t)} \log(1+s)\, ds &= \left(1 - e^{-\gamma t}\right) \cdot \left(\frac{1}{1 - e^{-\gamma t}} \cdot \int_0^t \gamma e^{\gamma(s-t)} \log(1+s)\, ds\right) \\
&\leq \left(1 - e^{-\gamma t}\right) \cdot \log\left(1 + \frac{1}{1 - e^{-\gamma t}} \int_0^t \gamma e^{\gamma(s-t)} s\, ds\right) \\
&= \left(1 - e^{-\gamma t}\right) \cdot \log\left(1 + \frac{1}{(1 - e^{-\gamma t})}\left(t - \frac{1 - e^{-\gamma t}}{\gamma}\right)\right).
\end{aligned}
$$

Thus,

$$
\begin{aligned}
\text{Cov}\left(\widetilde{\boldsymbol{\theta}}_\gamma^{(t)}\right) &\preceq \left(1 - e^{-\gamma t}\right) \cdot \log\left(1 + \frac{1}{(1 - e^{-\gamma t})}\left(t - \frac{1 - e^{-\gamma t}}{\gamma}\right)\right) \cdot \mathbf{I} \\
&\prec \eta^2 \log(1+t) \cdot \mathbf{I} \\
&= \text{Cov}\left(\boldsymbol{\theta}^{(t)}\right).
\end{aligned}
$$

As in the previous example, while the variance of the iterate process is unbounded, the variance of the EMA process remains bounded. In our empirical experiments, however, we often consider a linear decay schedule (after a warmup). In this case, we have the following:

**Example E.3** (Linear Decay Learning Rate). Here we suppose that $\eta_s = 1 - \frac{s}{t}$ for $s \in [0, t]$ and we compare the final iterate $\boldsymbol{\theta}^{(t)} = \theta_\eta^{(t)}$ of different processes (indexed by $t$) to the EMA versions $\widetilde{\boldsymbol{\theta}}_\gamma^{(t)} = \widetilde{\boldsymbol{\theta}}_{\eta,\gamma}^{(t)}$. We compute directly to get that

$$
\text{Cov}\left(\widetilde{\boldsymbol{\theta}}_\gamma^{(t)}\right) \preceq \left(\frac{t}{2} - \frac{1 - e^{-\gamma t}(\gamma t + 1)}{\gamma^2 t}\right) \cdot \mathbf{I} \prec \frac{t}{2} \cdot \mathbf{I} = \text{Cov}\left(\boldsymbol{\theta}^{(t)}\right).
$$

In all of our computations, we observe that EMA leads to potentially significant variance reduction, depending on the value of $\gamma$. While the previous examples are illustrative, the driftless assumption is limiting.

### E.5.2 ANALYSIS OF THE ORNSTEIN-UHLENBECK PROCESS

We now relax the assumption that the $\boldsymbol{\theta}^{(t)}$ have saturated in the sense that the population gradients all vanish and demonstrate that even in this relaxed setting, EMA can substantially improve the cliff reward in the continuous time limit of SGD. This analysis acts as a continuous time analogue of Proposition D.5. To proceed, we first recall that Mandt et al. (2017) shows that with the appropriate scaling of the learning rate, the continuous time limit of SGD for the quadratic loss $\ell_{\text{BC}}$ is an Ornstein-Uhlenbeck process:

$$
\mathrm{d}\boldsymbol{\theta}^{(t)} = -\mathbf{A}\left(\boldsymbol{\theta}^{(t)} - \boldsymbol{\mu}\right)\mathrm{d}t + \boldsymbol{\Sigma}\mathrm{d}B_t, \tag{E.5}
$$

where $\Sigma \succ \mathbf{0}$ is positive definite and, in the context of $\ell_{\text{BC}}$, $\mathbf{A}$ and $\Sigma$ are scaled identity matrices, with the scale depending on the precise learning rate. Above $\boldsymbol{\mu} \in \mathbb{R}^d$ is the vector that minimizes $\ell_{\text{BC}}$, and $B_t$ is a $d$-dimensional Brownian motion. We now show that in this limit, corresponding to a constant learning rate, EMA can offer substantial improvement for the cliff reward.

**Proposition E.3.** *Suppose that $\boldsymbol{\theta}^{(t)}$ and $\widetilde{\boldsymbol{\theta}}_\gamma^{(t)}$ are as in (E.5) and (D.6) respectively, with the dimension $d = 1$ and $\mathbf{A} = a > 0$ fixed; further suppose that $\Sigma = 1$. Let $\epsilon \leq ca^{-\frac{1}{2}}$ for some small constant $c > 0$ and let $\ell_{\text{BC}}, J$ be as in (D.1) and (D.2). Furthermore, suppose that $\boldsymbol{\theta}^{(0)} \in \mathbb{R}$ is deterministic. If $\left|\boldsymbol{\theta}^{(0)} - \boldsymbol{\mu}\right| < K^{-1}\epsilon$, then for all $t \gg 0$, there is some choice of $\gamma > 0$ such that*

$$
J(\boldsymbol{\theta}^\star) - \mathbb{E}\left[J\left(\boldsymbol{\theta}^{(t)}\right)\right] \geq \frac{C}{2}, \quad \text{yet} \quad J(\boldsymbol{\theta}^\star) - \mathbb{E}\left[J\left(\widetilde{\boldsymbol{\theta}}_\gamma^{(t)}\right)\right] \leq 2\epsilon \ll \frac{C}{2}.
$$

*In addition,* $\max\left(\mathbb{E}\left[\ell_{\text{BC}}\left(\boldsymbol{\theta}^{(t)}\right)\right], \mathbb{E}\left[\ell_{\text{BC}}\left(\widetilde{\boldsymbol{\theta}}_\gamma^{(t)}\right)\right]\right) \leq \mathcal{O}(\epsilon).$

The above result shows that as long as $\boldsymbol{\theta}^{(0)}$ is not too far from $\boldsymbol{\mu}$, EMA can lead to substantial improvement in the cliff reward. Conversely, using a similar computation, it is possible to show that if $\left|\boldsymbol{\theta}^{(0)} - \boldsymbol{\mu}\right| > K\epsilon$ for some fixed $K$ independent of $C, \epsilon$, then our approach cannot show similar gains.

We now characterize the covariance of the EMA and non EMA'd processes under the OU process. We first recall the classical fact that $\boldsymbol{\theta}^{(t)}$ can be expressed as

$$\boldsymbol{\theta}^{(t)} = e^{-\mathbf{A}t} \cdot \boldsymbol{\theta}^{(0)} + \left(\mathbf{I} - e^{-\mathbf{A}t}\right)\boldsymbol{\mu} + \int_0^t e^{\mathbf{A}(s-t)} \cdot \Sigma \, dB_t, \tag{E.6}$$

where the last term is an Ito integral (Le Gall, 2016). The following corollary specializes to the univariate case; the more general case is handled in Corollary E.3 below.

**Corollary E.2.** *Conside the univariate case of* (E.5) *with scalar* $\mathbf{A} = a \neq 0$, $\mathbf{\Sigma} = 1$, *and* $\gamma_t \equiv \gamma > 0$ *fixed, and* $a\gamma \notin \{1, 2\}$. *Then, it holds for almost every* $\gamma$ *that* $\widetilde{\boldsymbol{\theta}}_\gamma^{(t)}, \boldsymbol{\theta}^{(t)}$ *are Gaussian for each* $t$, *and*

$$\mathbb{E}\left[\widetilde{\boldsymbol{\theta}}_\gamma^{(t)}\right] = \mathbb{E}\left[\boldsymbol{\theta}^{(t)}\right] + \left(e^{-ta} - e^{-\gamma t}\right)\frac{\gamma}{\gamma - a}\left(\boldsymbol{\theta}^{(0)} - \boldsymbol{\mu}\right)$$

$$\mathrm{Cov}\left(\widetilde{\boldsymbol{\theta}}_\gamma^{(t)}\right) \preceq e^{-\gamma t}\,\mathrm{Cov}\left(\boldsymbol{\theta}^{(0)}\right) - \frac{1 - e^{-\gamma t}}{2a} - \frac{1}{2a(1 - 2a/\gamma)}\left(e^{-\gamma t} - e^{-2at}\right)$$

*On the other hand,*

$$\mathbb{E}\left[\boldsymbol{\theta}^{(t)}\right] - \boldsymbol{\mu} = e^{-at}\left(\boldsymbol{\theta}^{(0)} - \boldsymbol{\mu}\right), \quad \mathrm{Cov}\left(\boldsymbol{\theta}^{(t)} - \boldsymbol{\theta}^{(0)}|\boldsymbol{\theta}^{(0)}\right) = \frac{1 - e^{-2at}}{2a}$$

Note that if $\mathrm{Cov}\left(\boldsymbol{\theta}^{(0)}\right) = 0$, then it holds by rearranging that

$$\mathrm{Cov}\left(\widetilde{\boldsymbol{\theta}}_\gamma^{(t)}\right) \leq \mathrm{Cov}\left(\boldsymbol{\theta}^{(t)}\right) - \frac{a}{\gamma - a}\left(e^{-\gamma t} - e^{-2at}\right) < \mathrm{Cov}\left(\boldsymbol{\theta}^{(t)}\right)$$

and thus EMA leads to variance reduction. We also have the following, more general computation for higher dimensional processes:

**Corollary E.3.** *Consider* (E.5). *Suppose that* $\mathbf{\Sigma} \succ 0$ *is a positive definite matrices and* $\boldsymbol{\mu} \in \mathbb{R}^d$ *and* $\boldsymbol{\theta}^{(t)}$ *satisfies* (E.5) *and* $\theta^{(0)}$ *is independent of the Brownian motion* $B_t$. *Suppose that* $\gamma_t = \gamma > 0$ *fixed, and* $\mathbf{I} - \gamma^{-1}\mathbf{A}$ *is nonsingular. Then,*

$$\mathbb{E}\left[\widetilde{\boldsymbol{\theta}}_\gamma^{(t)}\right] = \boldsymbol{\mu} + e^{-\gamma t}\left(\boldsymbol{\theta}^{(0)} - \boldsymbol{\mu}\right) + \left(e^{-t\mathbf{A}} - e^{-\gamma t}\mathbf{I}\right)\left(\mathbf{I} - \gamma^{-1}\mathbf{A}\right)^{-1}\left(\boldsymbol{\theta}^{(0)} - \boldsymbol{\mu}\right).$$

*Furthermore, if* $\mathbf{\Sigma} = \sigma\mathbf{I}$, $\mathbf{A}\mathbf{A}^\top = \mathbf{A}^\top\mathbf{A}$, *and if* $\mathbf{A} + \mathbf{A}^\top$ *and* $\mathbf{I} - \gamma^{-1}(\mathbf{A} + \mathbf{A}^\top)$ *are nonsingular,*

$$\mathrm{Cov}\left(\widetilde{\boldsymbol{\theta}}_\gamma^{(t)}\right) \preceq e^{-\gamma t}\,\mathrm{Cov}\left(\boldsymbol{\theta}^{(0)}\right) + \sigma^2\left(1 - e^{-\gamma t}\right)\left(\mathbf{A} + \mathbf{A}^\top\right)^{-1}$$

$$- \sigma^2\left(e^{-(\mathbf{A}+\mathbf{A}^\top)t} - e^{-\gamma t}\mathbf{I}\right)\left(\mathbf{I} - \gamma^{-1}\left(\mathbf{A} + \mathbf{A}^\top\right)\right)^{-1}\left(\mathbf{A} + \mathbf{A}^\top\right)^{-1}.$$

*On the other hand,*

$$\mathbb{E}\left[\boldsymbol{\theta}^{(t)}\right] = \boldsymbol{\mu} + e^{-\mathbf{A}t}\left(\boldsymbol{\theta}^{(0)} - \boldsymbol{\mu}\right)$$

*and*

$$\mathrm{Cov}\left(\boldsymbol{\theta}^{(t)} - \boldsymbol{\theta}^{(0)}|\boldsymbol{\theta}^{(0)}\right) = \sigma^2\left(\mathbf{I} - e^{-t\left(\mathbf{A}+\mathbf{A}^\top\right)}\right)\left(\mathbf{A} + \mathbf{A}^\top\right)^{-1}$$

*Proof.* We begin by observing that by (E.6), it holds that

$$\mathbb{E}\left[\boldsymbol{\theta}^{(t)}\right] = \boldsymbol{\mu} + e^{-At}\left(\boldsymbol{\theta}^{(0)} - \boldsymbol{\mu}\right).$$

Thus by Proposition E.4, we have

$$
\begin{aligned}
\mathbb{E}\left[\widetilde{\boldsymbol{\theta}}_\gamma^{(t)}\right] &= e^{-G(t)} \cdot \boldsymbol{\theta}^{(0)} + \int_0^t e^{G(s)-G(t)} \gamma \cdot \mathbb{E}\left[\boldsymbol{\theta}^{(s)}\right] \, \mathrm{d}s \\
&= e^{-\gamma t} \cdot \boldsymbol{\theta}^{(0)} + \int_0^t e^{\gamma s - \gamma t} \gamma \cdot \left(\boldsymbol{\mu} + e^{-\mathbf{A}s}\left(\boldsymbol{\theta}^{(0)} - \mu\right)\right) \, \mathrm{d}s \\
&= e^{-\gamma t} \cdot \boldsymbol{\theta}^{(0)} + \left(1 - e^{-\gamma t}\right)\boldsymbol{\mu} + e^{-\gamma t}\gamma \cdot \int_0^t e^{\left(\mathbf{I}-\gamma^{-1}\mathbf{A}\right)\gamma s}\left(\boldsymbol{\theta}^{(0)} - \mu\right) \, \mathrm{d}s,
\end{aligned}
$$

where the final equality follows because $\gamma \mathbf{I}$ commutes with $\mathbf{A}$. A simple calculation then tells us that

$$
\int_0^t e^{\left(\mathbf{I}-\gamma^{-1}\mathbf{A}\right)\gamma s}\left(\boldsymbol{\theta}^{(0)} - \mu\right) \, ds = \left(e^{\gamma t\mathbf{I} - t\mathbf{A}} - \mathbf{I}\right)\left(\gamma\mathbf{I} - \mathbf{A}\right)^{-1}\left(\boldsymbol{\theta}^{(0)} - \mu\right),
$$

and thus

$$
\mathbb{E}\left[\widetilde{\boldsymbol{\theta}}_\gamma^{(t)}\right] = e^{-\gamma t} \cdot \boldsymbol{\theta}^{(0)} + \left(1 - e^{-\gamma t}\right)\mu + \left(e^{-t\mathbf{A}} - e^{-\gamma t}\mathbf{I}\right)\left(\mathbf{I} - \gamma^{-1}\mathbf{A}\right)^{-1}\left(\boldsymbol{\theta}^{(0)} - \mu\right).
$$

Thus the expressions of the means of both $\boldsymbol{\theta}^{(t)}$ and $\widetilde{\boldsymbol{\theta}}_\gamma^{(t)}$ hold.

For the covariances, we first observe that by (E.6), if $\mathbf{A}$ commutes with $\mathbf{A}^\top$ and $\boldsymbol{\Sigma} = \sigma\mathbf{I}$, then if $\mathbf{A} + \mathbf{A}^\top$ is invertible, we have

$$
\begin{aligned}
\mathrm{Cov}\left(\boldsymbol{\theta}^{(t)} - \boldsymbol{\theta}^{(0)} | \boldsymbol{\theta}^{(0)}\right) &= \sigma^2 \cdot \int_0^t e^{\mathbf{A}(s-t) + \mathbf{A}^\top(s-t)} ds \\
&= \sigma^2 \left(\mathbf{I} - e^{-t(\mathbf{A}+\mathbf{A}^\top)}\right)\left(\mathbf{A} + \mathbf{A}^\top\right)^{-1}.
\end{aligned}
$$

Now, by Proposition E.5, we have that

$$
\begin{aligned}
\mathrm{Cov}\left(\widetilde{\boldsymbol{\theta}}_\gamma^{(t)}\right) &\preceq e^{-\gamma t}\mathrm{Cov}\left(\boldsymbol{\theta}^{(0)}\right) + \int_0^t e^{\gamma(s-t)}\gamma s\sigma^2\left(\mathbf{I} - e^{-s(\mathbf{A}+\mathbf{A}^\top)}\right)\left(\mathbf{A} + \mathbf{A}^\top\right)^{-1} \, \mathrm{d}s \\
&= e^{-\gamma t}\mathrm{Cov}\left(\boldsymbol{\theta}^{(0)}\right) + \sigma^2\left(1 - e^{-\gamma t}\right)\left(\mathbf{A} + \mathbf{A}^\top\right)^{-1} \\
&\quad - \sigma^2\gamma\left(\int_0^t e^{\gamma(s-t)\mathbf{I} - s(\mathbf{A}+\mathbf{A}^\top)} \, \mathrm{d}s\right)\left(\mathbf{A} + \mathbf{A}^\top\right)^{-1} \\
&= e^{-\gamma t}\mathrm{Cov}\left(\boldsymbol{\theta}^{(0)}\right) + \sigma^2\left(1 - e^{-\gamma t}\right)\left(\mathbf{A} + \mathbf{A}^\top\right)^{-1} \\
&\quad - \sigma^2\left(e^{-(\mathbf{A}+\mathbf{A}^\top)t} - e^{-\gamma t}\mathbf{I}\right)\left(\mathbf{I} - \gamma^{-1}\left(\mathbf{A} + \mathbf{A}^\top\right)\right)^{-1}\left(\mathbf{A} + \mathbf{A}^\top\right)^{-1}.
\end{aligned}
$$

The result follows. $\qquad\square$

### E.5.3  PROOF OF THEOREM 3

Theorem 3 follows from three constituent propositions, corresponding to items (b), (c), and (a) of the theorem, respectively.

**Proposition E.4.** *Let $\boldsymbol{\theta}^{(t)}$ be a continuous semi-martingale (see Le Gall (2016, Definition 4.19)) and let $\widetilde{\boldsymbol{\theta}}_\gamma^{(t)}$ be defined such that $\widetilde{\boldsymbol{\theta}}_\gamma^{(0)} = \boldsymbol{\theta}^{(0)}$ and (D.6) holds. Suppose that $\gamma_t$ is continuous and let $G(t) = \int_0^t \gamma(s)ds$. Then,*

$$
\widetilde{\boldsymbol{\theta}}_\gamma^{(t)} = e^{-G(t)} \cdot \boldsymbol{\theta}^{(0)} + \int_0^t e^{G(s)-G(t)}\gamma_s \cdot \boldsymbol{\theta}^{(s)} \, ds. \tag{E.7}
$$

*Furthermore, if $\left\|\boldsymbol{\theta}^{(t)}\right\|$ has finite first moment for all $s \leq t$, then*

$$
\mathbb{E}\left[\widetilde{\boldsymbol{\theta}}_\gamma^{(t)}\right] = e^{-G(t)} \cdot \mathbb{E}\left[\boldsymbol{\theta}^{(0)}\right] + \int_0^t e^{G(s)-G(t)}\gamma_s \cdot \mathbb{E}\left[\boldsymbol{\theta}^{(s)}\right] ds. \tag{E.8}
$$

*Proof.* By the Picard-Lindelöf theorem (Lindelöf, 1894), $\widetilde{\boldsymbol{\theta}}_\gamma^{(t)}$ exists and is unique. Noting that $\widetilde{\boldsymbol{\theta}}_\gamma^{(t)}$ given in (E.7) is differentiable by the fundamental theorem of calculus, we have

$$\frac{d\widetilde{\boldsymbol{\theta}}_\gamma^{(t)}}{dt} = -\gamma_t e^{-G(t)}\boldsymbol{\theta}^{(0)} + \gamma_t\boldsymbol{\theta}^{(t)} - \gamma_t \cdot \int_0^t e^{G(s)-G(t)}\gamma_s \cdot \boldsymbol{\theta}^{(s)}\, ds$$
$$= \gamma_t\left(\boldsymbol{\theta}^{(t)} - \widetilde{\boldsymbol{\theta}}_\gamma^{(t)}\right).$$

Because $G(0) = 0$, we have $\widetilde{\boldsymbol{\theta}}_\gamma^{(0)} = \boldsymbol{\theta}^{(0)}$ and thus by the uniqueness of the solution the expression given in (E.7) is correct.

For the second statement, we may apply Fubini's theorem and the first moment assumption. $\qquad\square$

We remark that if (E.7) seems unmotivated, one can construct a guess for this solution informally in much the same way as the final part of the proof of Proposition E.5 below proceeds. Our second main computation is to upper bound the covariance matrix of $\widetilde{\boldsymbol{\theta}}_\gamma^{(t)}$ by that of $\boldsymbol{\theta}^{(t)}$:

**Proposition E.5.** *Let $\boldsymbol{\theta}^{(t)}$ and $\gamma_t$ be as in Proposition E.4. Further suppose that $\mathbb{E}\left[\left\|\boldsymbol{\theta}^{(t)}\right\|^2\right] < \infty$ for all $t$. Then,*

$$\mathbb{E}\left[\widetilde{\boldsymbol{\theta}}_\gamma^{(t)} \otimes \widetilde{\boldsymbol{\theta}}_\gamma^{(t)}\right] \preceq e^{-G(t)} \cdot \mathbb{E}\left[\boldsymbol{\theta}^{(0)} \otimes \boldsymbol{\theta}^{(0)}\right] + \int_0^t e^{G(s)-G(t)}\gamma_s \cdot \mathbb{E}\left[\boldsymbol{\theta}^{(s)} \otimes \boldsymbol{\theta}^{(s)}\right]\, ds, \qquad \text{(E.9)}$$

*where $\mathbf{u} \otimes \mathbf{v} = \mathbf{u}\mathbf{v}^\top$ is the outer product and $\preceq$ denotes the Loewner order.*

*Proof.* We begin by applying the chain rule to get

$$\frac{d\left(\widetilde{\boldsymbol{\theta}}_\gamma^{(t)} \otimes \widetilde{\boldsymbol{\theta}}_\gamma^{(t)}\right)}{dt} = \nabla\left(\widetilde{\boldsymbol{\theta}}_\gamma^{(t)} \otimes \widetilde{\boldsymbol{\theta}}_\gamma^{(t)}\right)_{ij}^k \left(\frac{d\widetilde{\boldsymbol{\theta}}_\gamma^{(t)}}{dt}\right)_k,$$

where we are using Einstein notation to express the contraction of the third order tensor coming from the Jacobean with the first order tensor arising from the time derivative of $\widetilde{\boldsymbol{\theta}}_\gamma$. By the definition of $\widetilde{\boldsymbol{\theta}}_\gamma^{(t)}$, we have then that

$$\frac{d\left(\widetilde{\boldsymbol{\theta}}_\gamma^{(t)} \otimes \widetilde{\boldsymbol{\theta}}_\gamma^{(t)}\right)}{dt} = \widetilde{\boldsymbol{\theta}}_\gamma^{(t)} \otimes \left(\frac{d\widetilde{\boldsymbol{\theta}}_\gamma^{(t)}}{dt}\right) + \left(\frac{d\widetilde{\boldsymbol{\theta}}_\gamma^{(t)}}{dt}\right) \otimes \widetilde{\boldsymbol{\theta}}_\gamma^{(t)}$$
$$= \gamma_t \cdot \left(\widetilde{\boldsymbol{\theta}}_\gamma^{(t)} \otimes \left(\boldsymbol{\theta}^{(t)} - \widetilde{\boldsymbol{\theta}}_\gamma^{(t)}\right) + \left(\boldsymbol{\theta}^{(t)} - \widetilde{\boldsymbol{\theta}}_\gamma^{(t)}\right) \otimes \widetilde{\boldsymbol{\theta}}_\gamma^{(t)}\right)$$
$$\preceq \gamma_t\left(\boldsymbol{\theta}^{(t)} \otimes \boldsymbol{\theta}^{(t)} - \widetilde{\boldsymbol{\theta}}_\gamma^{(t)} \otimes \widetilde{\boldsymbol{\theta}}_\gamma^{(t)}\right),$$

where we used the fact that for any two vectors $\mathbf{u}, \mathbf{v} \in \mathbb{R}^d$, it holds that

$$\mathbf{u} \otimes \mathbf{v} + \mathbf{v} \otimes \mathbf{u} \preceq \mathbf{u} \otimes \mathbf{u} + \mathbf{v} \otimes \mathbf{v}.$$

We may now take expectations on both sides of the preceding inequality and observe that the expectation exists by the finiteness of the second moment of the norm. Let $\boldsymbol{\Sigma}(t) = \mathbb{E}\left[\boldsymbol{\theta}^{(t)} \otimes \boldsymbol{\theta}^{(t)}\right]$ and $\widetilde{\Sigma}(t) = \mathbb{E}\left[\widetilde{\boldsymbol{\theta}}_\gamma^{(t)} \otimes \widetilde{\boldsymbol{\theta}}_\gamma^{(t)}\right]$; then, by Fubini's theorem and the above computation, we have

$$\frac{d\tilde{\boldsymbol{\Sigma}}(t)}{dt} \preceq \gamma_t\left(\boldsymbol{\Sigma}(t) - \tilde{\boldsymbol{\Sigma}}(t)\right).$$

Multiplying by $\exp(G(t))$ on both sides (which is positive definite and thus preserves the order) and rearranging gives

$$\frac{d}{dt}\left(e^{G(t)}\tilde{\boldsymbol{\Sigma}}(t)\right) = e^{G(t)} \cdot \frac{d\tilde{\boldsymbol{\Sigma}}(t)}{dt} + e^{G(t)}\gamma_t\tilde{\boldsymbol{\Sigma}}(t) \preceq e^{G(t)}\boldsymbol{\Sigma}(t).$$

Integrating and again rearranging tells us that

$$\tilde{\boldsymbol{\Sigma}}(t) \preceq e^{-G(t)}\int_0^t e^{G(s)}\gamma_s\boldsymbol{\Sigma}(s)\, ds.$$

The result follows. $\qquad\square$

Note that both (E.8) and (E.9) should make intuitive sense. Indeed in the two limits, as $\gamma \downarrow 0$ (corresponding to only keeping the first iterate) and $\gamma \uparrow \infty$ (only keeping the last iterate), both expressions behave as expected. We remark that in the case where $\boldsymbol{\theta}^{(t)} \in \mathbb{R}$ is a scalar, then Jensen's inequality proves the result directly, without the need for the more complicated computations. As a corollary of the above, letting $\mathrm{Cov}(\cdot)$ denote the covariance matrix of a random vector and combining (E.8) with (E.9), we see that

$$\mathrm{Cov}\left(\widetilde{\boldsymbol{\theta}}_\gamma^{(t)}\right) \preceq e^{-G(t)} \cdot \mathrm{Cov}\left(\boldsymbol{\theta}^{(0)}\right) + \int_0^t e^{G(s)-G(t)} \gamma_s \cdot \mathrm{Cov}\left(\boldsymbol{\theta}^{(s)}\right) ds.$$

In order to simplify our computations, we will suppose that $\boldsymbol{\theta}^{(t)}$ is Gaussian for all $t$. In this case, the following lemma shows that $\widetilde{\boldsymbol{\theta}}_\gamma^{(t)}$ is also Gaussian for all $t$:

**Lemma E.6.** *Suppose that $\left(\boldsymbol{\theta}^{(s)}\right)_{0<s\leq t}$ is a Gaussian process with almost surely continuous sample paths. Then, $\widetilde{\boldsymbol{\theta}}_\gamma^{(t)} - e^{-G(t)}\boldsymbol{\theta}^{(0)}$ is also Gaussian.*

*Proof.* We begin by observing that for any partition $0 = s_0 < s_1 < \cdots < s_{n-1} < s_n = t$, the random vector

$$\Theta_n = \sum_{j=1}^n \gamma_{s_j} e^{G(s_j)-G(t)} \boldsymbol{\theta}^{(s_j)} \left(s_j - s_{j-1}\right)$$

is Gaussian with

$$\mathbb{E}\left[\Theta_n\right] = \sum_{j=1}^n \gamma_{s_j} e^{G(s_j)-G(t)} \mathbb{E}\left[\boldsymbol{\theta}^{(s_j)}\right] \left(s_j - s_{j-1}\right)$$

and

$$\mathbb{E}\left[\Theta_n \otimes \Theta_n\right] = \sum_{j=1}^n \sum_{i=1}^n \gamma_{s_j} \gamma_{s_i} e^{(G(s_j)-G(t))+(G(s_i)-G(t))} \mathbb{E}\left(\boldsymbol{\theta}^{(s_i)} \otimes \boldsymbol{\theta}^{(s_j)}\right) \left(s_j - s_{j-1}\right)\left(s_i - s_{i-1}\right),$$

by the fact that $\left(\boldsymbol{\theta}^{(s)}\right)_{0<s\leq t}$ is a Gaussian process. Now, by the continuity assumption, we see that as $n \uparrow \infty$, it holds that $\mathbb{E}\left[\Theta_n\right]$ and $\mathbb{E}\left[\Theta_n \otimes \Theta_n\right]$ approach limits given by the Riemann integral. Thus, because the limit of a Gaussian sequence is Gaussian (in $L^p$ for all $1 \leq p < \infty$ by, for example Le Gall (2016, Proposition 1.1)), we see that $\widetilde{\boldsymbol{\theta}}_\gamma^{(t)} - e^{-G(t)}\boldsymbol{\theta}^{(0)}$ is Gaussian, as desired. □

### E.6 Instantiations of cliff loss in continuous time

We now prove the main results. We begin with the simpler result (presented second), where we do not concern ourselves with drift.

*Proof of Proposition D.7.* By Lemma E.6, $\widetilde{\boldsymbol{\theta}}_\gamma^{(t)}$ is Gaussian. Let $t \gg 0$ such that $c \cdot \frac{\epsilon}{\sqrt{\mathrm{tr}\left(\mathrm{Cov}\left(\boldsymbol{\theta}^{(t)}\right)\right)}} \leq \frac{1}{2}$, which occurs because in all regimes, except the third, $\mathrm{Cov}\left(\boldsymbol{\theta}^{(t)}\right) \uparrow \infty$; in the case of inverse learning rate, simply set $\eta$ large enough so that this holds. By Lemma E.1, it holds that

$$\mathbb{P}\left(J\left(\boldsymbol{\theta}^{(t)}\right) \leq -C\right) \geq 1 - c \cdot \frac{\epsilon}{\sqrt{\mathrm{tr}\left(\mathrm{Cov}\left(\boldsymbol{\theta}^{(t)}\right)\right)}} \geq \frac{1}{2}.$$

Thus $\mathbb{E}\left[J\left(\boldsymbol{\theta}^{(t)}\right)\right] \leq -\frac{C}{2}$. On the other hand, if we let $\gamma$ such that the upper bound on $\mathrm{Cov}\left(\widetilde{\boldsymbol{\theta}}_\gamma^{(t)}\right)$ is at most $c\frac{\epsilon^2}{d\log\left(\frac{C}{\epsilon}\right)}$, then by Lemma E.2 it holds that with probability at least $1 - \frac{\epsilon}{C}$ that $\left\|\widetilde{\boldsymbol{\theta}}_\gamma^{(t)}\right\| \leq \epsilon$. By the construction of $J$, the first result holds. For the second result, note that $\ell_{\mathrm{BC}} = J$ whenever $\left\|\boldsymbol{\theta}^{(t)}\right\| \leq \epsilon$; the result holds by letting $\eta = \Theta(\epsilon)$. □

Finally, we prove the second result, for OU processes.

*Proof of Proposition E.3.* We apply Corollary E.2 to compute the first and second moments of $\boldsymbol{\theta}^{(t)}$ and $\widetilde{\boldsymbol{\theta}}_\gamma^{(t)}$. Let $R = \left|\boldsymbol{\theta}^{(0)} - \boldsymbol{\mu}\right|$. We now note that for $t \gg 0$, by the assumption that $\epsilon \leq ca^{-\frac{1}{2}}$ for some small $c$, it holds that $\mathrm{Cov}\left(\boldsymbol{\theta}^{(t)}\right) \geq 4c^2\epsilon^2$. Thus, by Hoffmann-Jorgensen et al. (1979) and the convexity of the norm followed by Lemma E.1, it holds that

$$\mathbb{P}\left(\left|\boldsymbol{\theta}^{(t)} - \boldsymbol{\mu}\right| < \epsilon\right) \leq \mathbb{P}\left(\left|\boldsymbol{\theta}^{(t)} - \mathbb{E}\left[\boldsymbol{\theta}^{(t)}\right]\right| < \epsilon\right) \leq c \cdot \frac{\epsilon}{4c^2\epsilon^2} \leq \frac{1}{2}.$$

Thus, by the construction of the cliff loss,

$$\mathbb{E}\left[J\left(\boldsymbol{\theta}^{(t)}\right)\right] \leq -\frac{C}{2}.$$

On the other hand, we note that if $\gamma \leq \frac{a}{2}$ then by the computation of $\mathbb{E}\left[\widetilde{\boldsymbol{\theta}}_\gamma^{(t)}\right]$, we have

$$\left|\mathbb{E}\left[\widetilde{\boldsymbol{\theta}}_\gamma^{(t)} - \boldsymbol{\mu}\right] - \boldsymbol{\mu}\right| \leq 3R.$$

On the other hand, for $\gamma \downarrow 0$, we see that $\mathrm{Cov}\left(\widetilde{\boldsymbol{\theta}}_\gamma^{(t)}\right) \downarrow 0$ and thus we may apply Lemma E.2 to conclude that as long as $3R < 1$, there is some small $\gamma$ such that

$$\mathbb{P}\left(\left|\widetilde{\boldsymbol{\theta}}_\gamma^{(t)} - \boldsymbol{\mu}\right| > \epsilon\right) \leq \mathbb{P}\left(\left|\widetilde{\boldsymbol{\theta}}_\gamma^{(t)} - \mathbb{E}\left[\widetilde{\boldsymbol{\theta}}_\gamma^{(t)}\right]\right| > (1 - 3R)\epsilon\right) \leq \frac{\epsilon}{C}.$$

The the first statement then follows from the construction of the cliff loss as long as $K > 3$. The second statement follows in the same way as in the proof of Proposition D.7. $\square$

### E.7 PROOF OF THEOREM 2

First, we show a reduction to lower bounding a quantity involving the $k$-length suffix average.

**Lemma E.7.** *Let $F$ satisfy $F(\boldsymbol{\theta}) - F^\star \geq \langle \mathbf{h}, \boldsymbol{\theta} \rangle$, and let $(\boldsymbol{\theta}^{(t)})$ be a sequence of iterates with $\langle \mathbf{h}, \boldsymbol{\theta}^{(t)} \rangle \geq 0$. Finally, let $(w_t)$ be a sequence of weights which are non-increasing. Then, for any $1 \leq k \leq K$, we have*

$$F\left(\sum_{t=1}^T w_t \boldsymbol{\theta}^{(t)}\right) - F^\star \geq \max\left(w_{T-k+1} \sum_{t=T-k+1}^T \langle \boldsymbol{\theta}^{(t)}, \mathbf{h} \rangle, \quad w_1 \mathbf{h}^\top \boldsymbol{\theta}^{(1)}\right)$$

*Proof of Lemma E.7.* Under the assumption of the lemma, we lower bound

$$\begin{aligned}
F\left(\sum_{t=1}^T w_t \boldsymbol{\theta}^{(t)}\right) &\geq \mathbf{h}^\top \left(\sum_{t=1}^T w_t \boldsymbol{\theta}^{(t)}\right) \\
&\geq \sum_{t=1}^T w_t \langle \boldsymbol{\theta}^{(t)}, \mathbf{h} \rangle \\
&\geq \sum_{t=T-k+1}^T w_t \langle \boldsymbol{\theta}^{(t)}, \mathbf{h} \rangle \\
&\geq w_{T-k+1} \sum_{t=T-k+1}^T \langle \boldsymbol{\theta}^{(t)}, \mathbf{h} \rangle \\
&= k w_{T-k+1} \cdot \frac{1}{k} \sum_{t=T-k+1}^T \langle \boldsymbol{\theta}^{(t)}, \mathbf{h} \rangle.
\end{aligned}$$

It is straightforward to also bound the above expression by $w_1 \mathbf{h}^\top \boldsymbol{\theta}^{(1)}$. $\square$

We can now prove our desired lower bounds. We can write

$$\widetilde{\boldsymbol{\theta}}_\gamma^{(T)} = \sum_{t=1}^T w_t \boldsymbol{\theta}^{(t)},$$

where the weights induced by the EMA averaging are

$$w_1 = \prod_{t=2}^{T}(1 - \gamma_t), \quad w_k = \gamma_t \prod_{t=k+1}^{T}(1 - \gamma_t)$$

and which, for fixed $\gamma$, satisfy

$$w_1 = (1 - \gamma)^T, \quad w_k = \gamma(1 - \gamma)^{T-k}.$$

Harvey et al. (2019, Theorem 3.4) exhibit a function $F$ which is 1-strongly convex and 3-Lipschitz, satisfies $F(\boldsymbol{\theta}) - F^\star \geq \langle \mathbf{h}, \boldsymbol{\theta} \rangle$, and sequence of iterates $\boldsymbol{\theta}^{(t)}$ corresponding to gradient descent with step size $\eta_t = 1/t$ satisfying $\langle \boldsymbol{\theta}^{(t)}, \mathbf{h} \rangle \geq 0$ for all $t$ and

$$\frac{1}{k} \sum_{t=T-k+1}^{T} \langle \boldsymbol{\theta}^{(t)}, \mathbf{h} \rangle \geq \Omega\left(\frac{\log(T/k)}{T}\right), \quad \langle \mathbf{h}, \boldsymbol{\theta}_1 \rangle = \Omega(1).$$

Similarly, it is shown that there exists an $\mathcal{O}(1)$ Lipschitz function inducing iterates that satisfy

$$\frac{1}{k} \sum_{t=T-k+1}^{T} \langle \boldsymbol{\theta}^{(t)}, \mathbf{h} \rangle \geq \Omega\left(\frac{\log(T/k)}{\sqrt{T}}\right),$$

Thus, for any non-increasing weighting scheme $(w_t)$, we have for all $k$

$$F(\tilde{\boldsymbol{\theta}}_\gamma^{(T)}) - F^\star \geq k w_{T-k+1} \log(T/k) \cdot \begin{cases} \Omega\left(\frac{1}{T}\right) & \text{strongly convex case} \\ \Omega(\sqrt{T}) & \textit{non-strongly convex case} \end{cases}$$

We quickly note that for both these functions, we also have the lower bound $F(\tilde{\boldsymbol{\theta}}_\gamma^{(T)}) - F^\star \geq w_1 \boldsymbol{\theta}_1$ due to Lemma E.7. For $\gamma$ fixed, the expression $w_1 = (1 - \gamma)^T$ shows that $F(\tilde{\boldsymbol{\theta}}_\gamma^{(T)}) - F^\star \geq \frac{1}{T^\rho}$ provived that $\gamma = \frac{c\rho \log T}{T}$ for $c$ a universal constant. Selecting $\rho = 1/4$ gives us the last statement of our theorem.

It remains to quantify the worst-case behavior of

$$\max\left(k w_{T-k+1} \log(T/k), w_1\right)$$

under EMA.

- Consider EMA with parameter $\gamma = T^{-\beta}$ and choose $k = \lceil 1/\gamma \rceil = \lceil T^\beta \rceil$. Then,

$$k w_{T-k+1} \geq k\gamma(1 + \gamma)^k = \Omega(\gamma k) = \Omega(1).$$

  Moreover, $\log(T/k) = \Omega(\log(T^\beta)) = \Omega(\beta \log T)$. Thus, $k w_{T-k+1} \log(T/k) = \Omega(\beta \log T)$.

- Consider EMA with decaying parameter $\gamma_t = t^{-\beta}$. Then, we have for $k \geq T/2$

$$w_{T-k+1} = (T - k + 1)^{-\beta} \prod_{t=T-k+1}^{T} (1 - t^{-\beta}) \geq (T - k + 1)^{-\beta}(1 - (T - k + 1)^{-\beta})^k \tag{E.10}$$

  Again select $k = \lceil T^\beta \rceil$ and assume that $k \leq T/2$, which requires $T^\beta \leq T/2$, or $T \geq 2^{\frac{1}{1-\beta}}$. Then, we can lower bound (E.10) by is at least $2^{-\beta}T^{-\beta}(1 - 2T^{-\beta})^k$. Aggain, for $k = \Omega(1/T^{-\beta})$ and $\beta \in [0, 1]$, this expression is $\Omega(T^{-\beta})$. Hence $k w_{T-k+1} = \Omega(1)$ and $k w_{T-k+1} \log(T/k) = \Omega(\beta \log T)$.

$\square$

