# OpenReview forum: "Butterfly Effects of SGD Noise: Error Amplification in Behavior Cloning and Autoregression"
_ICLR.cc/2024/Conference — ICLR 2024 poster_

### Official Review · Reviewer_fhjo · 2023-10-27

**Soundness:** 1 poor
**Presentation:** 2 fair
**Contribution:** 2 fair
**Rating:** 3
**Confidence:** 2

**Summary:**

The paper proposes that the training instability in the policy network is due to the gradient variance amplication during training

**Strengths:**

The idea is novel and the problem being dealt with is sufficiently important, namely, during training in reinforcement learning, the model does experience strong instabilities, and good methods to mitigate them would be of great importance

**Weaknesses:**

I am quite confused by the main claim of the work.

The main claim is that the gradient variance is responsible for the instability and that it amplifies throughout training, but no numerical result in the paper really plots the gradient variance, and, of course, not a single experiment shows that this variance is amplified. It should not be difficult to plot the variance of the gradient at all, and the absence of such evidence makes it impossible for me to recommend acceptance.

**Questions:**

See weakness

---

> ### Author Response · Authors · 2023-11-17
> **Response to Review**
>
> Thank you for your feedback. We would like to clarify what we mean by GVA. The “GV”, or gradient variance, stands for stochasticity introduced by the stochastic gradient updates. The presence of gradient variance is a known phenomenon. By amplification, we **do not mean that the training procedure increases variance of gradients**. Rather, we mean that **the evaluation of a policy in a closed-loop setting, namely, imitation learning, is responsible for said amplification**. More specifically, as is **highlighted in bold on page 5 of the submission**, as well as *emphasized in italics several lines below*, GVA is the phenomenon by which small perturbations of parameters, caused by variance in the stochastic gradients used during training, is amplified by the closed feedback loop created when past model predictions are fed back into the model during the rollout.
>
> Thus, it is the disparity between the reward function of interest and the loss being optimized that leads to these oscillations.  **Nowhere do we claim that this variance is amplified during training, which is why no such plot is exhibited in our paper.**  To reiterate, the main claim is **not that gradient variance amplifies through training** but rather that closed feedback loops lead to parameter sensitivity, which leads to poor performance when coupled with gradient stochasticity.
>
> Our experiments were carefully chosen to justify our diagnosis of the GVA phenomenon:
> (a) More data, and various forms of data noising / regularization, **do not** address the issue
> (b) Increasing batch size does improve, but not entirely ameliorate GVA
> (c) EMA, which reduces iterate variance, nearly entirely fixes GVA
> (d) Through our analysis of local perturbations around imitator policies at different stages of training (Fig 1 right and Fig 13), we observe that while the *validation loss* landscape is relatively insensitive to parameter perturbations, the *policy evaluation loss* landscape is very sensitive to these.
>
> In summary, we conclude that (i) evaluation loss is highly sensitive to changes in parameters and (ii) that methods which reduce variance in iterates/gradients seem to ameliorate this sensitivity.  The role of our theory is to show that the above hypothesis is mathematically plausible.

---

### Official Review · Reviewer_xe8u · 2023-10-31

**Soundness:** 3 good
**Presentation:** 3 good
**Contribution:** 2 fair
**Rating:** 5
**Confidence:** 3

**Summary:**

The paper examines how noise in gradients affects systems with feedback loops. It argues that SGD error accumulates over longer horizons, leading to Gradient Variance Amplification (GVA), explaining sharp reward oscillations. GVA is shown to be the main cause of these oscillations, surpassing the influence of statistical and architectural factors. Empirical evidence shows that the empirical moving average (MVA) mitigates these oscillations, ensuring stability. Additionally, the paper revisits the theory of stochastic optimization for convex functions, assessing its explanatory power for empirical observations of EMA and various step size schedules.

**Strengths:**

a) The paper conducts meticulous experiments revealing that the variance in stochastic gradients is the true source of instability. It introduces exponential moving average as an effective solution to mitigate this issue. Additionally, this phenomenon is also demonstrated for other tasks with feedback loop, i.e., auto regressive processes for language generation.
b) The paper is well-written and the main message is clearly presented.

**Weaknesses:**

a) The main problem of the paper in my opinion is the explanation provided for the benefits of EMA.  Proposition 3.1 says that land scape is very intricate and for every $\delta$ there is a separation between $J$ and the behaviour cloning loss. However, the *cliff*-type loss framework used to study this does not capture this behaviour, as it is small in a neighbourhood of radius $\epsilon $ and is very large outside. In my opinion, this framework it too tailor-made to study the benefits of EMA and does not reveal the real reason behind its working mechanism.

**Questions:**

It is mentioned in comments after proposition 3.1 that there is a good subset in parameter space that do not experience this worst-case error amplification. Does such good neighbourhood exists around any element in the parameter space or only around elements with specific properties ?

---

> ### Author Response · Authors · 2023-11-17
> **Response to Review**
>
> We thank the reviewer for their feedback. We address their concerns below:
>
> We would like to clarify our exposition of GVA. First, note that our mechanistic understanding of GVA arises from *both* careful empirical findings and theoretical explanations. To reiterate, the empirical findings for GVA are:
> (a) More data and/or regularization **do not mitigate GVA**.
> (b) Increasing batch size improves, but does not entirely ameliorate, GVA.
> (c) EMA, which reduces iterate variance, nearly entirely fixes GVA
>
> Through our analysis of local perturbations around imitator policies at different stages of training (Fig 1 right and Fig 13), we observe that while the *validation loss* landscape is relatively insensitive to parameter perturbations, the *policy evaluation loss* landscape is very sensitive to these.
>
> In summary, we conclude that (i) evaluation loss is highly sensitive to changes in parameters and (ii) that methods which reduce variance in iterates/gradients seem to ameliorate this sensitivity.
>
> The role of our theory is to show that the above hypothesis is mathematically plausible, and revealed even without considerations of non-convex representation learning.
>
> **Proposition 3.1**:   We wish to clarify the precise statement of Proposition 3.1, which is intended as an example of exponential error amplification but not intended to be perfectly representative of what we observe in experiments. Proposition 3.1 **does not** state that for every delta, the BC error is exponential-in-H larger than validation. Instead, it shows that
> For all delta > O(1/H), there **exists** a perturbed policy of error delta which witnesses exponential blow up
> For **all** delta < O(1/H), there is no exponential blow up.
>
> The point of the proposition is that, to ensure no exponential blowup in  the worst case, for long rollout horizons, the BC loss must be made sufficiently small. It may be possible, with long training times and sufficient learning rate decay, that one could make the BC loss sufficiently small as to ensure the rollout reward does not suffer from exponential blowup. Indeed, our experiments corroborate this qualitative finding (point to Fig): that if we train for a longer time with sufficient learning rate decay, we do get consistently large rollout reward. This, however, is wasteful in terms of training time, as EMA does just as well without excessive numbers of training epochs.
>
> **Cliff Loss:**
>
> Surprising, we find that GVA is not a major obstacle when learning to control linear systems (which includes the example in Prop 3.1). If we examine how the cloned policy behave during rollout time, we found (and will include figures in the final revision) that the deviation between cloned policy and expert, under the rollout of the cloned policy, starts to deviate wildly after a specific time step in the RL environment. We interpret this as the presence of dynamical stiffness in the task, e.g. the contact that the locomoting agent makes with the floor surface. Importantly, MuJoCo environments are well approximated by  Piecewise Affine Systems, a classical model for contact dynamics.
>
>
> In the revision, we will clarify that the cliff loss is a simple and tractable mathematical abstraction of the above phenomenon. The cliff models a “two-region” contact system, where ‘falling off the cliff’ corresponds to a change in region.
>
> **Questions**
>
> This is a good question that we will address in the revision.  To be precise, in our particular example in the proof of Proposition 3.1, there are good subsets only around parameters that are `marginally stable' and thus many policies do not have these good neighborhoods.  We believe this phenomenon to persist in the general case, with Figure 1 (right) and Figure 13 providing empirical evidence in favor of this claim in the MuJoCo Walker environment.

---

### Official Review · Reviewer_1ahA · 2023-11-02

**Soundness:** 4 excellent
**Presentation:** 4 excellent
**Contribution:** 3 good
**Rating:** 8
**Confidence:** 3

**Summary:**

The paper examines a known problem in behavior cloning. This problem occurs during the training of such agents, where the validation criterion (return of an episode under long horizon rollouts) has large variance at all training steps, while the training surrogate loss has small, making it hard to perform model selection.

The authors perform a theoretical and empirical study of the origins of this problem. They argue that the observed variance is due to training instability, instead of insufficient dataset size, and they name the phenomenon as *Gradient Variance Amplification (GVA)*. They suggest that alternative training algorithms might not have the same issue. For this reason, they propose a very simple fix, which is to track the exponential moving average (EMA) of the parameter iterates of the SGD trained model. They demonstrate the effectiveness of the approach by performing experiments under various environments and by ablating design choices of the proposed EMA intervention.

In addition, they argue that GVA generally exists whenever agents are expected to operate by conditioning themselves in their past output (or effects of them). This definition aligns with conditional language modelling, and they argue with experiments that language generation quality in LLMs is similarly affected and that EMA can also help mitigate the problem there.

**Strengths:**

* The paper is well-organized, well-written and well-argumented. It introduces and motivates the problem sufficiently and discussed related work in depth.

 * They pose a single clear question at the heart of the problem: Do we just need more data? Or is there in the training algorithm that amplifies variance for the validation criterion? Their ablations convince that the second is the case (section 3.1/figure 2), and they provide with theoretical insight that the return in horizon H can be exponentially large for some suboptimal policy in a smooth neighborhood around the expert, even though for all of those suboptimal policies in the neighborhood the surrogate training loss (of behavioral cloning) is small.

 * They introduce various decision choices around implementing EMA as a candidate solution, and they ablate many of them, demonstrating that the problem is effectively mitigated in many RL environments.

 * They make connections to the relevance of the problem to other topics in ML, such as autoregressive language modelling.

**Weaknesses:**

1. Ablations regarding the design choice of $\gamma_t$ scheduling are missing. In the paper, a polynomial decay is used and ablated, but other schedules have also been used with EMA (like cosine). It would be nice to understand a bit better why this decay is important and how to design one which is tailored at the problem at hand.
2. Middle of Figure 4 misses y-axis values, which is important in order to know at which scale are we seeing the zoom at.
3. In the context of autoregressive language generation: Validation perplexity perhaps does not show the existence of GVA problem here in the most clear way. Ideally, some equivalent to a metric of generation quality of horizon H autoregressive rollouts should have been used instead.

**Questions:**

1. In **Proposition 3.1**, I guess that $\Delta$ refers to two different smooth “error functions” in each of the two inequalities.

### Typos

2. **Section 4.3**: “Here training loss is convex, but rollout reward is not.” The training loss is indeed convex, but the rollout reward is (still) concave, even if it is discountinuous. Is that right?
3. Later in **Section 4.3**: “SGD iterates:” $\theta_{t+1} = \theta_{t} - …$ instead of $\theta_{t+1} = \theta_{t+1} - …$
4. **Appendix A.1**/**Role of SGD noise**: “It is now well appreciate*d* that gradient noise facilitates the escape *from* saddle points.”

---

> ### Author Response · Authors · 2023-11-17
> **Response to Review**
>
> We thank the reviewer for their detailed and positive feedback.  We also would like to thank the reviewer for pointing out typos; we will fix them for the revision.  We respond to each of the points individually:
>
> 1. Anecdotally, we find that the effect of EMA on GVA is relatively insensitive to the specific decay schedule of the EMA.  We have not experimented with more complicated, non-monotonic decays, but will do this for one of the MuJoCo experiments and include the result in the final paper.
>
> 2. Thank you, this is a good point.  We will add the labeling in the final paper.  Note that for evaluation of the current paper, the approximate scaling is implied by the location of the highlighted boxes on the left of Figure 4; for example, the leftmost zoomed in box ranges from validation loss around 1.01 to loss around 1.14.
>
> 3. We agree that changes in validation perplexity are orthogonal to GVA (and we have pointed this out in the paper, which we will attempt to further clarify in the revision). As a more direct parallel to the imitation learning experiments, we also studied full-sequence switching costs in argmax decoding. That GVA appears in perplexity is interesting in its own right, in that perplexity is not intrinsically autoregressive (evaluating the likelihood of entire text generation); note that this also occurs in the MuJoCo setup (see Figure 6b). We are excited to understand how GVA is so widespread as to affect even the 1-step loss, as well as developing more refined metrics, to future work. We emphasize that our LM results are preliminary and primarily illustrative of the phenomenon of error amplification when a 1-step objective is used as a proxy in closed-loop settings.
>
> **Questions**
>
> 1. Yes, in each inequality in the proposition, the Delta is allowed to depend on the precise parameters and thus may be different.  Note that this is a stronger statement than if they must be the same.

---

### Meta-Review · Area_Chair_Nw3L · 2023-12-05

**Metareview:**

The paper identifies the problem of gradient variance amplification in the setting of behavioral cloning, claiming that updates to the policy network have an outsized impact on return, despite affecting the behavioral cloning loss minimally. The authors propose tackling this issue with exponential moving averages (EMA) of parameters during training. All three reviewers agree that this is an important problem to be addressed and reviewers 1ahA and xe8u acknowledge that EMA is a successful strategy to address this problem. Reviewers 1ahA and xe8u also identify several weaknesses of the paper, stemming from some ablations, the use of perplexity, and some confusion around the theoretical analysis. The authors have generally addressed these weaknesses in their responses. I also have several concerns about whether these findings generalize beyond the MuJoCo settings analyzed in the paper; the authors also include results on language modeling, but acknowledge that these are “preliminary.” In a future version of this paper, it would be nice to see additional results in other control domains.

**Justification For Why Not Higher Score:**

While the authors have addressed an important problem with a fairly simple solution, I did not find the quantitative results to be highly impressive: the main outcome of the paper is in reduced variance in return on a subset of MuJoCo environments. While this is interesting and useful, this does not strike me as warranting a spotlight or oral.

**Justification For Why Not Lower Score:**

The authors have addressed an important problem with a fairly simple solution. I find these papers to generally be useful for impacting the scientific community. Likewise, the authors have carried out a significant amount of analysis, both theoretical and empirical, to demonstrate the phenomenon of gradient variance amplification and the benefits of EMA.

---

### Decision · Program_Chairs · 2024-01-16

Accept (poster)